# The septin cytoskeleton is required for plasma membrane repair

M Isabella Prislusky [1], Jonathan G T Lam[1], Viviana Ruiz Contreras[1,2], Marilynn Ng[1], Madeline Chamberlain[1], Sarika Pathak-Sharma[1], Madalyn Fields[1], Xiaoli Zhang[3], Amal O Amer [1] & Stephanie Seveau [1]✉

## Abstract

**Plasma membrane repair is a fundamental homeostatic process of eukaryotic cells. Here, we report a new function for the conserved cytoskeletal proteins known as septins in the repair of cells perforated by pore-forming toxins or mechanical disruption. Using a silencing RNA screen, we identified known repair factors (e.g. annexin A2, ANXA2) and novel factors such as septin 7 (SEPT7) that is essential for septin assembly. Upon plasma membrane injury, the septin cytoskeleton is extensively redistributed to form submembranous domains arranged as knob and loop structures containing F-actin, myosin IIA, S100A11, and ANXA2. Formation of these domains is Ca$^{2+}$-dependent and correlates with plasma membrane repair efficiency. Super-resolution microscopy revealed that septins and F-actin form intertwined filaments associated with ANXA2. Depletion of SEPT7 prevented ANXA2 recruitment and formation of submembranous actomyosin domains. However, ANXA2 depletion had no effect on domain formation. Collectively, our data support a novel septin-based mechanism for resealing damaged cells, in which the septin cytoskeleton plays a key structural role in remodeling the plasma membrane by promoting the formation of SEPT/F-actin/myosin IIA/ANXA2/S100A11 repair domains.**

**Keywords** Plasma Membrane Repair; Septin; F-actin; Annexin A2; Pore-forming Toxin

**Subject Categories** Cell Adhesion, Polarity & Cytoskeleton; Membranes & Trafficking

## Introduction

The plasma membrane of eukaryotic cells forms a biophysical barrier that separates the cell from its external environment. Any mechanical or biochemical perturbations that compromise the integrity of the plasma membrane can be lethal for the cell. Therefore, robust plasma membrane repair mechanisms maintain cell and tissue homeostasis (Cooper and McNeil, 2015; Jimenez and Perez, 2017). Excessive plasma membrane damage or defective plasma membrane repair is involved in many pathological conditions including muscular dystrophy, ischemia-reperfusion, heart failure, chronic inflammation, and neurodegenerative diseases (Dias and Nylandsted, 2021). Pathogens including parasites, bacteria, and viruses use diverse strategies to perforate the host cell plasma membrane and exploit the host cell repair responses to successfully infect their host (Andrade, 2019; Ayyar et al, 2023; Banerji et al, 2021; Barisch et al, 2023; Bouillot et al, 2018; Thapa et al, 2020). In particular, the bacterial pathogen *Listeria monocytogenes* uses as a major virulence factor the pore-forming toxin listeriolysin O (LLO). LLO forms a pore complex across cholesterol-rich cytoplasmic and endosomal membranes promoting host cell invasion and cell-to-cell spreading (Osborne and Brumell, 2017; Petrisic et al, 2021; Seveau, 2014; Vadia et al, 2011). LLO-mediated plasma membrane perforation is moderate enough that effective plasma membrane repair maintains cell viability to support the intracellular lifecycle of the pathogen (Cassidy et al, 2012; Chen et al, 2018; Vadia and Seveau, 2014). How *L. monocytogenes*-infected cells repair their plasma membrane is not fully understood. LLO belongs to the cholesterol-dependent cytolysins (CDC) /Membrane Attack Complex/Perforin (MACPF) superfamily (Dunstone and Tweten, 2012). Several CDC members including streptolysin O (SLO) and perfringolysin O (PFO) have been successfully used to study conserved plasma membrane repair processes (Idone et al, 2008; Ray et al, 2022). The repair of CDC-perforated cells was proposed to involve the internalization of the damaged plasma membrane downstream from the release of the lysosomal enzyme acid sphingomyelinase (Idone et al, 2008; Tam et al, 2010). Other repair models proposed the shedding of vesicles containing CDC pores by annexin- or endosomal sorting complexes required for transport (ESCRT)-III-dependent processes (Atanassoff et al, 2014; Jimenez et al, 2014; Ray et al, 2022; Wolfmeier et al, 2016). Importantly, all these mechanisms also repair mechanical damage. Therefore, using LLO as a tool to inflict plasma membrane injury has the potential to uncover general repair machineries.

To uncover effectors that mediate the repair of LLO-perforated cells, we performed a siRNA screen targeting 245 protein-coding genes controlling endocytosis, exocytosis, and intracellular

[1]Department of Microbial Infection & Immunity, Wexner Medical Center, The Ohio State University, Columbus, OH, USA. [2]Grupo Investigaciones Biomédicas, Universidad de Sucre, Sincelejo, Sucre, Colombia. [3]Department of Biomedical Informatics, The Ohio State University, Columbus, OH, USA. ✉E-mail: Seveau.1@osu.edu

trafficking. The screen identified 47 candidates including previously known plasma membrane repair proteins such as calpain 1, the acid sphingomyelinase, several annexins, and components of ESCRT-III (Draeger et al, 2011; Jimenez et al, 2014; Mellgren et al, 2007; Tam et al, 2010). The screen also identified novel plasma membrane repair candidates. Of those, we focused on septin 7 (SEPT7) because the septins are highly conserved eukaryotic GTP-binding proteins described as the fourth component of the cytoskeleton after F-actin, microtubules, and intermediate filaments (Mostowy and Cossart, 2012). Human septins include 13 members which assemble into hetero-hexamers (SEPT-2,-6,-7,-7,-6,-2) and hetero-octamers (SEPT-2,-6,-7,-9,-9,-7,-6,-2) that further organize into filaments or rings in association with F-actin, microtubules or the plasma membrane (Benoit et al, 2023; Shuman and Momany, 2021). They form filaments and rings that determine the shape, curvature, and properties of the plasma membrane (Benoit et al, 2023; Nakamura et al, 2023). The septins control major cellular functions including cytokinesis, cell motility, tissue morphogenesis, and host-pathogen interactions (Longtine et al, 1996; Spiliotis and Nakos, 2021). However, any potential function for this protein family in plasma membrane repair was completely unknown. SEPT7 plays a central role in the formation of septin filaments as it is the only non-redundant septin (Brognara et al, 2019; Kinoshita, 2003). Our data show for the first time that SEPT7 plays a general role in plasma membrane repair of cells perforated by pore-forming toxins and mechanical wounding. Diverse microscopy methods, including super-resolution microscopy, established that the septin cytoskeleton reorganizes in injured cells to form knob and loop structures in subdomains of the plasma membrane together with the contractile actomyosin cytoskeleton. Mechanistically, our data support a novel model in which the septins act as scaffolds required to promote the formation of plasma membrane repair domains containing contractile F-actin and annexin A2.

# Results

## A siRNA screen identifies novel candidate genes controlling plasma membrane repair of listeriolysin O (LLO)-perforated cells

To identify the machineries involved in plasma membrane repair of LLO-perforated cells, we transfected HeLa cells with a library of siRNAs targeting 245 human genes controlling endocytosis, exocytosis, intracellular trafficking, and the cytoskeleton (Dataset EV1). Each gene was targeted for 72 h by a cocktail of three siRNAs with non-overlapping sequences. A fluorescence-based assay was then used to assess plasma membrane integrity (Lam et al, 2019). In this assay, siRNA-treated HeLa H2B-GFP cells (expressing green-fluorescent histone 2B) were exposed, or not, to sub-lytic LLO concentration (Fig. EV1B and Movies EV1, EV2) for 30 min at 37 °C (Lam et al, 2018; Vadia et al, 2011; Vadia and Seveau, 2014). The influx of the membrane-impermeant dye TO-PRO-3 into damaged cells was measured as a readout for plasma membrane integrity. We identified 57 genes in which silencing significantly affected membrane integrity, either negatively (47) or positively (10) (Table EV1). Several of the 47 genes encode proteins previously shown to control plasma membrane repair such as the annexins (ANXA2, ANXA7, and ANXA11), the lysosomal enzyme

acid sphingomyelinase (SMPD1), calpain 1 (CAPN1), and components of the ESCRT-III machinery (CHMP2) (Jimenez et al, 2014; Mellgren et al, 2007; Satoh et al, 2002; Sonder et al, 2019; Tam et al, 2010). We also identified genes that encode proteins involved in membrane fusion (4 synaptotagmins and 12 SNAREs), lysosome biogenesis and functions (HPS3, HPS5, M6PRBP1, AP3S2, AP3M2, AP3S1, and Rab7) and proteins involved in exocytosis and the secretion of exosomes (EXOC1, EXOC6, Rab27A, Rab27B, and Rab11A). Finally, we identified genes encoding guanine nucleotide exchange factors (GBF1, IQSEC1) and proteins controlling the actin cytoskeleton (ABl1 and SEPT7) that were not previously known to control plasma membrane repair. Of these, we focused on SEPT7 because septins are a ubiquitous family of cytoskeletal proteins not previously known to control plasma membrane repair.

## The septins (SEPT6 and SEPT7) are required for plasma membrane repair

To confirm the role of the septins in plasma membrane repair, we first measured the knockdown efficiencies of each siRNA used in the screen to silence septins' expression (SEPT2, 6, 7, and 9) (Figs. 1A and EV1A). For each siRNA targeting SEPT7, SEPT9, and SEPT2, septin protein levels were decreased by 90–95% compared to control cells treated with non-targeting siRNA (Ctr. siRNA). However, SEPT6 expression was only decreased by about 35–40% for two siRNA whereas the third siRNA had no effect (Fig. 1A). Silencing SEPT7 expression markedly reduced SEPT2, SEPT6, and SEPT9 protein levels (Fig. EV1E) and the formation of septin filaments (Fig. EV1G) (Kremer et al, 2005). However, silencing SEPT2, SEPT6, and SEPT9 had little effect on the expression of the other tested septins (Fig. EV1C–F) and the formation of septin filaments (Fig. EV1G). We then evaluated the effect of silencing septin expression on the plasma membrane integrity of LLO-perforated cells. Silencing SEPT7 with each siRNA resulted in a significant loss in plasma membrane integrity of LLO-treated cells (Fig. 1B). This was unlikely due to an off-target effect because the three non-overlapping siRNAs were tested independently. To distinguish if the loss in plasma membrane integrity of SEPT7-deficient cells was due to an increase in plasma membrane perforation by LLO or to a defect in plasma membrane repair, we also measured the integrity of cells incubated in $Ca^{2+}$-free medium, an experimental condition preventing plasma membrane repair. In $Ca^{2+}$-free medium, Ctr. siRNA-treated and SEPT7-deficient cells were similarly damaged leading to the conclusion that SEPT7-deficiency does not affect plasma membrane perforation by LLO (Fig. 1D). Furthermore, we previously showed that extracellular $Ca^{2+}$ does not affect LLO pore formation (Vadia et al, 2011). Thus, SEPT7 plays a significant role in plasma membrane repair. Although the siRNA screen did not reveal a role for SEPT6, we found that SEPT6-siRNA1 and SEPT6-siRNA3, which both reduced SEPT6 expression, significantly impaired plasma membrane repair (Fig. 1C,D). As observed in the screen, silencing SEPT2 and SEPT9 expression did not affect plasma membrane integrity (Fig. EV1A'). We next generated a stable HeLa cell line with doxycycline-inducible expression of a short hairpin RNA (shRNA) targeting SEPT7 (DiSEPT7-shRNA1, Fig. 1E). The targeting sequence was selected from the Genetic Perturbation Platform (Broad Institute) for its high specificity and was not overlapping with any of the three SEPT7 siRNA sequences used in

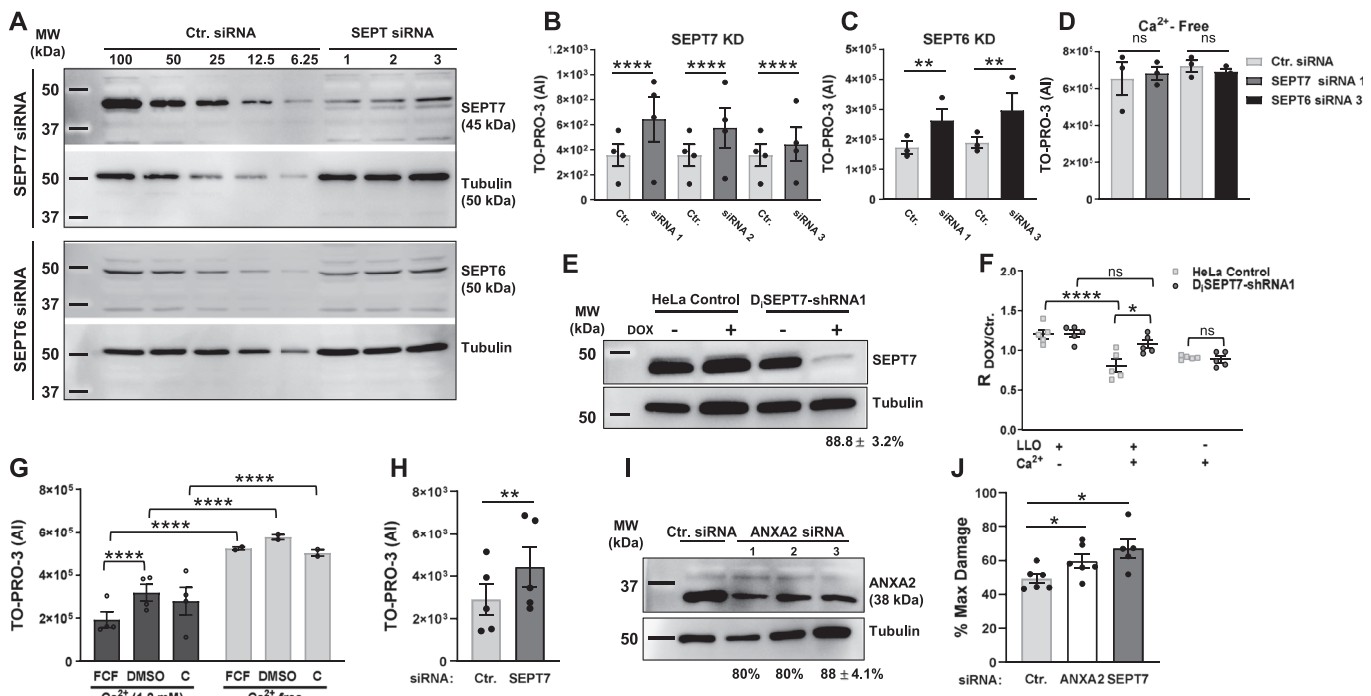

**Figure 1. SEPT6 and SEPT7 are required for efficient plasma membrane repair.**

(A) HeLa cells were transfected with non-targeting siRNA (Ctr. siRNA) or with siRNAs targeting SEPT6 and SEPT7 (Dataset EV1). After 72 h, cells were lysed and analyzed by SDS-PAGE and immunoblotting. Serial dilutions of Ctr. (100% to 6.25%) and undiluted septin siRNA-treated (100%) cell lysates were loaded in the gel to facilitate the quantification of the KD efficiencies. Blots are representative of at least 5 independent experiments ($N = 5$). KD efficiencies for SEPT6 were 39.5% ± 8.6; 3.6% ± 3.6; and 45.5% ± 12.5 for siRNA1, 2, and 3, respectively (average ± standard error of mean (SEM)). KD efficiencies for all SEPT7 siRNAs were >90%. (B–D) Ctr. and SEPT-specific siRNAs treated cells were exposed to LLO (0.5 nM) for 30 min in M1 (1.2 mM $Ca^{2+}$) (B, C) or M2 ($Ca^{2+}$-free) (D), supplemented with TO-PRO-3. Data are the average TO-PRO-3 fluorescence intensities expressed in arbitrary units (AI) ± SEM of at least $N = 3$ independent experiments at time point 30 min. (E) Control HeLa cells and D$_i$SEPT7-shRNA HeLa cells were cultured for 72 h in the presence (+) or in the absence (−) of Dox (50 μg/ml) and were tested for SEPT7 KD efficiency by immunoblotting. The presented blot is representative of $N = 8$ independent experiments with a KD efficiency of 88.75 ± 3.2% (Average ± SEM). (F) Control HeLa cells and D$_i$SEPT7-shRNA1 HeLa cells were cultured for 72 h with or without DOX and then washed to remove the DOX. Cells were exposed, or not, to 0.5 nM LLO for 30 min in M1 or M2 supplemented with TO-PRO-3. Data are expressed as the average ratio of the TO-PRO-3 fluorescence intensity of DOX-treated over TO-PRO-3 fluorescence intensity of non-treated (Ctr.) cells (R$_{DOX/Ctr.}$) ± SEM of $N = 5$ independent experiments at time point 30 min. (G) HeLa cells were pre-treated with 100 μM FCF (in DMSO), vehicle DMSO, or untreated, C for 16 h. Cells were then subjected to the repair assay for 30 min with 0.5 nM LLO in M1 or M2 with TO-PRO-3. Due to FCF reversibility, FCF and DMSO were added to the buffers during the repair assay. Data are the average ± SEM of $N = 2$ independent experiments for the $Ca^{2+}$-free condition and $N = 4$ independent experiments for 1.2 mM $Ca^{2+}$ condition at 30 min. (H) HeLa cells were treated for 72 h with Ctr. siRNA or SEPT7-siRNA 1 and were exposed to PLY for 30 min in M1, supplemented with TO-PRO-3. Data are the average TO-PRO-3 fluorescence intensity expressed in arbitrary units (AI) ± SEM of $N = 5$ independent experiments at time point 30 min. (I) HeLa cells were transfected with Ctr. siRNA or ANXA2-siRNAs (Dataset EV1). After 72 h, cells were lysed. Control cell lysates (100%) and annexin A2 siRNA-treated cell lysates (100%) were loaded in the gel to facilitate the quantification of the KD efficiencies. The blot is representative of 6 independent experiments ($N = 6$, Average ± SEM) for ANXA2 siRNA 3, and $N = 1$ for ANXA2 siRNA 1 and siRNA 2. (J) Ctr.-, SEPT7-siRNA 1-, and ANXA2-siRNA 3-treated cells were mechanically wounded in M1 or M2 buffers. The number of Emerald cells were counted to represent the number of damaged cells and the number of Ruby cells that were originally green were counted to represent the number of not recovered cells. The data is expressed as the number of non-recovered cells (Ruby)/the number of Damaged cells (Emerald) which have been normalized to the $Ca^{2+}$-free condition, %Max Damage ± SEM of $N = 6$ independent experiments. Data Information: TO-PRO-3 intensity was measured by fluorescence microscopy (B, F, H) or by spectrofluorometry (C, D, G). In (B–D) and (F–G), data were $\log_{10}$ transformed and analyzed using linear mixed-effects models and a Holm's procedure was used to control for multiple comparisons in (B). (*$P < 0.05$, **$P < 0.01$, ***$P < 0.001$, and ****$P < 0.0001$). In (H) and (J), data were analyzed by a one-tailed Students Paired T-Test (*$P < 0.05$, **$P < 0.01$). Source data are available online for this figure.

the screen. We verified that doxycycline had no effect on LLO pore formation (Fig. EV1H) and then repeated the repair assay comparing the ratio intensity of doxycycline-treated to non-treated D$_i$SEPT7-shRNA1 and control HeLa cells in the presence or absence of extracellular $Ca^{2+}$. Only in the presence of LLO and extracellular $Ca^{2+}$, plasma membrane integrity of SEPT7-deficient cells was significantly impaired in comparison to control cells, which confirmed the role of SEPT7 in plasma membrane repair of LLO-injured cells (Fig. 1F).

As a third experimental approach to establish the role of the septin cytoskeleton in plasma membrane repair, we used the pharmacological agent forchlorfenuron (FCF) known to reversibly

bind to and stabilize septin oligomers and/or filaments (DeMay et al, 2010; Hu et al, 2008; Iwase et al, 2004). In the presence of $Ca^{2+}$, cell treatment with FCF significantly decreased plasma membrane damage of LLO-treated cells (Figs. 1G and EV1I,J). In $Ca^{2+}$-free medium, FCF treatment did not affect cell susceptibility to LLO-perforation (Figs. 1G and EV1I) and FCF had no direct effect on LLO pore formation (Fig. EV1K). Together, these data indicate that stabilization of the septin cytoskeleton by FCF significantly increases the plasma membrane repair efficiency of LLO-treated cells.

To ensure that the role of SEPT7 in plasma membrane repair was not specific to LLO-perforated cells, Ctr.- and SEPT7-siRNA

treated cells were exposed to the pore-forming toxin pneumolysin (PLY), another CDC produced by the pathogen *Streptococcus pneumoniae* (Walker et al, 1987). As previously observed with LLO, SEPT7 expression was required for effective plasma membrane repair of PLY-perforated cells (Fig. 1H). We next subjected cells to mechanical wounding using glass beads. As a positive control, we silenced the expression of ANXA2 (Fig. 1I), which was previously shown to repair mechanical wounds (Jaiswal et al, 2014). We found that SEPT7 deficiency significantly impaired plasma membrane repair of mechanically wounded cells (Fig. 1J). Collectively, our data demonstrate a general role for SEPT7 in plasma membrane repair of toxin-perforated and mechanically wounded cells.

## Plasma membrane perforation remodels the septin cytoskeleton together with cortical F-actin, myosin-IIA, annexin A2, and S100A11

The septin cytoskeleton is known to associate with and to regulate the actin cytoskeleton. Therefore, we studied the distribution of the septins relative to F-actin. HeLa cells were treated, or not, with LLO for 5 to 15 min at 37 °C, followed by chemical fixation and fluorescent labeling of F-actin, SEPT2, SEPT7, or SEPT9. We found that SEPT2, SEPT7, and SEPT9 display similar labeling patterns in all experimental conditions (Figs. 2A and EV2A,B). This was expected since septins of different groups co-assemble to form hexamers (SEPT-2,-6,-7,-7,-6,-2) and octamers (SEPT-2,-6,-7,-9,-9,-7,-6,-2) (Benoit et al, 2023). We observed that septin filaments predominantly colocalized with actin stress fibers in control untreated cells (Figs. 2Aa, EV2Aa, 2Ba). Strikingly, in LLO-treated cells, a fraction of the septin cytoskeleton clearly dissociated from the actin stress fibers (Figs. 2Ab, EV2Ab, 2Bb) and reorganized into knob- and loop (or ring)-like structures in association with cortical F-actin (Figs. 2Ac, 2Ad, EV2Ac, 2Bc). The number of these new septin structures significantly increased over time from 5 to 15 min post-LLO-exposure (Fig. 2B,C). Of note, septin dissociation from the actin stress fibers was only partial, as some septins were still associated with actin stress fibers in LLO-treated cells (Figs. 2Abii, EV2Abi, 2Bbii).

Importantly, in cells wounded by glass beads, septin filaments rearranged in a similar fashion at the site of membrane damage, depicted by the uptake of Emerald fluorescent dextran (Fig. EV2C,D). This redistribution was not observed in neighboring, non-damaged cells.

We further analyzed the properties of the redistributed septin cytoskeleton in LLO-injured cells, which is a more amenable experimental system. As shown in Fig. 3A, the knob- and loop-like septin/F-actin structures strongly colocalized with myosin-IIA indicating that the newly formed structures are contractile. As the septins have been proposed to regulate plasma membrane properties (Benoit et al, 2023), we thought that the septins may control the organization of plasma membrane repair domains. To test this hypothesis, we investigated if the remodeled septins co-distribute with known plasma membrane repair machineries. Our screen identified that annexins and the ESCRT-III are important for the integrity of LLO-perforated cells which is in accordance with the literature (Koerdt et al, 2019; Sonder et al, 2019). We observed that the ESCRT-III ALG-2-interacting protein X (ALIX) formed larger puncta in LLO-treated cells in comparison to control cells, but rarely colocalized with the septins (Fig. EV3A). However,

in the presence of LLO, there was a clear redistribution of ANXA2, which colocalized with the septins and F-actin, as observed in cells transiently expressing ANXA2-GFP and in cells fluorescently labeled with anti-ANXA2 antibodies (Fig. 3B,B'). The co-distribution of ANXA2, the septins, and F-actin was observed throughout the cell surface and within the knob and loop structures (Fig. 3B'). Quantitative analysis showed that the septins display a high level of colocalization with both F-actin and ANXA2 (Fig. 3C). It was shown by others that ANXA2, the protein S100A11, and cortical F-actin act together to repair mechanically ruptured cells by excision (Jaiswal et al, 2014). In accordance with this model, we show that SEPT, ANXA2, and F-actin colocalize with S100A11 after LLO exposure (Fig. EV3C). In conclusion, the septin cytoskeleton, visualized via labeling of SEPT2, SEPT7, or SEPT9, is remodeled in damaged cells to colocalize with contractile actomyosin cytoskeleton, ANXA2, and S100A11.

## Septins are redistributed at the cell surface to form circular filaments intertwined with F-actin and are associated with annexin A2 in injured cell

To establish the localization of the remodeled septin cytoskeleton relative to the plasma membrane, HeLa cells were transiently transfected to express the fluorescent plasma membrane marker Lck-mTurquoise2 (Zacharias et al, 2002). Cells were exposed to LLO for 15 min at 37 °C, then fixed and labeled for F-actin and SEPT2. As shown in Fig. 4A,A', SEPT2 and F-actin knobs and loops colocalized with the plasma membrane marker in LLO-injured cells. As a second approach, cells were analyzed by confocal microscopy to better localize the remodeled septin cytoskeleton relative to F-actin and ANXA2 in LLO-treated cells. Z-depth representation of SEPT2-, F-actin-, and ANXA2-labeled HeLa cells showed that knob and loop structures are formed on the upper plasma membranes of LLO-treated cells, but not in control cells (Fig. 4B). Next, FCF-treated cells (to increase the septin remodeling) were exposed to LLO, fixed, and labeled for SEPT2, F-actin, and ANXA2. Confocal images were displayed in 3D with z-depth coding confirming that the SEPT2/F-actin/ANXA2 knobs and loops are present on the cell surface and often protrude outward (Fig. 4C–D'). The SEPT2 and F-actin structures are in close association with ANXA2, forming rings and spirals (Fig. 5A,Ai). The diameter of the rings and spirals ranged from 2 to 6 μm and the structures could be as high as 4 μm, independently of FCF pre-treatment (Fig. 5Ai). Finally, we performed super-resolution microscopy imaging to better resolve the relative distribution of F-actin, SEPT2, and ANXA2 (Fig. 5B). Images showed that SEPT2 and F-actin are closely associated forming circular, intertwining filaments. ANXA2 is distributed as patches or elongated structures closely connected to SEPT2 or F-actin and which density increases towards the top of the structure. Collectively, these data show that in response to plasma membrane injury, the septin cytoskeleton is remodeled together with F-actin to form submembranous, inter-twined filaments in close connection with ANXA2.

## Septin redistribution is functionally correlated with plasma membrane repair efficiency

Our data strongly support the role of the septins in forming submembranous structures that promote plasma membrane repair.

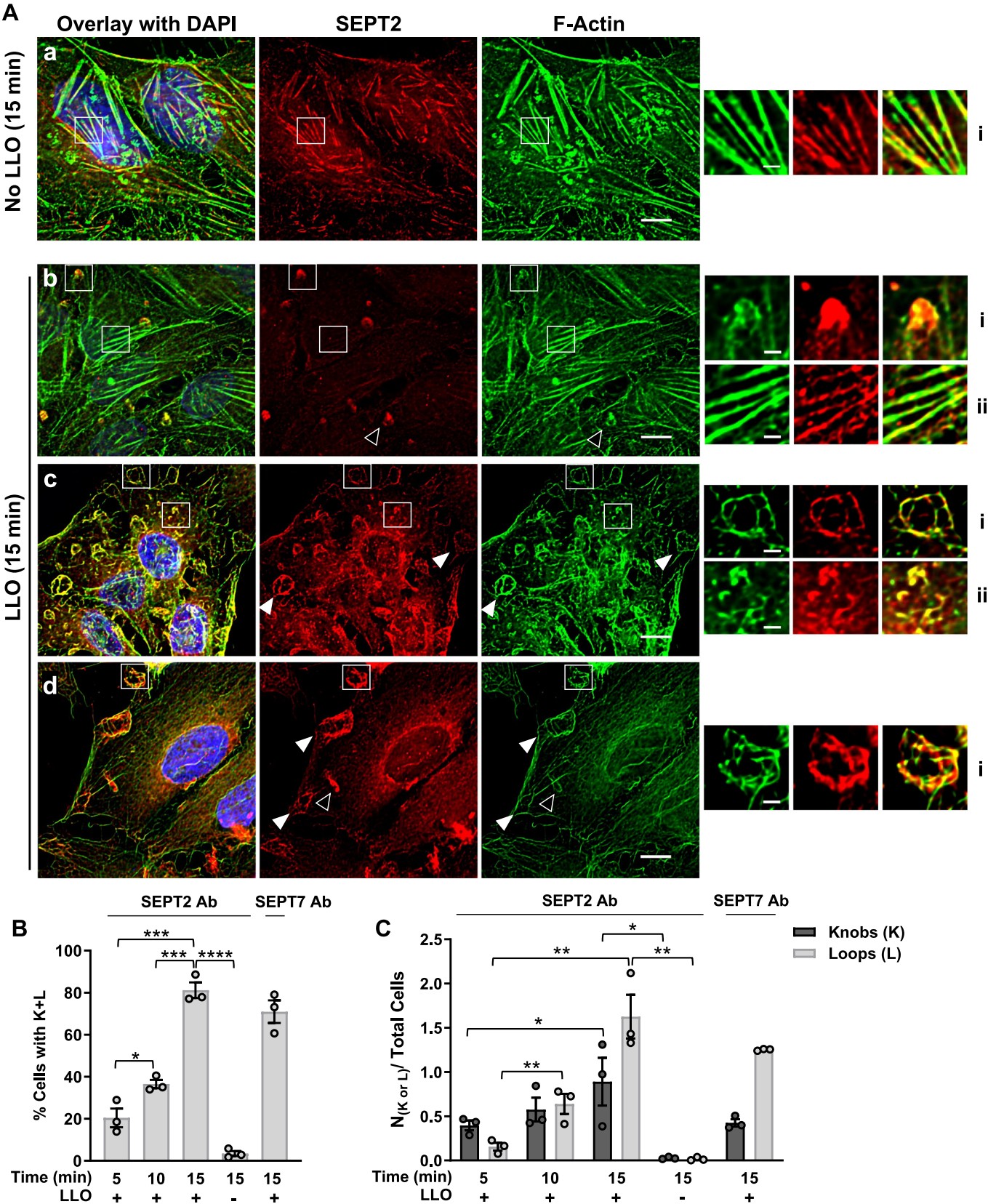

**Figure 2. The septin cytoskeleton is remodeled in LLO-perforated cells.**

HeLa cells were incubated without (No LLO) or with 0.5 nM LLO for 5–15 min in repair permissive buffer M1. Cells were fixed, permeabilized and fluorescently labeled with anti-septin primary Abs (Alexa Fluor 568-conjugated secondary), F-Actin (Alexa Fluor 488- or 647-conjugated phalloidin), and nuclei (DAPI). (A) Representative images at time point 15 min. Septin fluorescence images are presented with the same intensity scaling to visualize the partial dissociation of septin filaments from the actin stress fibers. Selected regions were magnified and the septin fluorescence display was artificially enhanced only in Abii to show that a portion of the septin cytoskeleton remains associated with actin stress fibers. Scale bars are 10 µm (Aa–d) and 2 µm in magnified images. Images were acquired by z-stack widefield microscopy, denoised, deconvolved, and presented as the best focus images, except for (Aa) and (Ab) which are single planes focused on actin stress fibers. Representative knob and loop structures are indicated by unfilled and filled arrowheads, respectively. (B) The percentage of cells with septin knobs and loops (%Cells with K + L) and (C) the average number of knobs or loops per total cells ($N_{(K\ or\ L)}$/Total cell) were enumerated based on SEPT2 or SEPT7 fluorescence. A total of 300–750 cells were analyzed in each experimental condition, error bar show SEM of at least $N = 3$. Data were analyzed by a one-tailed Students Unpaired T-Test (*$P < 0.05$, **$P < 0.01$, ***$P < 0.001$, and ****$P < 0.0001$). Source data are available online for this figure.

In this model, septin remodeling should correlate with the plasma membrane repair efficiency, i.e., septin remodeling should decrease when plasma membrane repair is prevented in $Ca^{2+}$-free medium, and conversely, septin remodeling should increase when plasma membrane repair is enhanced in the presence of FCF. Consistent with this model, in $Ca^{2+}$-free medium, the septins did not dissociate from the actin stress fibers, the formation of septin/F-actin knobs and loops was markedly decreased, and ANXA2 redistribution was impaired compared to cells exposed to LLO in $Ca^{2+}$-containing medium (Fig. 6A). Also consistent with this scenario, FCF treatment led to a more pronounced formation of septin loops in LLO-treated cells, which all colocalized with F-actin and ANXA2 (Figs. 4C–D', 6D and EV4). Quantitative analyses confirmed that the number of septin/F-actin knobs and loops per cell and the percentage of cells presenting such structures were significantly decreased in LLO-treated cells in $Ca^{2+}$-free medium, and significantly increased in LLO-treated cells in the presence of FCF (Fig. 6B,C). In mechanically wounded cells, FCF pre-treatment increased septin remodeling (Fig. EV2C), whereas septins were not remodeled in $Ca^{2+}$-free medium (Fig. EV2D Together, these data establish that the redistribution of the septin cytoskeleton with F-actin and ANXA2 is functionally correlated with the plasma membrane repair efficiency, further supporting the role of these submembranous domains in plasma membrane repair.

### The septins control both the remodeling of cortical F-actin into knob and loop structures and ANXA2 recruitment into these structures

To determine the role of the septins in remodeling F-actin and ANXA2 during plasma membrane repair, we next silenced SEPT7 expression. In SEPT7-deficient cells, the septin cytoskeleton (labeled with anti-SEPT2, 7, or 9 Abs) was markedly disrupted in accordance with the essential role of SEPT7 in the assembly of septin filaments (Figs. 7A, EV1G and EV5A,B). In LLO-injured cells, the formation of F-actin/myosin IIA knobs and loops was significantly decreased in SEPT7-deficient cells compared to cells treated with control siRNA (Figs. 7A,B and EV5C). Thus, the septin cytoskeleton controls the remodeling of cortical F-actin in knob and loop structures, in accordance with the known role of the septins in bending actomyosin filaments (Nakamura et al, 2023). Importantly, the rare F-actin loops still observed in SEPT7-siRNA-treated cells colocalized with low levels of septins, likely in cells that were less effectively silenced (Fig. 7A). In SEPT7-deficient cells, ANXA2 distribution was unaffected in the absence of LLO in comparison to control siRNA-treated cells (Figs. 7C and EV5B).

However, in the presence of LLO, ANXA2 remodeling was significantly decreased in SEPT7-deficient cells in comparison to SEPT7-proficient cells (Fig. 7A,C). These data show that ANXA2 remodeling in LLO-injured cells is septin-dependent. We also found that septin and F-actin remodeling induced by LLO was similar in ANXA2-deficient and control cells treated with non-targeting siRNA (Figs. 7B,D and EV5D). Collectively, our data support a novel model in which the septin cytoskeleton promotes plasma membrane repair by playing a key structural role in organizing membrane domains containing the actomyosin cytoskeleton and ANXA2.

## Discussion

This work reports the role of the septin cytoskeleton in plasma membrane repair of cells perforated by pore-forming toxins and mechanical wounding. Our data support a model in which the septin cytoskeleton acts as a scaffold to promote the formation of membrane repair domains containing the contractile F-actin cytoskeleton and annexin A2.

Septin family members and ANXA2 silencing, the use of the septin-targeting drug forchlorfenuron (FCF), and fluorescence microscopy analyses of septin and actin filaments organization, provided a comprehensive array of experimental approaches to study the role of the septin cytoskeleton in plasma membrane repair. Septins are conserved GTP-binding proteins divided into four groups based on sequence homologies (SEPT 2, 3, 6, and 7). Each septin group includes several interchangeable members except for SEPT7, the only member of its group (Martins et al, 2023). Silencing SEPT7 and SEPT6 had a significant deleterious effect on plasma membrane repair, whereas silencing of SEPT2 or SEPT9 had no effect (Figs. 1 and EV1). This result can be explained by the fact that SEPT7 silencing disrupts the whole septin cytoskeleton due to: (i) the essential, non-redundant function of SEPT7 in septin hetero-oligomer and filament formation, and (ii) the degradation of numerous septin members in SEPT7-deficient cells (Figs. 7, EV1C–F, and 5A) (Kremer et al, 2005; Spiliotis and Nakos, 2021). However, the septin-6 group consists of septin-6, 8, 10, 11 and 14, therefore, the defect in plasma membrane repair of SEPT6-deficient cells suggests that septin 6 displays unique properties, which cannot be rescued by other members of its subgroup during plasma membrane repair. On the contrary, SEPT2 and 9 appears to be interchangeable with other septins from their respective subgroups during plasma membrane repair.

The redistribution of the septin cytoskeleton during plasma membrane repair requires extracellular calcium (Figs. 2, 6A–C, and

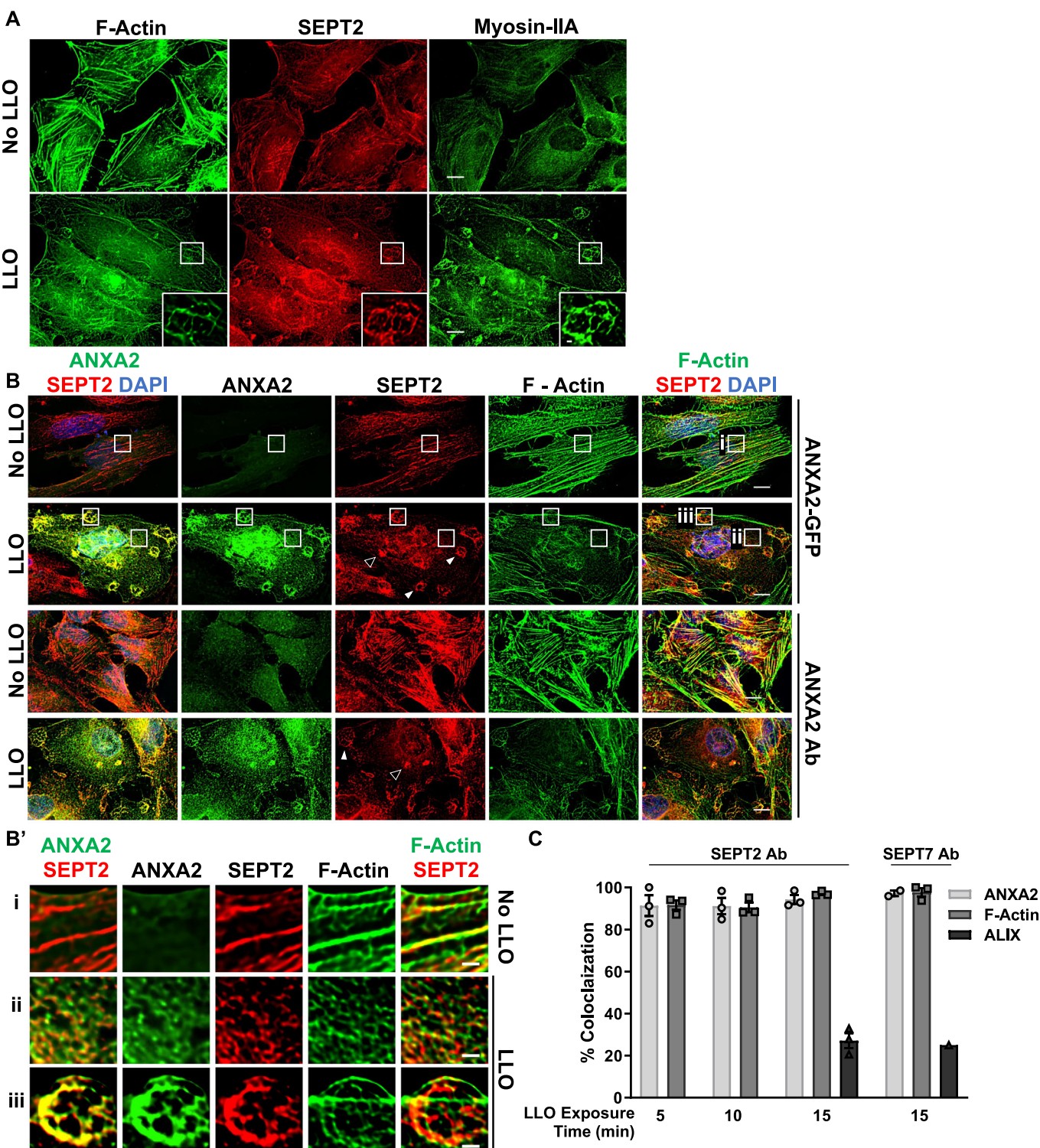

**Figure 3. The septin cytoskeleton colocalizes with F-actin, myosin-IIA, and ANXA2 in LLO-perforated cells.**

HeLa cells were incubated without (No LLO) or with 0.5 nM LLO for 5–15 min in repair permissive buffer M1. Cells were fixed, permeabilized and fluorescently labeled for SEPT2 (Alexa Fluor 568-conjugated secondary), myosin-IIA (Alexa Fluor 488-conjugated secondary), F-actin (Alexa Fluor 647-conjugated phalloidin), ANXA2 (Alexa Fluor 488-conjugated secondary), and nuclei (DAPI) for microscopy analyses. (A) Representative images of cells labeled for SEPT2, F-Actin, and myosin-IIA at time point 15 min. Scale bars are 10 μm or 1 μm in zoomed regions. (B, B') HeLa cells were transiently transfected, or not, to express ANXA2-GFP. Representative images of cells labeled for SEPT2, F-Actin, and ANXA2 at time point 15 min. Scale bars are 10 μm in (B) and 2 μm in (B'). (B') Enlarged regions from (B) to better visualize the septin colocalization with F-actin and ANXA2-GFP in both loop and non-loop regions. Fluorescence display was enhanced only in (B'ii). (A, B, B') All images were acquired by z-stacks widefield microscopy, denoised, deconvolved, and presented as the best focus images. (C) The percentages of colocalization of the septins with ANXA2, F-actin, and ALIX were measured from a total of 300–750 cells in each experimental condition, N = 3, Average ± SEM for all SEPT2 Ab conditions. Source data are available online for this figure.

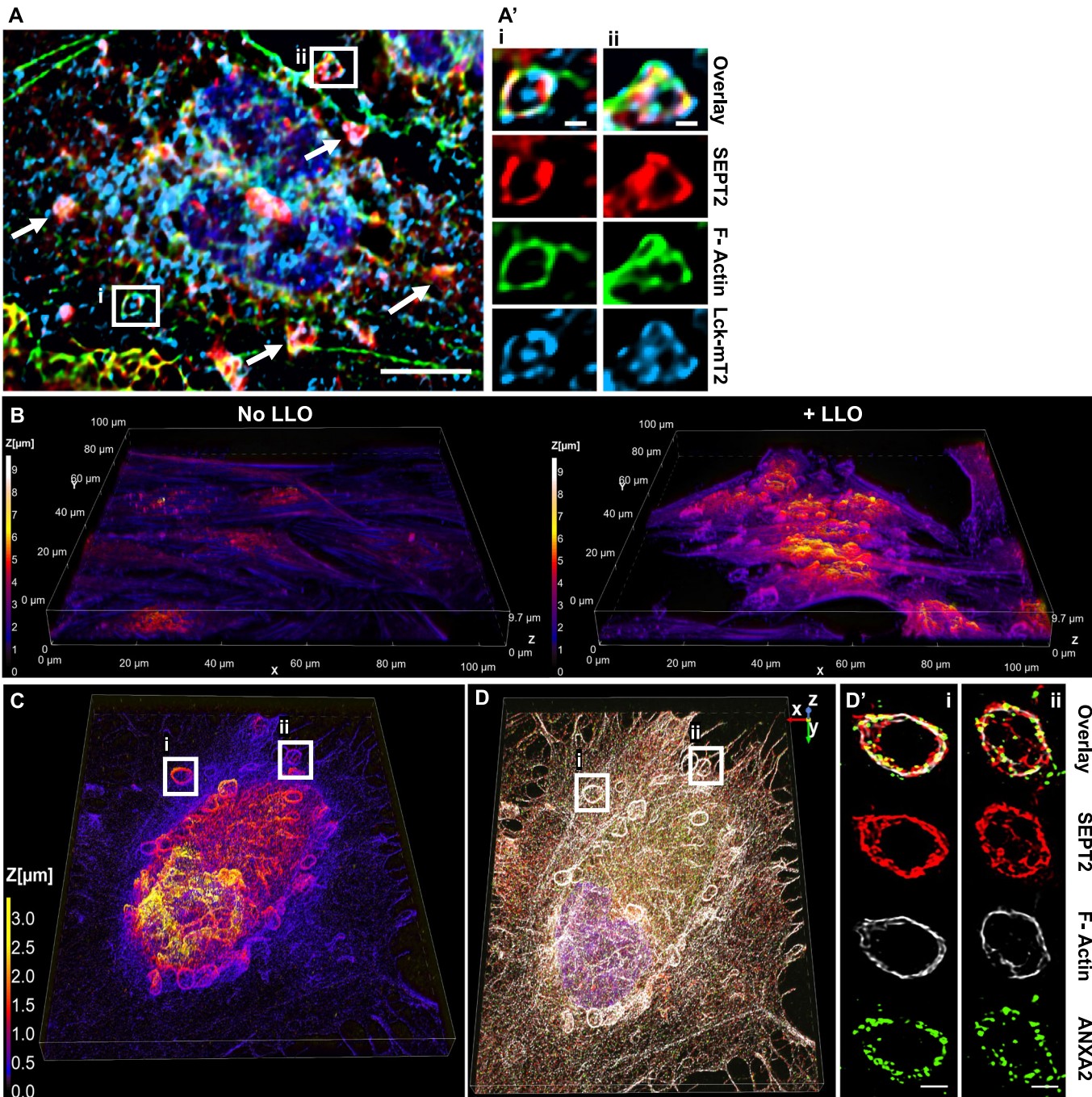

**Figure 4. The septin cytoskeleton redistributes with F-actin and ANXA2 in close association with the plasma membrane.**

(A, A') HeLa cells were transiently transfected to express Lck-mTurquoise2 (Lck-mT2) and incubated with 0.5 nM LLO for 15 min in M1. Cells were fixed, permeabilized, and fluorescently labeled for SEPT2 (Alexa Fluor 568) and F-actin (Alexa Fluor 647). Widefield Z-stack was deconvolved and a single plane is presented. Scale bar is 10 μm. Arrows point to additional SEPT2, F-actin, and Lck colocalization. (Ai) and (Aii) are regions enlarged from (A), scale bars are 1 μm. (B) HeLa cells were exposed to 0.5 nM LLO, or not, for 15 min in M1 and were fixed, permeabilized, and fluorescently labeled for SEPT2 (Alexa Fluor 568-conjugated secondary), ANXA2 (Alexa Fluor 488-conjugated secondary), and F-actin (Alexa Fluor 647-conjugated phalloidin). Z-stack images were acquired by resonant scanning confocal microscopy with 0.1 μm steps. Images show the 3-D projection of the deconvolved overlay of SEPT2, F-actin, and ANXA2 displayed with a z-depth coding. (C, D, D') FCF-treated cells were exposed to 0.5 nM LLO for 15 min in M1 and were fixed, permeabilized, and fluorescently labeled for SEPT2 (Alexa Fluor 568-conjugated secondary), ANXA2 (Alexa Fluor 488-conjugated secondary), and F-actin (Alexa Fluor 647-conjugated phalloidin). Images were acquired by spinning disk confocal microscopy. (C) A 3-D projection of all deconvolved images (44.99 μm × 56.07 μm × 3.4 μm) and overlayed with a z depth code. (D) A 3-D overlay of SEPT2, F-actin, and ANXA2. (D') The overlay and individual channels of two selected loop structures with an optical depth of 1.4 μm (scale bar is 1 μm).

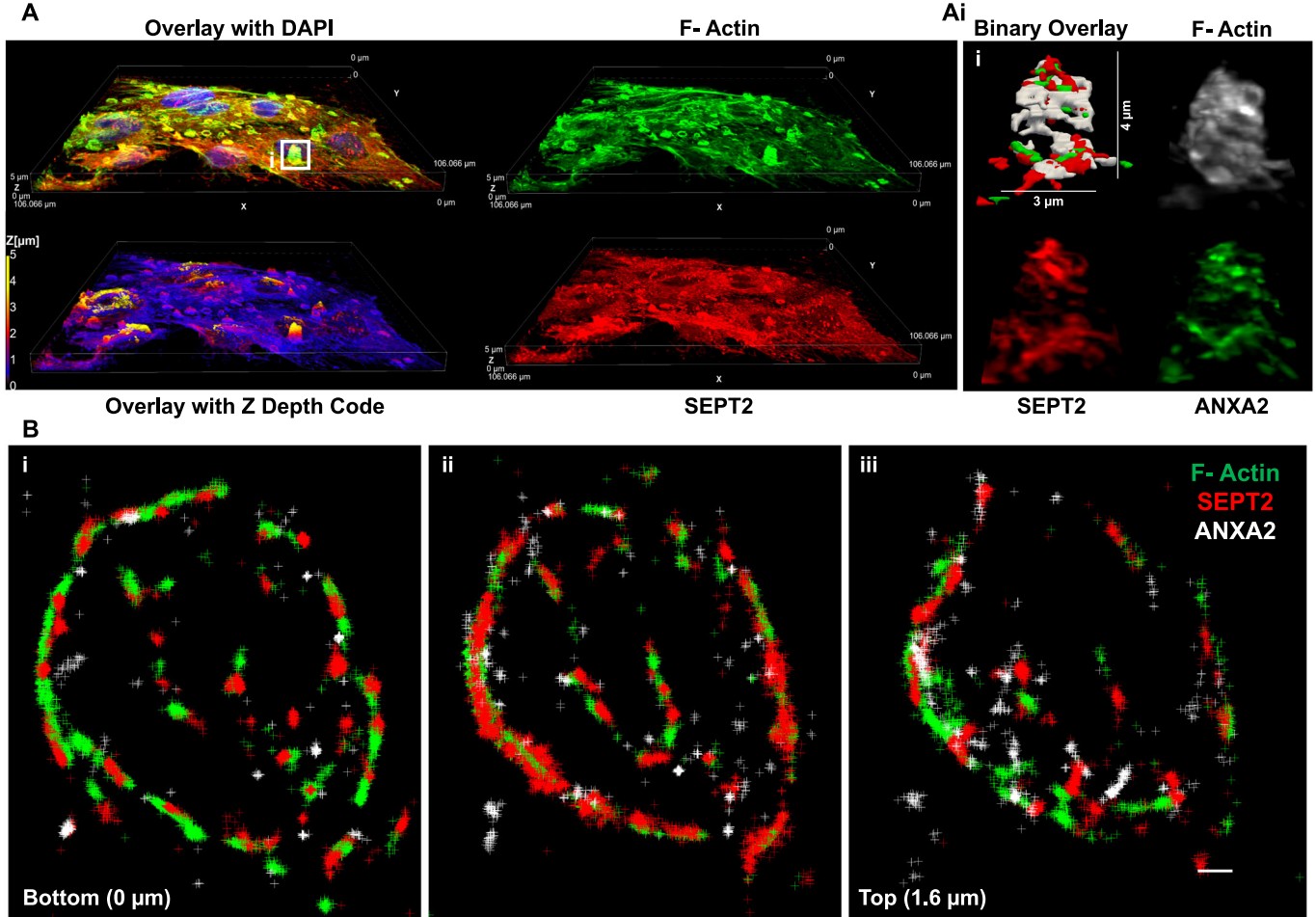

**Figure 5. Septins form circular filaments intertwined with F-actin and connected to ANX A2.**

(A) HeLa cells, pre-treated with 100 μM FCF, were incubated with 0.5 nM LLO for 15 min in M1. Cells were fixed, permeabilized, and fluorescently labeled for SEPT2 (Alexa Fluor 568-conjugated secondary), F-actin (Alexa Fluor 647-conjugated phalloidin), and nuclei (DAPI). 3-D representation of individual channels and 3-D overlay with depth coding of a denoised and deconvolved Z-stack acquired by resonant scanning confocal microscopy (106 μm × 106 μm × 5 μm), 0.2 μm step size. (Ai) Enlarged 3-D representations of (A) and its binary overlay. (B) HeLa cells were incubated with 0.5 nM LLO for 15 min in M1. Cells were fixed, permeabilized, and fluorescently labeled for SEPT2 (Alexa Fluor 568-conjugated secondary), ANXA2 (Alexa Fluor 647-conjugated secondary) and F-actin (ATTO 488-conjugated phalloidin). Images were acquired by super-resolution STORM microscopy (step size 0.2 μm). Each panel (Bi–iii) displays molecules detected within a 0.8 μm optical depth. Scale bar is 1 μm.

EV2), which is a hallmark of plasma membrane repair mechanisms (Jimenez and Perez, 2017; McNeil, 2002). To date, there is no calcium-dependent mechanism known to control the organization of the septin cytoskeleton and the septins are not known to directly bind calcium ions. Therefore, how calcium controls septin remodeling during cell repair remains to be elucidated. In injured cells, the septins formed structures that we described as knobs and loops, proximal to the plasma membrane and that can protrude from the cell surface (Figs. 4 and 5). It is not known if knobs and loops form independently, but based on the timing of their respective appearance, the knobs might be an early stage of the loop structure formation (Fig. 2C). The fact that septins redistribute within subdomains of the plasma membrane favors their role in plasma membrane repair (Figs. 4 and 5). This is further supported by the observation that in mechanically wounded cells, septins redistribute at the wound site (Fig. EV2C). There was a significant increase in septin knob and loop formation in FCF-treated cells paralleled with a significant increase in plasma membrane repair

efficiency, further confirming the role of septin knobs and loops in plasma membrane repair (Figs. 6B–D and 1G). FCF is a drug that specifically associates with the septin GTP-binding pocket, mimicking a nucleotide-bound state (Angelis et al, 2014). FCF was proposed to disrupt septins' functions (Iwase et al, 2004) but was also shown to facilitate their assemblies (DeMay et al, 2010; Hu et al, 2008). Therefore, mechanisms by which FCF alters the septin dynamics are still not completely understood and need further elucidation. Our data clearly show that FCF treatment enhances the formation and/or stabilizes septin loops during plasma membrane repair.

During plasma membrane repair, the remodeled septins closely associate with F-actin, myosin IIA, and ANXA2 throughout the cell surface and within the septin knobs and loops (Figs. 2, 3, 4, 5 and EV2, 3 and 5), which suggests that these molecules act together to repair the membrane. Super-resolution microscopy allowed for visualization of intertwined F-actin and septin filaments in the septin loops (Fig. 5B), in accordance with the role of septins

in bending actin filaments (Benoit et al, 2023). The septins mostly run parallel to the F-actin and, at times, appear perpendicular as if interconnecting actin fibers (Fig. 5B). In accordance with our observation, recent literature showed that the septins form filaments that preferentially associate with contractile F-actin/myosin II cytoskeleton (Martins et al, 2023). The same study also showed that septin filaments can anchor the actin cytoskeleton to membranes (Martins et al, 2023). However, it will remain to elucidate if the septins are directly anchored to both the plasma membrane and F-actin during plasma membrane repair. Super-resolution microscopy revealed that ANXA2 is organized in patches closely connected to both F-actin and septin filaments (Fig. 5B). There is no known direct interaction between the septins and ANXA2 and it is unclear if septins and ANXA2 are directly or indirectly associated. Similar to the septins, ANXA2 can interact directly with F-actin and phospholipids (Benaud et al, 2004). Whether the septins and/or ANXA2 play a role in anchoring the cortical actomyosin network to the plasma membrane during plasma membrane repair is to be determined. Silencing SEPT7 led to a strong and significant decrease in the formation of cortical F-actin/myosin IIA knobs and loops, whereas silencing ANXA2 had no noticeable effect (Fig. 7). These results demonstrate that the septins play a critical structural role for the remodeling of the actomyosin cytoskeleton and the recruitment of ANXA2 during plasma membrane repair.

ANXA2 is a $Ca^{2+}$-sensor critical for the repair of CDC (perfringolysin O)-injured and of mechanically wounded cells (Koerdt et al, 2019; Ray et al, 2022) and was identified in our screen (Table EV1). ANXA2 was proposed to facilitate shedding of microvesicles in perfringolysin O-injured cells (Ray et al, 2022) and to promote F-actin-driven closure or excision of mechanical wounds (Bement et al, 1999; Jaiswal et al, 2014). In the latter model, ANXA2, the protein S100A11, and cortical F-actin act together to repair mechanically ruptured cells by excision (Jaiswal et al, 2014). We also observed the colocalization of ANXA2/S100A11 and F-actin (Fig. EV3C) which supports that such repair mechanism is also involved in cells wounded by pore-forming toxins. Collectively, (i) the identification of ANXA2 in our screen, (ii) the known role of ANXA2 in plasma membrane repair, (iii) the colocalization of ANXA2 with S100A11 and the reorganized F-actin/myosin IIA cytoskeleton, and (iv) the structural role of the septin in the formation of the submembranous ANXA2/S100A11/F-actin/myosin IIA domains strongly support the key role of the septins in organizing plasma membrane repair domains. Based on the known role of ANXA2 and contractile F-actin in plasma membrane repair, the septin-organized domains likely act as repair platform by sequestering and/or removing the damaged plasma membrane.

The septins are known to organize and bend the actomyosin cytoskeleton during cytokinesis (Mavrakis et al, 2014; Nakamura et al, 2023; Russo and Krauss, 2021) and ANXA2 is required for cytokinesis (Benaud al, 2015). Therefore, septin-dependent reorganization of the cortical F-actin network and ANXA2 likely share common mechanisms during cytokinesis and plasma membrane repair. Septin filaments are scaffolding proteins that interact with multiple cellular components including inositol phosphates, SNARE machinery, clathrin, actin, motor proteins, and tubulin. Through these interactions, the septins control multiple processes that encompass signaling, membrane fusion, membrane curvature

control, endocytosis, and exocytosis. (Benoit et al, 2023; Robertin and Mostowy, 2020; Spiliotis and Nakos, 2021). Therefore, the septins may facilitate several membrane repair mechanisms dependently or independently of the actomyosin cytoskeleton and ANXA2. Our screen indicates that membrane fusion, lysosome exocytosis, annexins- and ESCRT-III play a role in the repair of LLO-injured cells. In particular, silencing CHMP2 and ANXA7 (Table EV1) negatively affected plasma membrane integrity, supporting that the ESCRT-III machinery repairs LLO-injured cells by shedding microvesicles (Jimenez et al, 2014; Sonder et al, 2019). These findings suggest that plasma membrane repair likely involves complex multipartite mechanisms that together maintain cell integrity. Whether these repair mechanisms are mechanistically related and if they operate in similar or distinct time scales will need to be established.

*L. monocytogenes* is a facultative intracellular pathogen, which intracellular replication is essential for pathogenesis. Paradoxically, this bacterium uses as its major virulence factor the pore-forming toxin LLO that has the potential to perforate and kill cells, yet infected cells remain viable to support bacterial replication (Czuczman et al, 2014; Goldfine et al, 1995; Goldfine et al, 2000; Vadia et al, 2011). Viability of infected cells results from the combination of (i) LLO intrinsic properties that decrease its stability/lifetime, (ii) LLO internalization, and (iii) plasma membrane repair (Cassidy et al, 2012; Chen et al, 2018; Rogers et al, 1986; Schuerch et al, 2005; Seveau, 2014; Vadia and Seveau, 2014). The septins were previously shown to regulate *L. monocytogenes* internalization and to form a cage-like structure around cytosolic bacteria (Robertin and Mostowy, 2020). Our data highlight a novel function of the septins in the repair of LLO-perforated host cells and show that LLO redistributes the septin cytoskeleton. It will be important to establish if the roles of the septins in bacterial uptake and in their cytosolic entrapment are regulated by LLO-induced host cell perforation.

In conclusion, our unexpected findings that septins control plasma membrane repair opens entirely new avenues for understanding the underlying cell biology of this process. Moreover, the novel septin-based pathway could provide new insights into pathological conditions involving excessive loss of plasma membrane integrity, including muscular dystrophy, ischemia-reperfusion, heart failure, chronic inflammation, neurodegeneration, and infectious diseases.

# Methods

## Cell culture and reagents

Our studies used HeLa cells which have been extensively used to identify plasma membrane repair mechanisms. HeLa cells (ATCC, # CCL-2™, authenticated by STR profiling, STRA7778) and HeLa Histone 2B-GFP (HeLa H2B-GFP, EMD Millipore SCC117, authenticated by STR profiling, STRA7779) were cultured in Dulbecco's modified Eagles medium (DMEM, Gibco #11995) containing 10% heat-inactivated fetal bovine serum (HIFBS), 100 units/ml penicillin and 100 μg/ml streptomycin at 37 °C and 5% $CO_2$ atmosphere. Repair permissive buffer (M1) (Hanks Balanced Salt Solution (HBSS, Gibco #14175) supplemented with 0.5 mM $MgCl_2$, 1.26 mM $CaCl_2$, 25 mM glucose, and 10 mM HEPES) and repair restrictive buffer (M2) (HBSS supplemented with 0.5 mM

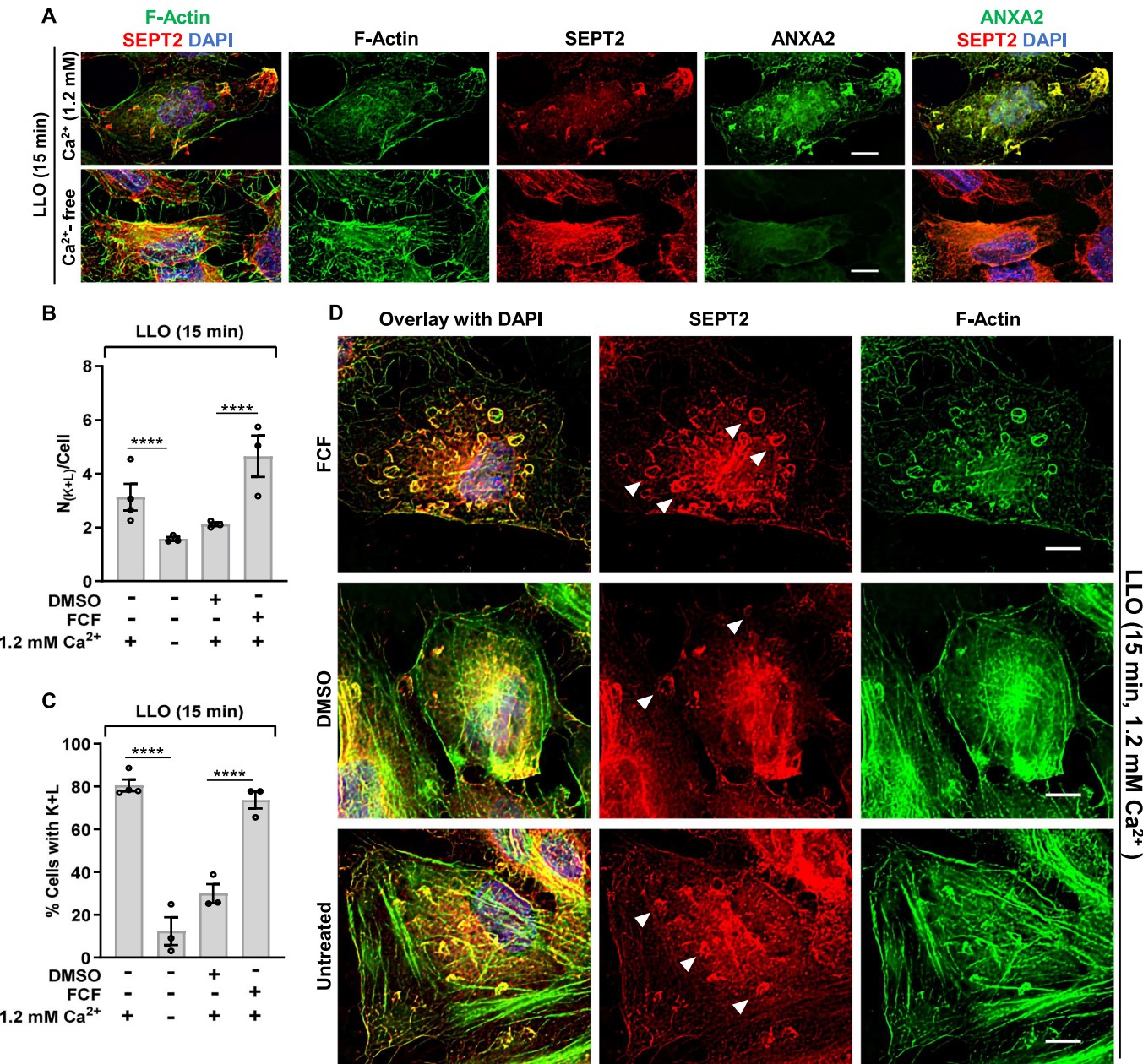

**Figure 6.  Redistribution of the septin cytoskeleton is functionally correlated with the plasma membrane repair efficiency.**

(A) HeLa cells transfected with ANXA2-GFP were incubated with 0.5 nM LLO for 15 min in M1 or M2 buffers. Cells were fixed, permeabilized, and fluorescently labeled for SEPT2 (Alexa Fluor 568-conjugated secondary), F-actin (Alexa Fluor 647-conjugated phalloidin), and nuclei (DAPI). Scale bars are 10 μm. (B, C) Cells were treated as indicated in (A) and (D). The average number of septin knobs and loops per cell presenting these structures ($N_{(K + L)}$/Cell) Average ± SEM (B), and the percentage of cells presenting these structures (% Cells with K + L) Average ± SEM (C) were enumerated based on septin widefield fluorescence images. A total of 300–750 cells were analyzed in each experimental condition from duplicate wells, $N = 3$ independent experiments at minimum. (D) Control untreated, FCF (100 μM) pre-incubated, and vehicle DMSO pre-incubated HeLa cells were incubated with 0.5 nM LLO for 15 min in M1. Equivalent concentrations of FCF and DMSO were maintained during LLO exposure. Cells were fixed, permeabilized, and fluorescently labeled for SEPT2 (Alexa Fluor 568-conjugated secondary), F-actin (Alexa Fluor 647-conjugated phalloidin), and nuclei (DAPI). Z stack images were acquired by widefield microscopy and were denoised, deconvolved, and presented as best-focus projection images. Representative loops structures are indicated by filled arrowheads. Scale bars are 10 μm. Data Information: In (B and C), linear mixed-effects models were used for analysis, ****$P < 0.0001$. Source data are available online for this figure.

MgCl$_2$, 25 mM glucose and 10 mM HEPES) buffers were used for the repair assays. For lentiviral production, Lenti-X™ 293T Cell Line (Takara) was cultured in 90% DMEM with high glucose (4.5 g/L), 4 mM L-glutamine, and 10% FBS; 100 units/ml penicillin G

sodium, and 100 μg/ml streptomycin sulfate. When appropriate, the cell culture medium contained the Tet-system-approved FBS (Takara). Recombinant six-histidine-tagged LLO and PLY was purified from *E. coli* as previously described (Vadia et al, 2011). The

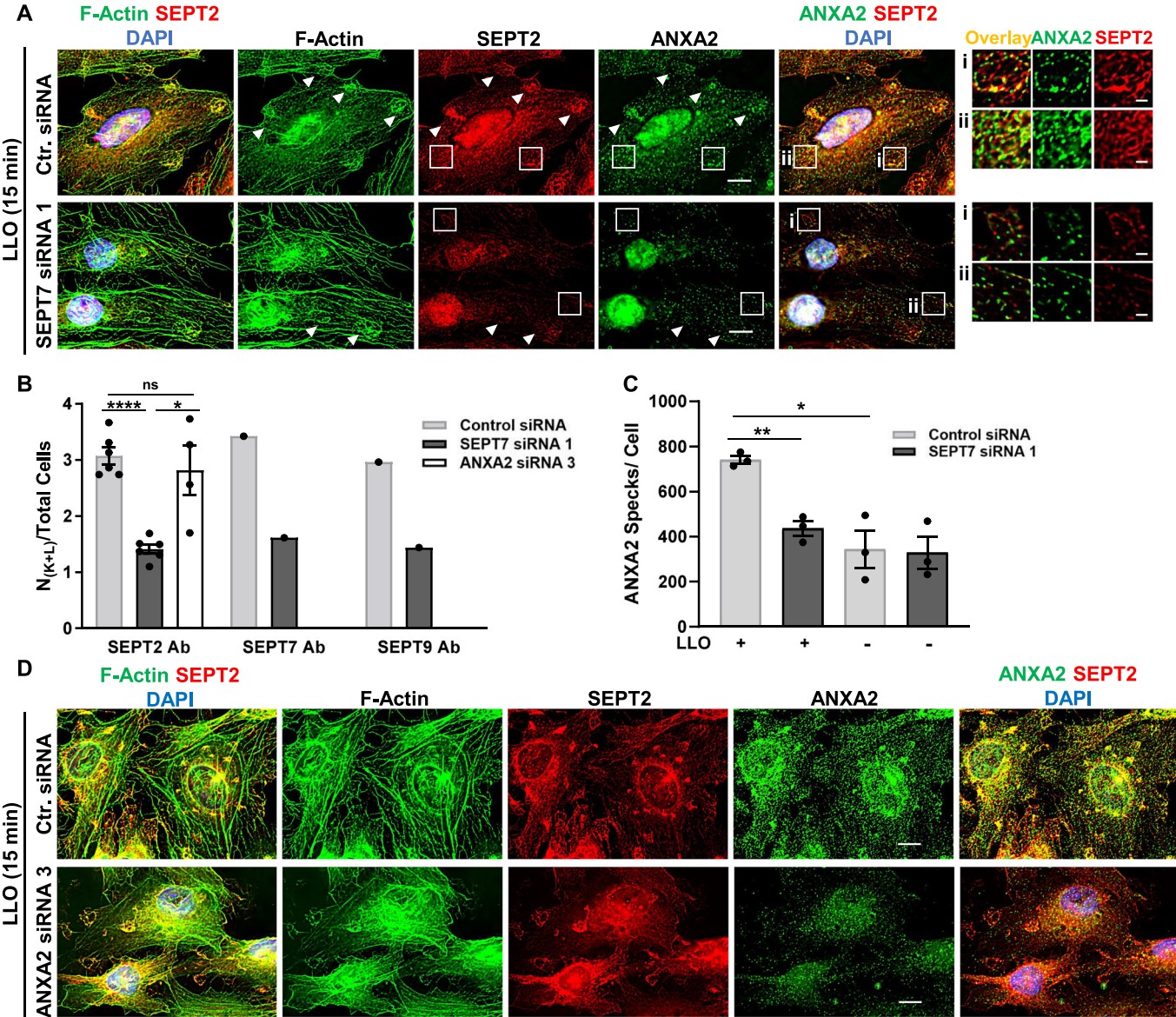

**Figure 7. Redistribution of F-actin and ANXA2 is dependent on SEPT7 expression.**

(A) HeLa cells transfected with Ctr. siRNA or SEPT7-siRNA 1 were incubated with 0.5 nM LLO for 15 min in M1. Cells were fixed, permeabilized, and labeled for SEPT2 (Alexa Fluor 568-conjugated secondary), F-Actin (Alexa Fluor 647-conjugated phalloidin), ANXA2 (Alexa Fluor 488-conjugated secondary), and nuclei (DAPI). Presented images are the best-focus projections generated from deconvolved widefield Z-stacks and are displayed with the same intensity scaling for each fluorescence setting. Representative loop structures are indicated by filled arrowheads. Scale bars are 10 μm. (Ai) and (Aii) are regions enlarged from (A), scale bars are 2 μm. (B) Same experiment as in (A) but cells were transfected with Ctr. siRNA, SEPT7-siRNA 1, or ANXA2-siRNA 3 and SEPT7 and SEPT9 were also labeled with fluorescent Abs. The number of knobs and loops per total cells ($N_{K+L}$/Total cells) were enumerated based on F-actin labeling in cells fluorescently labeled with anti-SEPT2, -SEPT7, and -SEPT9 antibodies. A total of 150–300 cells were analyzed from duplicate wells, per experimental condition. For the SEPT2-labeled cells, $N = 6$ independent experiments for the Ctr.- and SEPT7-siRNA treated cells, $N = 4$ in the ANXA2-siRNA treated cells. $N = 1$ for SEPT7 and SEPT9 Ab labeling. (C) HeLa cells were transfected with Ctr. siRNA or with SEPT7-siRNA 1 and incubated with 0.5 nM LLO ($+$), or not ($-$), for 15 min in M1. Cells were fixed, permeabilized, and labeled for SEPT2 (Alexa Fluor 568-conjugated secondary), F-Actin (Alexa Fluor 647-conjugated phalloidin), ANXA2 (Alexa Fluor 488-conjugated secondary), and nuclei (DAPI). Fluorescence images (z-stacks) were acquired by widefield microscopy. The ANXA2 images were deconvolved and the number of ANXA2 specks were automatically counted in 3D by the software. At least 100 cells were analyzed for each experimental condition, $N = 3$ independent experiments. (D) HeLa cells were transfected with Ctr. siRNA or with ANXA2-siRNA 3 and incubated with 0.5 nM LLO, or not (Ctr.), for 15 min in M1. Cells were labeled indicated in (A). Data Information: In (A and D), the presented images are the best-focus projections from deconvolved z-stacks acquired by widefield microscopy and are displayed with the same intensity scaling for each fluorescence setting. Scale bars are 10 μm. In (B) and (C), data are expressed as average ± SEM and a one-tailed Students Paired T Test was used for analysis, *$P < 0.05$, **$P < 0.01$, and ****$P < 0.0001$. Source data are available online for this figure.

hemolytic activity of LLO and PLY were measured as previously described (Vadia et al, 2011). In all of our experimental conditions, cells are viable and fully recover from LLO injury (Lam et al, 2018; Vadia et al, 2011; Vadia and Seveau, 2014). LLO was used at 0.5 nM in all experiments unless otherwise indicated. The stock solution of Forchlorfenuron (FCF) (Sigma, #32974) was prepared in DMSO at 50 mM. FCF is used at 100 μM and DMSO is used at a 1:500 dilution for a vehicle control. Anti-SEPT2 (Protein Tech, mouse clone 2F3B2, for Western Blot, #RQ-1000347808 Atlas, Rabbit polyclonal for imaging, #HPA018481), anti-SEPT6 (Protein Tech rabbit polyclonal for Western Blot, #12805-1-AP), anti-SEPT7 (Protein Tech rabbit polyclonal for Western Blot and imaging, #13818-1-AP), anti-SEPT9 (Atlas, rabbit polyclonal for Western Blot and imaging, #HPA042564), anti-annexin A2 (Invitrogen, mouse clone Z014 polyclonal for Western Blot and imaging, #03-4400), anti-Alix (Biolegend, clone 3A9, #634501), anti-myosin-IIA (Abnova, clone 2B3, #H00004627-M03), anti-tubulin (Sigma, clone B-5-1-2), anti-S100A11 mouse (Protein Tech, clone 1A3F1, #60024-1-Ig), anti-S100A11 rabbit (Protein Tech, #10237-1-AP), Alexa Fluor 488 conjugated Phalloidin (Invitrogen, #A12379), Alexa Fluor 647 conjugated Phalloidin (Invitrogen, #A22287), and ATTO 488 conjugated Phalloidin (ATTO-TEC, #AD 488-81) were the primary antibodies used for immunoblotting and fluorescence microscopy experiments. The secondary antibodies used were anti-mouse IgG horseradish peroxidase (HRP)-conjugated (Cell Signal Technology, #7076S), anti-rabbit IgG (HRP)-conjugated (Cell Signal Technology, #7074S), anti-mouse IgG Alexa Fluor-conjugated (Invitrogen), anti-rabbit IgG Alexa Fluor-conjugated (Invitrogen). 10 kDa lysine fixable Ruby dextran (ThermoFisher, #D1817) and 10 kDa lysine fixable Emerald dextran (Thermo-Fisher, #D1820) were used for the mechanical wounding assay. The vector Lck-mTurquoise2, in the backbone pEGFP-C1, was a gift from Dorus Gadella (Addgene #98822) (https://doi.org/10.1101/160374). The vector encoding annexin A2-GFP, in the backbone pEGFP-N3, was a gift from Volker Gerke & Ursula Rescher (Addgene #107196) (Rescher et al, 2000). EZ-Tet-pLKO-Puro (Addgene #85966) was a gift from Cindy Miranti and both psPaX2 (Addgene #12260), and pMD2.G (Addgene #12259) were gifts from Didier Trono. Cells were transiently transfected using Lipofecta-mine 2000 (Invitrogen) and Opti MEM according to the manufacturer's instruction.

## siRNA screen

A siRNA screen was performed in 96-well tissue culture-treated clear bottom black plate (Corning). HeLa H2B-GFP cells were plated at $7 \times 10^3$ cells/well (DMEM, 10% HIFBS) and reverse transfected with 1 pmol/well of a combination of 3 non-overlapping 11-mer siRNAs (Ambion Silencer® Select, Thermo-fisher) using Lipofectamine RNAiMax transfection reagent (Invitrogen) and Opti-MEM Reduced Serum Medium (Gibco) according to the manufacturer's instructions. For the list of targeted genes and siRNA sequences, see Dataset EV1. To account for potential variations in cell growth during the 72-h treatment with the various siRNAs, control cells were plated at three different concentrations ($7 \times 10^3$, $6.3 \times 10^3$, and $5.6 \times 10^3$ cells/well) in each plate and reverse transfected with 1 pmol of non-targeting Silencer Select negative control siRNA No. 1 (Invitrogen). All experimental conditions were carried out in quadruplicates. To assess membrane resealing,

siRNA-treated cells were washed twice with 37 °C M1 and replaced with M1 containing 1 μM TO-PRO-3 iodide (ThermoFisher) or washed once with 37 °C M2 containing 5 mM ethylene glycol-bis(2-aminoethylether)-N,N,N',N',tetraacetic acid (EGTA), followed by a wash with M2, and replaced with M2 containing 1 μM TO-PRO-3 iodide (ThermoFisher). A SpectraMax i3X multimode detection platform equipped with a MiniMax cytometer (Molecular Devices) was used to obtain fluorescent kinetic readings and 4× magnification images (transmitted light, and green and red fluorescence; 535 nm and 638 nm excitations and 617 nm and 665 nm emissions, respectively). Pre-kinetic images were acquired as previously described (Lam et al, 2019). The cell culture plate was cooled to 4 °C for 5 min and ice-cold LLO and TO-PRO-3 iodide were added at 0.5 nM and 1 μM final concentrations, respectively. The plates were then placed into the spectrofluorometer for the 30 min kinetic assay at 37 °C, then post-kinetic fluorescence images were acquired as described previously (Lam et al, 2019). The total number of cells in each well was enumerated based on their nuclear fluorescence (H2B-GFP), pre- and post-kinetic. We found that the silencing of 217 proteins resulted in cell numbers ranging from 85–115% relative to control cells. Gene knockdown (KD) resulting in cell densities outside of this range were excluded (10 gene KD with cell counts >115% and 18 with cell counts <85% of control cells were excluded) (Dataset EV2). Importantly, cells did not detach during the assay as the average cell count ratio, defined as the cell count post-kinetic relative to the cell count pre-kinetic, was unaffected in all samples, and across all siRNA- and LLO-treatment conditions was 1.0061 ± 0.0065 (average ± SEM). Control wells with the closest cell count to the specific knockdown condition were chosen for comparison. TO-PRO-3 fluorescence intensities were compared between cells treated with specific-siRNA and control-siRNA treated cells. The average TO-PRO-3 fluorescence intensities at time point 30 min were log-transformed and then analyzed with ANOVA to establish the list of candidate genes with positive or negative effects, respectively, in plasma membrane repair.

## SDS-PAGE and immunoblotting

HeLa cells were lysed in 150 mM NaCl, 20 mM Tris HCl, 2 mM EDTA, NP40 1% and protease inhibitors (Roche # 04693132001) pH 7.4. Samples of cells lysates were used for BCA (BioRad) protein assay. Lysates were boiled in reducing Laemmli buffer and subjected to SDS-PAGE electrophoresis followed by transfer to PVDF membranes. Membranes were labeled with the indicated antibodies (septin or annexin) and then stripped (2.35% SDS, 73.5 mM Tris HCl pH 6.8, 134.6 mM beta-mercaptoethanol) and re-labeled with tubulin as a loading control. The knockdown efficiencies were measured using Image Lab (BioRad) software. Quantification included septin (or annexin) normalization to their respective loading controls.

## Generation of doxycycline-inducible SEPT7 shRNA HeLa cell line

Annealed oligonucleotides (non-overlapping with the 3 siRNAs used in the screen) were cloned into the vector EZ-Tet-pLKO-Puro using the restriction sites NheI and EcoRI (Frank et al, 2017). The resulting construct was validated by sequencing. Oligonucleo-tide and primer sequences are presented in the table below. For lentiviral production, Lenti-X™ 293T cells were transfected with

the vectors pMD2.G, psPAX2, and EZ-Tet-pLKO-Puro using Lipofectamine 2000 and Opti MEM according to the manufacturer's instructions (Invitrogen). Lentiviruses were collected, HeLa cells were transduced and selected using 1 µg/ml puromycin. Isolated colonies were pooled and further cultured. Stable HeLa-D$_i$SEPT7-shRNA1 was cultured with 50 ng/ml doxycycline (DOX) for 72 h and SEPT7 expression was measured by immunoblotting (Fig. 1E).

The silencing sequences are highlighted in gray. The loop sequence is in bold.

| | Oligonucleotides (shRNA) |
|---|---|
| SEPTIN7 shRNA1[a] | 5'-CTAGCGGGAAGCTCAACAACGTATTT**CTCGAG**AAATACGTTGTTGAGCTTCCCTTTTT-3' Sense |
| | 5'-AATTAAAAAGGGAAGCTCAACAACGTATTT**CTCGAG**AAATACGTTGTTGAGCTTCCCG-3' antisense |
| | **Sequencing Primers** |
| EZ-pLKO | 5'-GATTAGTGAACGGATCTCGACGG-3' forward |
| | 5'-AACCCAGGGCTGCCTTGG-3' reverse |

[a] SEPTIN 7 shRNA1 sequence obtained from the Genetic Perturbation Platform (Broad Institute).

## Plasma membrane repair assays

All plasma membrane repair assays (cells treated with siRNAs, shRNAs, or pharmacological agents) were based on the quantification of TO-PRO-3 fluorescence intensity. Cells were plated in 96- or 24-well plates and treated with siRNA and/or drugs (doxycycline, DMSO, or FCF) for the indicated times prior to the assay. Cells were washed twice with 37 °C M1 and incubated in M1 containing 1.5 µM TO-PRO-3. For calcium-free conditions, cells were washed once with M2 (5 mM EGTA) followed by a wash with M2 at 37 °C and incubated with M2 containing 1.5 µM TO-PRO-3. Cells were then incubated with or without 0.5 nM LLO or 2 nM PLY. In some experiments, FCF (100 µM) or DMSO vehicle (1/500 dilution) was added during LLO exposure for up to 30 min. TO-PRO-3 fluorescence intensity was measured in the temperature-controlled plate reader (Spectra Max i3X Multi-Mode Detection Platform, Molecular Devices) or by quantitative fluorescence microscopy as indicated in the Figure legends.

## Mechanical wounding assay (adapted from (Defour et al, 2014))

Cells were cultured on 35 mm round coverglasses and transfected with siRNAs (control, SEPT7 siRNA 1, ANXA2 siRNA 3). After 72 h, cells were washed twice with M1 at 37 °C, or washed once with M2 (5 mM EGTA) followed by a wash with M2 at 37 °C. The coverglasses were placed on a silicone O-ring and damaged by gently rolling glass beads in the presence of 1.5 mg/ml Emerald Dextran to label the damaged cells, followed by 5 min incubation at 37 °C. The Emerald Dextran and glass beads were then washed with M1 or M2 buffer. Ruby dextran (2 mg/ml) was added to the cells for 2 min at 37 °C to label the "not recovered" cells. Cells were washed with M1 or M2 and fixed. Coverglasses were mounted on glass slides and immediately imaged on the NikonTi2-E microscope with the 40× air objective. 50 images were acquired per slide. Emerald-fluorescent cells were enumerated as the total number of damaged cells and cells double-fluorescent for Ruby and Emerald were counted to represent the not recovered cells. Data are expressed as the percentage of not recovered cells relative to cells incubated in Ca$^{2+}$-free medium in each experimental condition.

## Fluorescence labeling

Cells were plated in a 24-well plate containing glass slides. Cells were exposed to 0.5 nM LLO or not for 5–15 min in M1 or M2 at 37 °C. In some experiments, FCF (100 µM) or DMSO vehicle (1:500) was added during LLO exposure. Cells were treated as indicated, washed twice in PHEM buffer (60 mM PIPES, 25 mM HEPES, 2 mM MgCl$_2$,10 mM EGTA; pH 6.9), fixed in 4% PFA in PBS for 20 min at room temperature (RT), and permeabilized for 5 min with 0.1% Triton X-100 in PBS. Blocking was performed at RT for 1 h in PBS 10% serum and cells were labeled with antibodies diluted in PBS, 10% serum for 1 h (except for septin Abs, 2 h). Due to its low expression level, SEPT6 labeling was very faint and was not carried out. Nuclei were labeled with DAPI. All primary antibodies were non fluorescent and secondary antibodies were Alexa Fluor-conjugated. F-actin was labeled with Alexa Fluor-conjugated phalloidin. Figure legends indicate the fluorochromes used in each experiment. Fluorescence images were acquired in gray scale and were color-coded. Widefield images were acquired using a 60× water objective and confocal images were acquired using a 60× oil objective.

## Super-resolution fluorescence labeling

Cells, plated on a glass bottom 35 mm dish, were exposed to 0.5 nM LLO for 15 min in M1 at 37 °C. Cells were washed twice in PHEM buffer, fixed in 3% PFA and 0.1% glutaraldehyde in PHEM for 10 min at RT and permeabilized for 15 min with 0.2% Triton X-100 in PBS. Blocking was performed by incubating cells at RT for 90 min in PBS with 10% serum and 0.05% Triton X-100 and labeling by incubating cells with antibodies diluted in PBS, 5% serum and 0.05% Triton X-100 for 1 h for all labeling. After labeling with primary Abs (anti-ANXA2, anti-SEPT2 Abs, and ATTO 488-conjugated Phalloidin) and secondary antibodies (Alexa Flour 568 and Alexa Fluor 647), cells were fixed again in 3% paraformaldehyde and 0.1% glutaraldehyde in PHEM for 10 min at RT. Protocol was modified from (Dempsey et al, 2011). Cells were imaged in a STORM imaging buffer that was prepared on ice right before imaging (Buffer A: 50 mM Tris pH 8.0 with 10 mM NaCl and 10% glucose mixed with Buffer B: 0.5 mg/mL glucose oxidase and 40 µg/mL catalase in 50 mM Tris pH 8 10 mM NaCl, 1% 2-mercaptoethanol and 20 mM Cysteamine).

## Microscopes, image acquisitions, and analyses

*Widefield microscopy (Nikon):* Images were acquired using a Nikon Ti2-E microscope equipped with a temperature, humidity, and $CO_2$-controlled blackout enclosure. Ten excitation wavelengths (Spectra III pad 365, 440, 488, 514, 561, 594, 640, and 730 nm) and emission filter sets for DAPI ($435 \leq$ emission $\leq 485$ nm), CFP ($460 \leq$ emission $\leq 500$ nm), FITC ($515 \leq$ emission $\leq 555$ nm), YFP HYQ (from $520 \leq$ emission $\leq 560$ nm), Cy3 (from $573 \leq$ emission $\leq 648$ nm) and Texas Red HYQ ($573 \leq$ emission $\leq 648$ nm) imaging with a high-speed wheel; back-thin illuminated SCMOS camera (Orca-Fusion BT, Hamamatsu) of resolution of 5.3 Megapixels; nanopositioning Piezo Sample Scanner (Prior). The automated stage allows multi-positioning (x–y) for high-content screening involving multi-well cell culture plates. The objectives include: 10× air Plan Apo lambda (0.45NA), 20× air S Plan Fluor ELWD (0.45 NA), 40× air S Plan Fluor ELWD 40x (0.6 NA), 40× water immersion (1.25NA), 40× air Plan Apo lambda (0.95 NA), a 60× water immersion Plan Apo IR (1.27NA). The imaging system has a "smart" auto-focus. NIS Elements AR Software was used for image analysis and NIS Elements HC Software for image acquisition. *Images were randomly acquired* from duplicate wells in each experimental condition. The number of independent experiments (*N*) is indicated in figure legends. For each field of view, 17 to 30 planes were acquired with a step size of 0.3 μm using a 60× water immersion objective. Image deconvolution was done using the Richardson-Lucy algorithm and the best-focused projection image was created using the extended depth of focus module. *Widefield microscopy (Zeiss):* Images were acquired on a motorized, inverted, wide-field fluorescence microscope (Axio Observer D1, TempModule S, heating unit XL S; Zeiss) equipped with a PZ-2000 XYZ automated stage, 20× Plan Neofluar (numerical aperture [NA] = 0.5), 40× Plan Neofluar (NA = 1.3), and 63× Plan Apochromat (NA = 1.4) objectives, a high-speed Xenon fluorescence emission device (Lambda DG-4, 300 W; Sutter Instrument Company), a Lambda 10-3 optical emission filter wheel for the fluorescence imaging, a SmartShutter to control the illumination for phase-contrast and DIC imaging (Sutter Instrument Company), an ORCA-Flash 4.0 sCMOS camera (Hamamatsu). The filter sets (Chroma Technology Corporation) were as follows: DAPI (49000), Alexa Fluor 488 (49002), Alexa Fluor 568 (49005), and Cy5 (49006). Images were acquired and analyzed using MetaMorph imaging software (Molecular Devices). *SoRa confocal microscope (Nikon):* Nikon Ti2-E microscope equipped with the Yokogawa CSU-W1 spinning disk with a 50 μm pinhole size (4000 rpm); a 60X Plan Apo lambda D OFN25 DIC N2 (NA 1.42) oil objective; and Quest Camera (apparent pixel size is 0.027 μm/pixel) from Hamamatsu. *Resonant laser scan confocal A1R plus (Nikon):* Nikon Ti2-E microscope equipped with Plan Apo lambda 60x oil (NA 1.4), DU4 detector, pinhole size 39.59. For confocal images, 17–30 planes were acquired per field of view with a 0.1 or 0.2 μm step size. Images were denoised with denoise.ai (artificial intelligence) and deconvolved using the Richardson-Lucy algorithm (NIS Elements, Nikon). *N-STORM System, Stochastic Optical Reconstruction Microscopy (STORM) (Nikon):* Nikon Ti2-E microscope equipped with a SR HP Apo TRIF 100xH Oil objective (NA 1.49), an ORCA Fusion BT sCMOS camera (Hamamatsu), and a LUD-H series laser unit (405 nm, 488 nm, 561 nm, and 640 nm). The XY resolution is 20 nm and the Z resolution is 50 nm. Images (z stacks) were acquired using 3D-STORM with a 0.2 μm step size. *3D ANXA2 Counting:* Z stack images were acquired by widefield fluorescence microscopy (60× water objective, 0.3 μm steps). Images were denoised with denoise.ai (artificial intelligence) and deconvolved using the Richardson-Lucy algorithm (NIS Elements, Nikon). The cellular area excluding the nuclei was selected. The "count 3D objects" tool was then used in the Nikon NIS software to automatically count the 3D objects which represent the total number of ANXA2 specks.

## Statistical analyses

For most of the data such as experiments with cell counts and fluorescence intensity as the measurements, log transformation was first used to reduce skewness and variation. ANOVA or linear mixed-effects models were then used for analyses depending on whether the data was independent or with repeated measurements over time. For the screening data, a *p*-value $< 0.05$ without adjustment for multiple comparisons was used as the cutoff because the results were further validated in the subsequent assays. A *p*-value $< 0.05$ for a single test or after adjustment for multiple comparisons with Holm's procedure was considered significant. Additionally, GraphPad Prism software was used to perform one-tailed Students Paired or Unpaired T-Tests and a *p*-value $< 0.05$ was considered significant. All $N = x$ values describe the number of independent experiments.

# Data availability

Source microscopy images were deposited in BioImage Archives. Accession Number: S-BIAD1122.

The source data of this paper are collected in the following database record: biostudies:S-SCDT-10_1038-S44319-024-00195-6.

# Peer review information

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

## Acknowledgements

Research reported in this publication was supported by the Institute of Allergy and Infectious Diseases of the National Institutes of Health under award number R01AI157205 and R03AI164337. The content is solely the responsibility of the authors and does not necessarily represent the official views of the National Institutes of Health. Viviana Ruiz received financial public funding from the Colombian Ministry of Science, Technology, and Innovation (Ministerio de Ciencia, Tecnología e Innovación, MinCiencias). We acknowledge resources from the Campus Microscopy and Imaging Facility (CMIF) and the OSU Comprehensive Cancer Center (OSUCCC) Microscopy Shared Resource (MSR), The Ohio State University. This facility is supported in part by grant P30 CA016058, National Cancer Institute, Bethesda, MD. We acknowledge the precious and kind help received from Dr. Anthony Vetter for assisting with fluorescence microscopy imaging and analysis using the NIS Elements software. We also acknowledge BioRender.com which was used to create the synopsis thumbnail.

## Author contributions

**M Isabella Prislusky**: Data curation; Formal analysis; Investigation; Methodology; Writing—original draft; Writing—review and editing. **Jonathan G T Lam**: Data curation; Formal analysis; Investigation; Methodology. **Viviana Ruiz Contreras**: Data curation; Formal analysis; Investigation; Methodology. **Marilynn Ng**: Data curation; Formal analysis; Investigation; Methodology. **Madeline Chamberlain**: Data curation; Formal analysis. **Sarika Pathak-Sharma**: Data curation; Formal analysis; Investigation; Methodology. **Madalyn Fields**: Writing—review and editing. **Xiaoli Zhang**: Formal analysis; Writing—review and editing. **Amal O Amer**: Writing—review and editing. **Stephanie Seveau**: Conceptualization; Resources; Formal analysis; Supervision; Funding acquisition; Validation; Investigation; Visualization; Methodology; Writing—original draft; Project administration; Writing—review and editing.

Source data underlying figure panels in this paper may have individual authorship assigned. Where available, figure panel/source data authorship is listed in the following database record: biostudies:S-SCDT-10_1038-S44319-024-00195-6.

## Disclosure and competing interests statement

The authors declare no competing interests.

# Expanded View Figures

**Figure EV1. SEPT2 and SEPT9 silencing do not affect plasma membrane repair.**

(A) HeLa cells were transfected with Ctr.-, SEPT2-, or SEPT9-siRNAs (Dataset EV1). After 72 h, cells were lysed and analyzed by SDS-PAGE and immunoblotting for septin and tubulin expression. Serial dilutions of control- (100% to 6.25%) and undiluted septin- (100%) siRNA treated cell lysates were loaded in the gel to facilitate the quantification of the KD efficiencies. Blots are representative of at least $N = 5$. KD efficiencies were >90%. (A′) HeLa cells were treated for 72 h with Ctr. or SEPT-siRNAs and were exposed to LLO (0.5 nM) for 30 min in M1 containing TO-PRO-3. Data are the average TO-PRO-3 fluorescence intensities expressed in arbitrary units (AI) ± SEM of at least $N = 3$ independent experiments at time point 30 min. (B) HeLa cells were exposed, or not, to 0.5 nM LLO in M1 or M2 for 5 min on ice to allow LLO to bind the plasma membrane. Cells were washed and warmed up to 37 °C for 5, 10, 15, 30 min, 2 h, and 24 h. In the last min of incubation, cells were exposed for 1 min to 100 μM propidium iodide. About 1300 cells were analyzed per experimental condition and data are expressed as the average nuclear Propidium Iodide fluorescence intensities expressed in arbitrary units (AI) ± SEM of at least $N = 3$ independent experiments. $N = 4$ independent experiments for the 5-30 min conditions. Data show that plasma membrane integrity is recovered fully between 30 min to 2 h, at the whole population level. (C–F) HeLa cells were transfected with Ctr.- or SEPT-siRNAs as in (A). Blots are representative of at least $N = 5$. (E) Comparing control siRNA-treated cells to SEPT7-siRNA-treated cells, SEPT6 expression was reduced by 44.8% ± 24.4; 70% ± 21.2; and 52.3% ± 19.6 for siRNA1, 2, and 3, respectively, SEPT2 expression was reduced by 59.6% ± 4.3; 46.4% ± 7.7; and 40.1% ± 6.9 for siRNA1, 2, and 3, respectively. SEPT 9 expression was reduced by at least 90% for all three siRNAs. KD of SEPT2, 6, and 9 did not appear to affect the expression of the other tested septins. (G) HeLa cells were transfected with non-targeting Ctr. siRNA, SEPT7-siRNA 1, or SEPT2-siRNA 2. Cells were incubated without LLO for 15 min in M1 (1.2 mM Ca$^{2+}$), fixed, permeabilized, and labeled with anti-SEPT9 primary Ab (and Alexa Fluor 568-conjugated secondary), F-actin (Alexa Fluor 488-conjugated phalloidin), and nuclei (DAPI). Z-stack images were acquired by widefield microscopy, denoised, deconvolved, and displayed as best-focus projection images. Scale bars: 10 μm. (H) The hemolytic assay showing that the presence of DOX at the indicated concentrations does not affect LLO pore formation ($N = 3$ independent experiments, Average ± SEM). Note that in the repair assay, DOX-treated cells were washed three times, so only traces of DOX were present in the medium during the repair assay. (I, J) HeLa cells were pre-treated with the indicated concentrations of FCF, or corresponding DMSO dilutions, or left untreated (Ctr.) for 16 h. Equivalent concentrations of FCF and DMSO were maintained in the buffer during the repair assay. Cells were exposed to LLO for 30 min in M1 or M2 supplemented with TO-PRO-3. Data are expressed as the average TO-PRO-3 fluorescence intensities in arbitrary unit of $N = 2$ to $N = 4$ (each with 4 to 8 technical replicates) ± SEM at each time point. (K) The hemolytic assay showed that FCF at the highest concentration does not affect LLO pore formation ($N = 3$ independent experiments, Average ± SEM). Data Information: (A′, B) Data were log$_{10}$ transformed and analyzed using linear mixed-effects models. *$P < 0.05$.

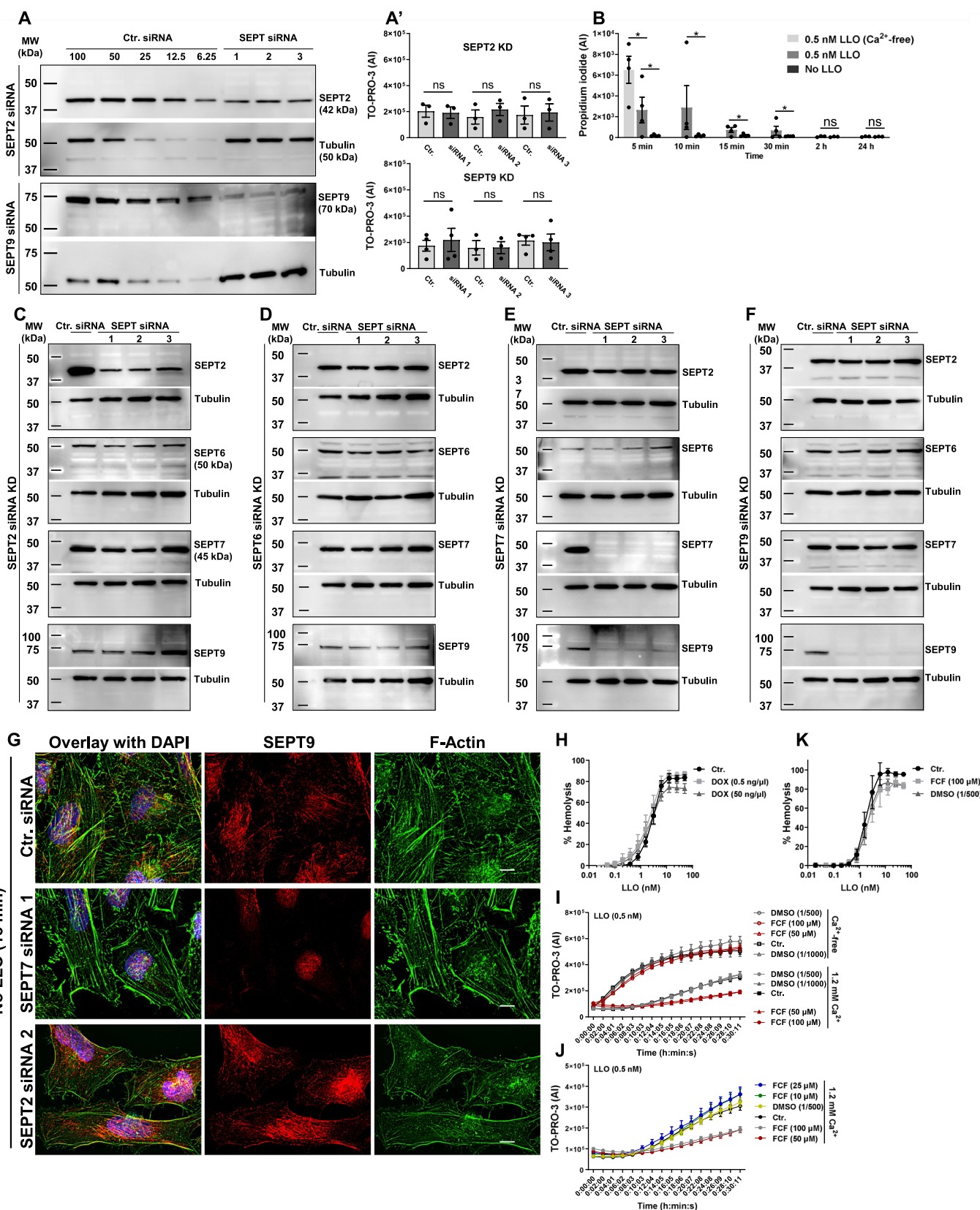

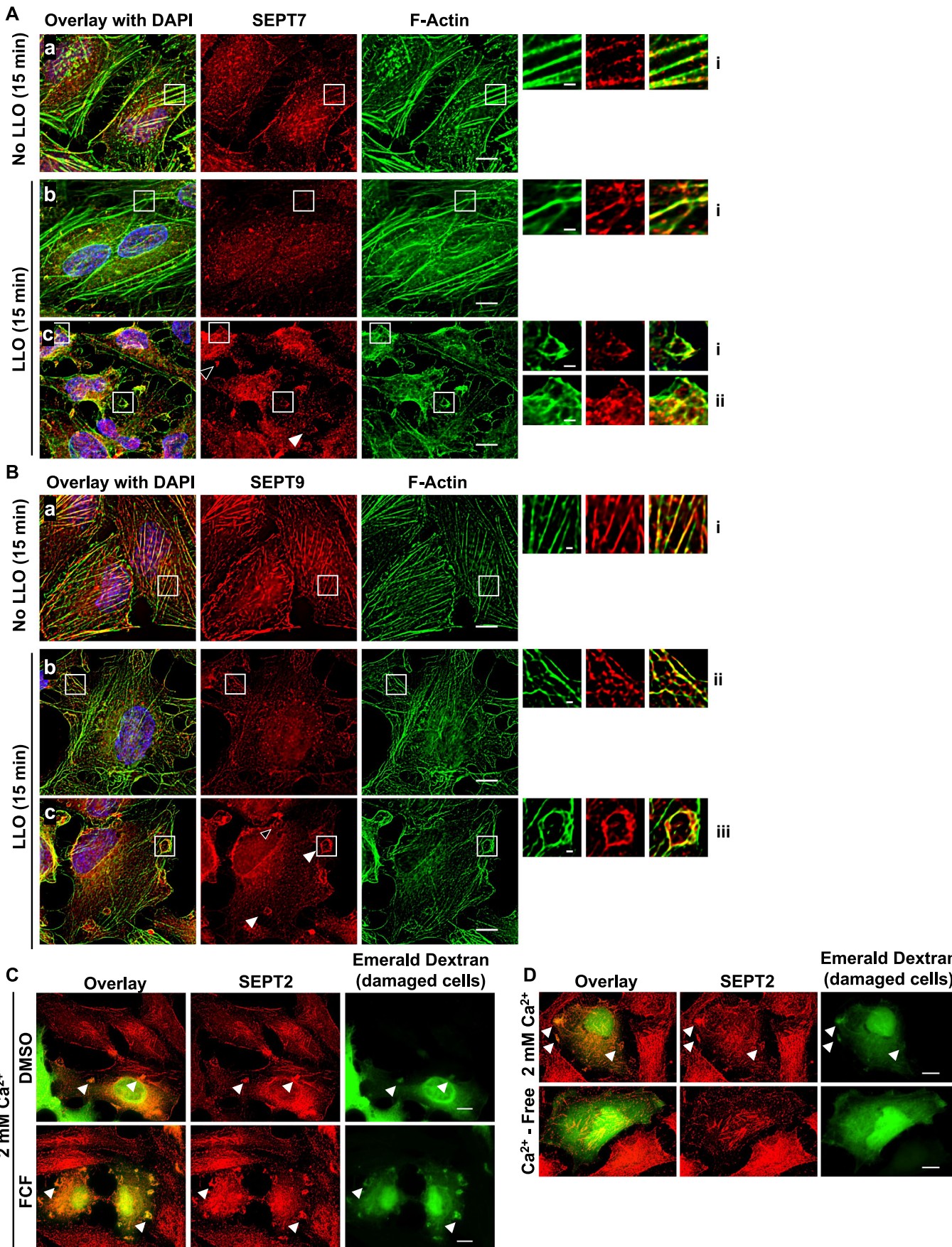

◀  **Figure EV2.  The septin cytoskeleton is remodeled in LLO-perforated and mechanically-wounded cells.**

HeLa cells were incubated without (No LLO) or with 0.5 nM LLO for 5–15 min in M1. Cells were chemically fixed, permeabilized, and fluorescently labeled with anti-SEPT9, or SEPT7, or SEPT2 primary Abs (Alexa Fluor 568-conjugated secondary), F-Actin (Alexa Fluor 488-conjugated phalloidin), and nuclei (DAPI). (A, B) SEPT7 (**A**) and SEPT9 (**B**) fluorescence images are presented with the same intensity scaling showing the loss of septin association with actin stress fibers. To better visualize septin and actin filaments, selected regions were enlarged (**A**ai, **A**bi, **A**ci,ii, **B**ai, **B**bii, **B**ciii) and the septin fluorescence display was the same for all images except for Abi and Bbii which intensity was amplified. All images were acquired by z-stack widefield microscopy, deconvolved, and presented as the best focus images, except for **A**a, **A**b, **B**a, and **B**b images which are single planes focused on actin stress fibers. Representative septin knob and loop structures are indicated by unfilled and filled arrowheads, respectively. Scale bars are 10 μm and 2 μm in the enlarged images (**C, D**) HeLa cells, pre-treated with 100 μM FCF, with vehicle DMSO (**C**) or untreated (**D**), were mechanically wounded in M1 (2 mM Ca$^{2+}$) or M2 (Ca$^{2+}$- Free) medium in the presence of Emerald Dextran to identify the sites of cell wounding. Cells were washed, fixed, permeabilized and fluorescently labeled with anti-septin primary Abs (Alexa Fluor 647-conjugated secondary), and nuclei (DAPI). All images were acquired by z-stacks widefield microscopy, denoised, deconvolved, and presented as the best focus images. Scale bars are 10 μm. Filled arrowheads point to remodeled septin structures.

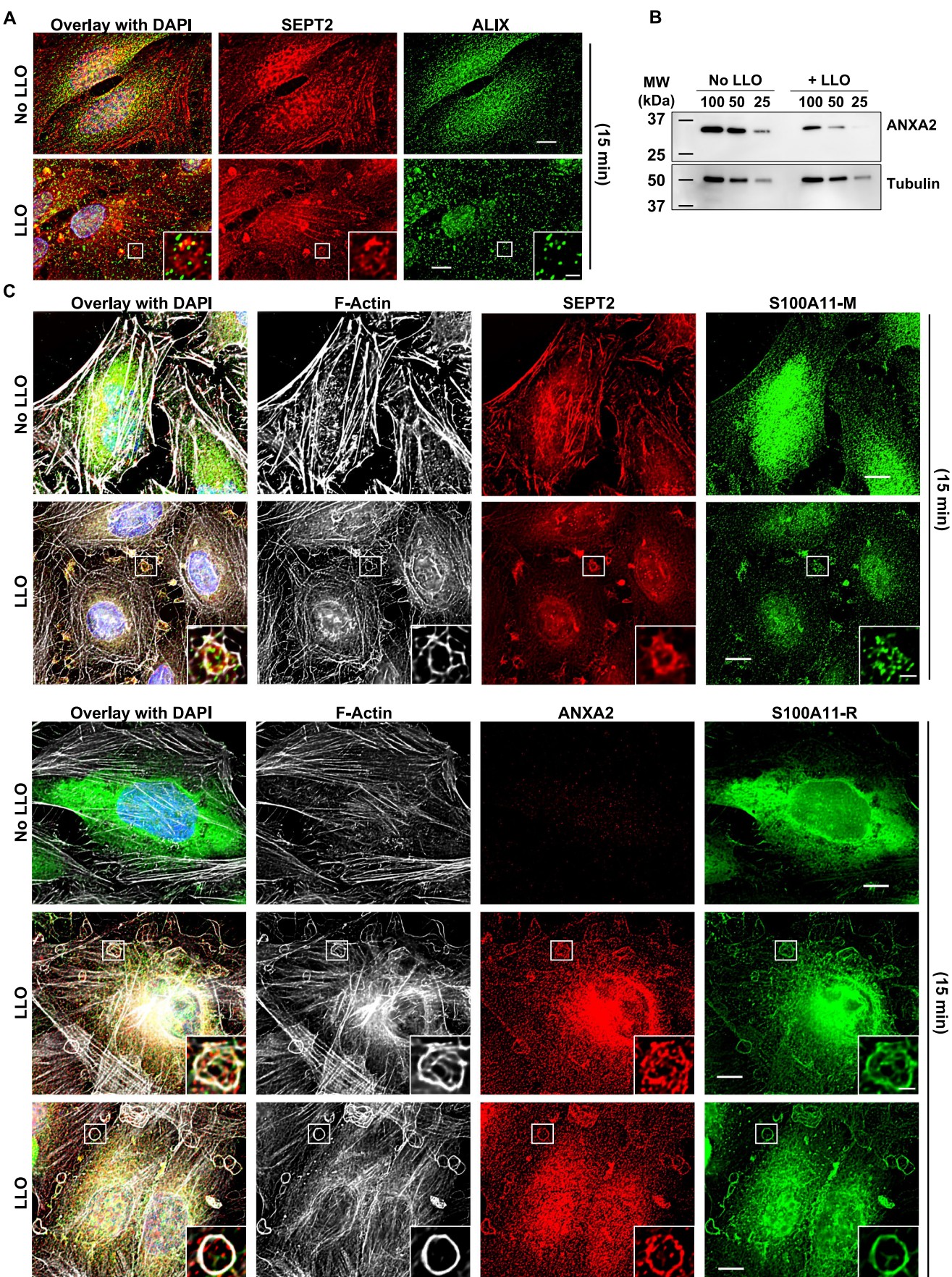

◀ **Figure EV3. The septin cytoskeleton redistributes with S100A11.**

(A) Cells were fluorescently labeled for SEPT2 (Alexa Fluor 568-conjugated secondary), ALIX (Alexa Fluor 488-conjugated secondary), and nuclei (DAPI) at time point 15 min. (B) HeLa cells were exposed, or not, to 0.5 nM LLO for 15 min in M1. Cells were lysed and analyzed by SDS-PAGE and immunoblotting to measure ANXA2 expression level. Serial dilutions (100% to 25%) of cell lysates were loaded in the gel (representative of $N = 3$ independent experiments) and tubulin was used as loading control. Data showed that ANXA2 expression was not increased under LLO treatment. (C) HeLa cells were fluorescently labeled for SEPT2 (Alexa Fluor 568-conjugated secondary), F-Actin (Alexa Fluor 488-conjugated phalloidin), ANXA2 (Alexa Fluor 647-conjugated secondary), S100A11 (either anti-mouse (-M) Alexa Fluor 647-conjugated secondary or anti-rabbit (-R) Alexa Fluor 568-conjugated secondary) and nuclei (DAPI) at time point 15 min. Data Information: In (A and C), all images were acquired by z-stacks widefield microscopy, denoised, deconvolved, and presented as the best focus images except (A) which are single planes. Scale bars are 10 μm or 2 μm in zoomed-in regions.

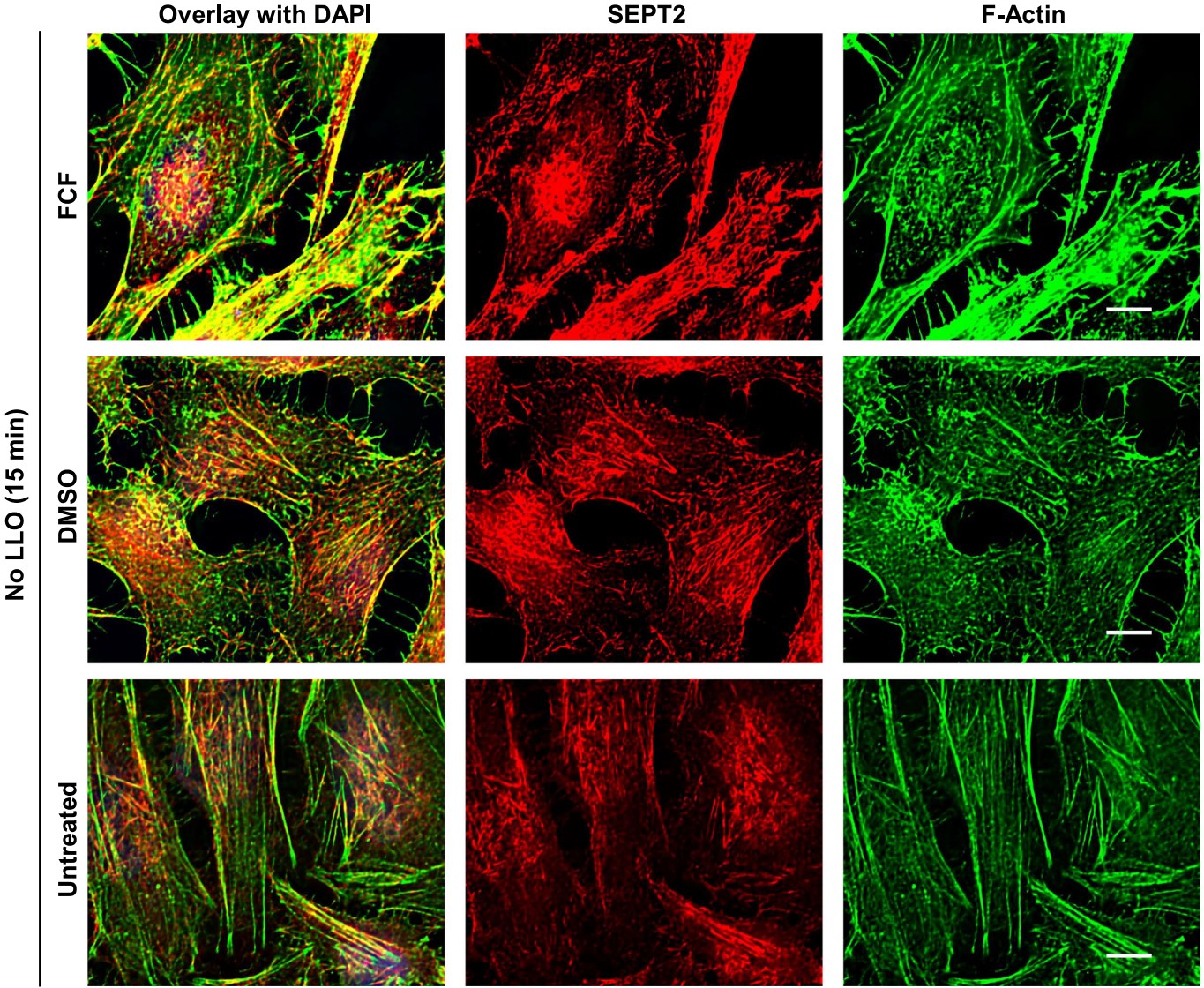

**Figure EV4. Organization of the septin and actin cytoskeletons in FCF- and DMSO-pre-treated control cells.**

Untreated, FCF (100 μM) pre-treated, and vehicle DMSO pre-treated HeLa cells were incubated without LLO (No LLO) for 15 min in M1. Cells were fixed, permeabilized, and fluorescently labeled for SEPT2 (Alexa Fluor 568-conjugated secondary Abs), F-actin (Alexa Fluor 647-conjugated phalloidin), and nuclei (DAPI). Z-stack images were acquired by widefield microscopy and were denoised, deconvolved, and presented as best-focus projection images. Scale bars are 10 μm.

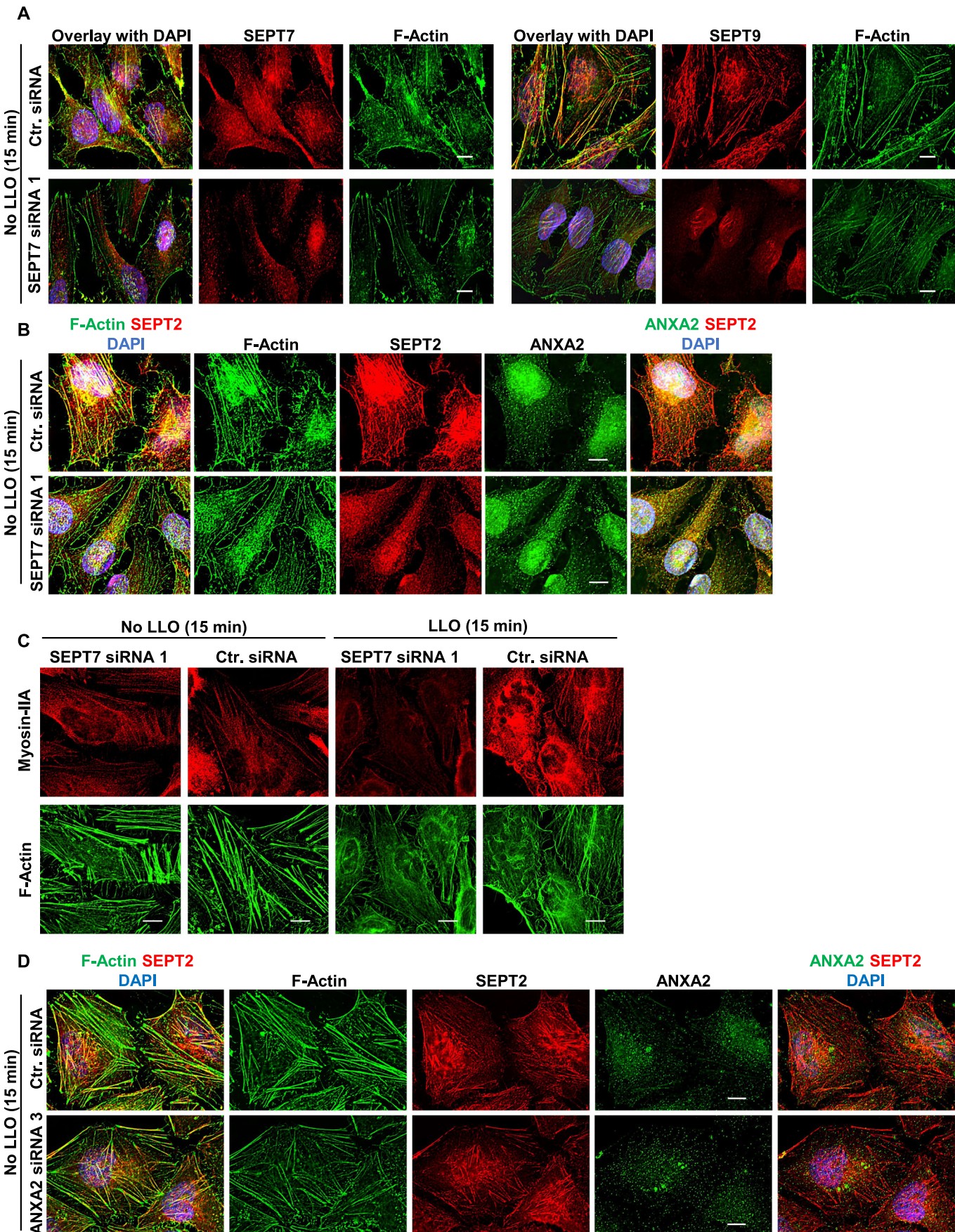

◀ **Figure EV5.   Myosin-IIA redistribution is SEPT7-dependent.**

(**A–D**) HeLa cells were transfected with Ctr. siRNA, SEPT7-siRNA 1 (**A–C**), or ANXA2-siRNA 3 (**D**) for 72 h. Cells were incubated with (**C**) or without (**A–D**) LLO for 15 min in M1. Cells were fixed, permeabilized, and labeled with SEPT Ab (Alexa Fluor 568-conjugated secondary), F-Actin (Alexa Fluor 488-conjugated phalloidin), nuclei (DAPI), (**B, D**) ANXA2 (Alexa Fluor 647-conjugated secondary) and (**C**) Myosin IIA (Alexa Fluor 568-conjugated secondary). Scale bars: 10 μm. Data Information: (**B–D**) Z-stack images were acquired by widefield microscopy, denoised, deconvolved, and displayed as best-focus projection images, except for (**A**) which are single-plane images focused on actin stress fibers.

