## [Peer Review File · EMBO Reports]

The Septin Cytoskeleton is Required for Plasma Membrane Repair

M. Isabella Prislusky, Jonathan GT Lam, Viviana Ruiz Contreras, Marilyn Ng, Madeline Chamberlain, Sarika Pathak-Sharma, Madalyn Fields, Xiaoli Zhang, Amal O. Amer, and Stephanie Seveau

Corresponding author(s): Stephanie Seveau (Stephanie.Seveau@osumc.edu)

Review Timeline:

Transfer Date:	27th Mar 24
Editorial Decision:	4th Apr 24
Revision Received:	9th Apr 24
Editorial Decision:	8th May 24
Revision Received:	30th May 24
Accepted:	7th Jun 24

**Transaction Report: This manuscript was transferred to
EMBO reports following peer review at Review Commons.**

**Review
COMMONS**

Review #1

1. Evidence, reproducibility and clarity:

Evidence, reproducibility and clarity (Required)

Review of "The septin cytoskeleton is required for plasma membrane repair" by Prislusky et al.

Eukaryotic cells rapidly repair damage to their plasma membrane and underlying cortical cytoskeleton. Such repair is increasingly recognized as being of major importance to human health (PMID: 33849525). Two broadly conserved cell damage responses have been described: a very rapid membrane resealing response which commences within a second or so following damage, and a cortical cytoskeletal response which commences within ~15-30s and which is based on activation of the Rho GTPases. However, our understanding of either of these responses is extremely limited, a situation which has engendered considerable debate about not only the mechanistic bases of these responses but also their relative roles and the extent to which they may be interdependent.

In the current study, the authors use an siRNA screen to identify Septin7 (hereinafter SEPT7) as a critical participant in the cell repair response. They further demonstrate that cell damage, as induced either by bacterial pore-forming proteins or by mechanical abrasion results in accumulation of septins (including SEPT7) in curious ring-like structures associated at the plasma membranes; these structures are often associated with plasma membrane protrusions, which are a common feature of damaged cells. Additionally, the authors show that the septins colocalize with F-actin, myosin-2 (an F-actin-based motor protein) and annexin-2A, a protein previously implicated in cell repair. Lastly, the authors show that depletion of septins reduces the recruitment of annexin-2A to the plasma membrane in wounded cells, implying that the septins are upstream of the annexin in the wound response.

This is an exciting study that is also very well-documented. The excitement is provided by the following observations: first, septins have not previously been implicated in cell repair; second, the association of the septins with F-actin and myosin-2 in ring-like structures at the plasma membrane is suggestive of the possibility that local contraction may promote healing, a long-standing idea derived from studies of frog oocyte healing (PMID: 10359696; PMID: 11502762) which has proven controversial for healing of other cell types (see below); third, a link between septins and annexins in cell repair or, for that matter, any other process, is novel. With

respect to the support for their claims, the authors go above and beyond to make their case-every point is supported by multiple approaches-for example the importance of septins is shown via siRNA, shRNA, and inducible depletion-and the imaging is very, very nice.

The potential role for actomyosin-powered contraction in healing of wounds made in cultured mammalian cells has been largely discounted because of studies wherein cells are wounded after pharmacological treatment with actin poisons have shown that healing is actually improved. The problem with such studies, is that depolymerization of actin prior to cell damage will dramatically alter the response to damage due to loss of cortical tension (PMID: 19846787). Thus, besides being important in its own right, the current study opens new doors for experimental assessment of the possible roles for cortical actomyosin in cell repair.

I have only minor concerns or questions:

1. What is the spatial relationship between the septin rings and actual damage sites? This could be addressed by wounding in the presence of a lysine fixable dextran.
2. The information in table 1 could be made more reader-friendly. In particular, it is not clear how the authors are getting their gene/protein names for their hits and what they correspond to. This was most noticeable for IQSEC1, ABI1, and GBF1 which the authors describe in the text as "genes that control the actin cytoskeleton" but in the table are listed as "Signaling proteins". I may have the abbreviations wrong (which is more reason for additional clarity) but GBF1 is the abbreviation for a protein involved in intracellular trafficking; IQSEC1 is a GEF for Arf proteins, and ABI1 is best known as a subunit of the WAVE complex.
3. The statement that begins the abstract "Mammalian cells are frequently exposed to mechanical and biochemical stresses..." could just as easily be "Eukaryotic cells..." or even "Cells..." as the membrane repair response is apparently universal and, indeed, was first described in nonmammalian cells. Similarly, the introduction begins "The plasma membrane of mammalian cells forms a biophysical barrier that separates the cell from its external environment". As far as I know, this is not a specific feature of mammalian plasma membranes but rather all plasma membranes. I don't know if it is the author's intention to imply their work is only relevant to mammals, but that is certainly not the case and they end up reducing the impact of their work by making it sound like cell repair is a phenomenon specific to mammalian cells.
4. The word "subplasmalemmal" is likely to be confusing for those who are not aware that plasmalemma is an antiquated term for the plasma membrane. It might be easier for the reader if the authors refer to "subdomains of the plasma membrane".

2. Significance:

Significance (Required)

This is an exciting study that is also very well-documented. The excitement is provided by the following observations: first, septins have not previously been implicated in cell repair; second, the association of the septins with F-actin and myosin-2 in ring-like structures at the plasma membrane is suggestive of the possibility that local contraction may promote healing, a long-standing idea derived from studies of frog oocyte healing (PMID: 10359696; PMID: 11502762) which has proven controversial for healing of other cell types (see below); third, a link between septins and annexins in cell repair or, for that matter, any other process, is novel. With respect to the support for their claims, the authors go above and beyond to make their case-every point is supported by multiple approaches-for example the importance of septins is shown via siRNA, shRNA, and inducible depletion-and the imaging is very, very nice.

The potential role for actomyosin-powered contraction in healing of wounds made in cultured mammalian cells has been largely discounted because of studies wherein cells are wounded after pharmacological treatment with actin poisons have shown that healing is actually improved. The problem with such studies, is that depolymerization of actin prior to cell damage will dramatically alter the response to damage due to loss of cortical tension (PMID: 19846787). Thus, besides being important in its own right, the current study opens new doors for experimental assessment of the possible roles for cortical actomyosin in cell repair.

3. How much time do you estimate the authors will need to complete the suggested revisions:

Estimated time to Complete Revisions (Required)

(Decision Recommendation)

Less than 1 month

No

Review #2

1. Evidence, reproducibility and clarity:

Evidence, reproducibility and clarity (Required)

Summary

This study highlights the role of the septin cytoskeleton in plasma membrane repair in HeLa cells perforated by the pore-forming toxin listeriolysin O (LLO). The authors performed a silencing RNA screen targeting protein-coding genes involved in endocytosis, exocytosis and intracellular trafficking. Besides the recovery of proteins that were previously identified to be part of the membrane repair machinery, they uncovered novel plasma membrane repair candidates, including septin 7 (SEPT7).

They found that upon LLO treatment, septins redistribute from actin stress fibers to the cell surface where they form knobs and loops together with F-actin, Myosin-IIA and Annexin A2 (ANXA2). Using super resolution microscopy and 3D reconstruction, they showed that these structures often protruding from the cell surface are formed by septins and F-actin that are organized in intertwined filaments associated with Annexin A2, and that they are functionally correlated with plasma membrane repair efficiency. Silencing SEPT7 further revealed that the remodeling of the repair protein ANXA2 at the cell surface is greatly decreased in LLO-injured cells, whereas the down regulation of ANXA2 had no impact on the arrangement of septins and F-actin into knobs and loops.

Altogether, their results evidenced that the septin cytoskeleton triggers the organization of membrane domains containing the actomyosin cytoskeleton and ANXA2, that are essential for the repair to occur.

Major comments:

- The authors show that silencing SEPT6 or SEPT7, but not SEPT2 or SEPT9, perturbed plasma membrane repair of LLO-injured cells. The authors explain this result by indicating that the reduced expression of SEPT7 and

SEPT6 (according to the siRNA), but not that of SEPT2 results in a reduced expression of septins from other groups. This could have been an explanation but, in Fig. S2B, downregulating SEPT2 clearly seem to impact the expression of SEPT6 and SEPT7 (except for siRNA#3) once normalized with the loading control tubulin. Moreover, it is well accepted in the literature and has been observed in many cell types, including HeLa cells, that knocking down a septin from one group (with sometimes the exception of septins of Group 3) induces the downregulation of septins of the other groups, and that it consistently results in the loss of septin filaments. Therefore, the fact that silencing SEPT2 does not perturb plasma membrane repair is quite surprising. This could suggest that SEPT6 and SEPT7, independently of their filament organization, play a role in membrane repair after LLO treatment. Nevertheless, the SEPT2 staining to study the fate of septin filaments following LLO exposure indicated that it is the redistribution of septin filaments that is crucial in this repair process.

Interestingly, BORG proteins which are involved in the association of septin filaments to the actin cytoskeleton in interphase cells bind to the SEPT6/SEPT7 coiled-coil region of septin polymers. Could these proteins be involved, knowing that they are Cdc42 effector proteins, and that links exist between Cdc42 activation and Ca²⁺ entry?

OPTIONAL: silencing BORG proteins (BORG2 for example) and studying septin and F-actin remodeling following LLO exposure could help the authors to understand the reason of such a redistribution.

- What about the terms "knob" and "loop": Are they structurally related to the "specks" described in other papers? Or are they new structures that no one observed before? Nobody has never looked at septins in this repair process before, but actin has long been described to be involved.

- It seems that knobs are formed before loops take over. This would deserve further investigation. Is that a reality? Or are they two independent structures?

OPTIONAL: it would be interesting to do time-lapse video microscopy to follow the fate of a knob.

Related to the previous point: why to show 3 sets of images in the LLO condition in Fig. 2A? Does the top b-panel represent the knob stage? Where there are still many stress fibers indicating that septins have not yet fully redistributed? And when septins are fully dissociated from actin cables, which are then lost, loops are forming (middle c-panel) and then increase in size (bottom d-panel)?

- Even though, some information is given in the discussion section, it would be helpful to mention in the introduction section the different pathways that cells activate to repair plasma membrane defects, and to precise which one(s) has(ve) already been described in the literature to be switched on in response to the LLO toxin.

- Some experiments are not rigorous enough: Sometimes, they have not been repeated, as exemplified in Fig. 7B. Count less cells but repeat the experiment at least three

times. Sometimes, one condition is missing, as in Fig. 6C. Where is the DMSO condition? What about the statistics?

Fig. 6A: In the calcium free condition, it seems that the two cells that are illustrated depict a telophase. The subcellular organization is obviously different at the end of cell division. Show only one interphase cell as in the top panels.

Fig. 6B and C: It is mentioned in the figure legend: "Cells were treated as indicated in (A) and (D)". But the "C" condition is not mentioned anywhere: Does "C" stand for no DMSO or FCF treatment, in the presence of calcium, but under LLO treatment? Likewise, it would be very helpful to indicate in each figure panel whether cells have been treated with LLO or FCF. Please help the reader.

Fig. 7A: Whatever SEPT7 is expressed or downregulated, the actin stress fibers are still present. If these cells were not well transfected, replace the images.

- Concerning the FCF experiments: The FCF cytokinin has been used by many authors to perturb septin dynamics. It induces the stabilization of septin polymers, thus promoting the formation of thick ectopic fibers. It is a potent inducer of septin polymerization and acts as a stabilizer.

Fig. 6D: In the Ctr condition, a DMSO condition is needed to visualize the impact of FCF on septin filaments. Does FCF stabilize septins and induce the formation of thick filaments? The SEPT2 image in the FCF condition without LLO is of bad quality (see above remark).

Also in Fig. 6D (FCF condition without LLO), the F-actin staining revealed that there are no stress fibers!!??? Usually, the more septins are associated with actin, the thicker stress fibers you get, since septins stabilize actin cables. FCF treatment often induces thick ectopic septin filaments that are not associated with stress fibers (which are therefore lost). Was it the case in all FCF-treated cells? Does FCF treatment really mimic what happens physiologically in the cell? Many off-target effects have been observed with this molecule in non-plant cells.

- The image quality in Figs 3A and B, and 6A and D needs to be improved regarding the septin staining. In control conditions, septin filaments cannot be clearly distinguished.

- Fig. 3B: It seems that ANXA2 is overexpressed in LLO-injured cells. Its accumulation level between both conditions should be compared by immunoblot. ANXA2 is indeed recovered on loops, but it is difficult to consider whether it is a redistribution.

- Fig. 7D: Compared to the control condition (we have to refer to Fig. S5D), ANXA2 again seems to be overexpressed under LLO treatment. To affirm that ANXA2 remodeling in LLO-injured cells requires the formation of septin/F-actin knobs and loops, data in Fig. 7D must be quantified.

- Fig. 6 (B-D): In panel B, there is a significant difference between the "C" and "FCF" conditions regarding the number of knobs + loops per cell. Where are the images corresponding to the "C" condition?

****Minor comments:****

- Fig. 2A: Report the white squares (selected enlarged areas) in all panels (SEPT2 and overlay). In panels b, do not place an arrowhead where we are supposed to observe an enlarged area. Also, from panels b, it would be worth showing an enlarged area including a knob. Show enlarged areas also from panels d.

- Fig. 3B'i: Septins are not on stress fibers. Select a transfected cell where septins still coalign with actin fibers, not a cell that was impaired by the transfection.

- Fig. 3C: Add the time point "0min". What was the % of colocalization before LLO treatment? and in DMSO condition?

What about ALIX at 5 and 10min? Again, it's only one experiment.

- Figs 4 and 5: Very nice images but obtained following FCF exposure. Hopefully FCF would not have induced an aberrant organization!

- The "Ctr" abbreviation is often used, in different conditions, and may be confusing. Precise in the figure (not in the figure legend) whether it is siRNA ("Ctr siRNA"). Mention "DMSO" for the controls of your drugs (like in Figs S4 and S5).

- Fig. S4C: How is this figure different or does it provide additional information compared to Fig. 2C?

- Fig. S5D: It is hard to know that the ANXA2 siRNA worked, since no difference of staining between the Ctr and the transfected cells can be observed. Were these cells really transfected? It would have been helpful to use fluorescent siRNAs.

The same applies to Fig. S5C: Silencing SEPT7 supposedly greatly reduces the level of expression of all septins. The SEPT2 staining is still high, and many actin stress fibers are still observable (whereas the loss of septin filaments results in the loss of actin stress fibers, as observed by many authors, including in HeLa cells). Same remark for Fig. S6, regarding the SEPT7 silencing in the Ctr condition (no LLO). No impact on stress fibers! Are these cells transfected? The authors themselves mention that sometimes cells are less effectively silenced (like in Fig. 7A, B). Why not to show cells effectively silenced!!

- In the abstract, it is specified that SEPT7 also plays a role in membrane repair after mechanical wounding. Based only on one type of experiment (SEPT7 silencing, Fig. 1H), this statement should only be mentioned in the text or used to discuss the putative repair mechanisms that septins are involved in, but not stated in the abstract as a main conclusion.

2. Significance:

Significance (Required)

Strengths:

Despite septins have been involved in endocytosis, exocytosis, membrane protrusions, cell junction integrity or actomyosin constriction at cytokinesis, the involvement of the septin cytoskeleton in the plasma membrane repair machinery has, to my knowledge, never been reported before. The authors not only showed that septins are present in specific membrane protrusions (knobs and loops) but also evidenced that septin filaments trigger the formation of these plasma membrane repair domains by recruiting F-actin and ANXA2, essential for the repair to occur. The novelty of this study has therefore to be acknowledged, and these data will benefit the scientific community, and the septin community in particular.

This is a descriptive paper that nevertheless clearly shows, by different means, the reorganization of the septin cytoskeleton in LLO-injured cells. The use of high-resolution microscopy coupled to 3D reconstruction which enables to easily appreciate the organization of septins, F-actin and ANXA2 in the knobs and loops is a true strength of the paper.

Limitations:

The authors mention in the abstract that septins act as scaffolds to recruit contractile actin fibers and ANXA2. Biochemical experiments such as co-immunoprecipitations could strengthen this notion.

The molecular mechanism by which septins are involved in this repair process has not been addressed at all in the paper. Even though the silencing RNA screen highlighted several proteins involved in known membrane repair mechanisms, the authors just presented a few data concerning ALIX, a component of the ESCRT-III machinery. A % of colocalization of SEPT2 and SEPT7 with ALIX is reported in Fig. 3C but this experiment has only been done once (n=1) and only following 15-min exposure to LLO. Is that too late? Immunofluorescence images of SEPT2 and ALIX with or without LLO (15min) are also provided in Fig. S5A but no quantification is reported. Is it sufficient to say that the ESCRT machinery is not involved?

My field of expertise:

Cytoskeleton, Septin, Actin, Microtubule, Signaling pathways

3. How much time do you estimate the authors will need to complete the suggested revisions:

Estimated time to Complete Revisions (Required)

(Decision Recommendation)

Between 1 and 3 months

Yes

Manuscript number: RC-2023-02328

Corresponding author(s): Stephanie Seveau

We are grateful to the reviewers for their enthusiasm about the novelty of our findings and their insightful comments. Below is a point-by-point response to all comments (our responses are indicated in blue font and modifications already included in the revised article are underlined).

Reviewer #1 (Evidence, reproducibility and clarity (Required)):

Review of "The septin cytoskeleton is required for plasma membrane repair" by Prislusky et al.

Eukaryotic cells rapidly repair damage to their plasma membrane and underlying cortical cytoskeleton. Such repair is increasingly recognized as being of major importance to human health (PMID: 33849525). Two broadly conserved cell damage responses have been described: a very rapid membrane resealing response which commences within a second or so following damage, and a cortical cytoskeletal response which commences within ~15-30s and which is based on activation of the Rho GTPases. However, our understanding of either of these responses is extremely limited, a situation which has engendered considerable debate about not only the mechanistic bases of these responses but also their relative roles and the extent to which they may be interdependent.

In the current study, the authors use an siRNA screen to identify Septin7 (hereinafter SEPT7) as a critical participant in the cell repair response. They further demonstrate that cell damage, as induced either by bacterial pore-forming proteins or by mechanical abrasion results in accumulation of septins (including SEPT7) in curious ring-like structures associated at the plasma membranes; these structures are often associated with plasma membrane protrusions, which are a common feature of damaged cells. Additionally, the authors show that the septins colocalize with F-actin, myosin-2 (an F-actin-based motor protein) and annexin-2A, a protein previously implicated in cell repair. Lastly, the authors show that depletion of septins reduces the recruitment of annexin-2A to the plasma membrane in wounded cells, implying that the septins are upstream of the annexin in the wound response.

This is an exciting study that is also very well-documented. The excitement is provided by the following observations: first, septins have not previously been implicated in cell repair; second, the association of the septins with F-actin and myosin-2 in ring-like structures at the plasma membrane is suggestive of the possibility that local contraction may promote healing, a long-standing idea derived from studies of frog oocyte healing (PMID: 10359696; PMID: 11502762) which has proven controversial for healing of other cell types (see below); third, a link between septins and annexins in cell repair or, for that matter, any other process, is novel. With respect to the support for their claims, the authors go above and beyond to make their case-every point is supported by multiple approaches-for example the importance of septins is shown via siRNA, shRNA, and inducible depletion-and the imaging is very, very nice.

The potential role for actomyosin-powered contraction in healing of wounds made in cultured mammalian cells has been largely discounted because of studies wherein cells are wounded after pharmacological treatment with actin poisons have shown that healing is actually improved. The problem with such studies, is that depolymerization of actin prior to cell damage will dramatically alter the response to damage due to loss of cortical tension (PMID: 19846787).

Thus, besides being important in its own right, the current study opens new doors for experimental assessment of the possible roles for cortical actomyosin in cell repair. We thank the reviewer for his/her enthusiasm about our novel findings.

I have only minor concerns or questions:

1. What is the spatial relationship between the septin rings and actual damage sites? This could be addressed by wounding in the presence of a lysine fixable dextran.

The reviewer proposed to add lysine fixable fluorescent dextrans to the culture medium during cell wounding allowing the dextrans to flux into cells via the plasma membrane wounds. After washes, fixation, and labeling of the septins, we should be able to visualize the location of the membrane wounds (dextran-positive cell portions) relative to the redistributed septins. We tried this approach: Visualization of dextran fluxes through the 30-50 nM toxin transmembrane pores was very challenging at non-lethal toxin concentrations (0.5 nM). Dextran diffusion could only be observed in cells damaged with higher toxin concentrations or in cells damaged with non-lethal toxin concentrations in calcium-free medium. Furthermore, we could not visualize the sites of damage as the entire cytosol was uniformly fluorescent. We also used lysine-fixable fluorescent dextrans in mechanically damaged cells. This approach was more successful, likely due to the larger (μm) size of mechanical damages. We found that the dextrans entered the entire cell cytoplasm but the dextran fluorescence intensity was higher at the locations where the septins were redistributed. This is consistent with the notion that septins are redistributed in proximity to the wound sites which are dextrans entry sites (new Figure 3). Importantly, septins were only redistributed in damaged cells, but not in non-damaged surrounding cells which did not incorporate fluorescent dextrans, and FCF pre-treatment increased septin redistribution only in wounded cells. Collectively, these data support that the septins are redistributed on the edges of the mechanical wounds. We thank the reviewer for this great suggestion; we accordingly modified the results section (page 7, lines 207-211, and page 9, lines 275-277).

2. The information in table 1 could be made more reader-friendly. In particular, it is not clear how the authors are getting their gene/protein names for their hits and what they correspond to. This was most noticeable for IQSEC1, ABI1, and GBF1 which the authors describe in the text as "genes that control the actin cytoskeleton" but in the table are listed as "Signaling proteins". I may have the abbreviations wrong (which is more reason for additional clarity) but GBF1 is the abbreviation for a protein involved in intracellular trafficking; IQSEC1 is a GEF for Arf proteins, and ABI1 is best known as a subunit of the WAVE complex.

We verified all gene names using the HUGO Gene Nomenclature (HGNC, genenames.org) (PMID: 32747822) and the Human Gene Database (GeneCards.org). This information has been added to the legend of Table 1. We also rearranged Table 1 as follows: changed *M6PRBP1* for *PLIN3*, which is the approved gene name, and italicized all gene symbols. We replaced "signaling" with "Guanine Nucleotide Exchange Factors" including the GEFs IQSEC1 and GBF1. We created the category "Cytoskeletal Regulation" including SEPT7 and Abl1 and modified accordingly the text of the results section (page 5, line 131).

3. The statement that begins the abstract "Mammalian cells are frequently exposed to mechanical and biochemical stresses..." could just as easily be "Eukaryotic cells..." or even "Cells..." as the membrane repair response is apparently universal and, indeed, was first described in nonmammalian cells. Similarly, the introduction begins "The plasma membrane of mammalian cells forms a biophysical barrier that separates the cell from its external environment". As far as I know, this is not a specific feature of mammalian plasma membranes but rather all plasma membranes. I don't know if it is the author's intention to imply their work is

only relevant to mammals, but that is certainly not the case and they end up reducing the impact of their work by making it sound like cell repair is a phenomenon specific to mammalian cells. We accordingly modified the abstract and introduction.

4. The word "subplasmalemmal" is likely to be confusing for those who are not aware that plasmalemma is an antiquated term for the plasma membrane. It might be easier for the reader if the authors refer to "subdomains of the plasma membrane".

We replaced subplasmalemmal with "subdomains of the plasma membrane" or equivalent throughout the article.

Reviewer #2 (Evidence, reproducibility and clarity (Required)):

1. Evidence, reproducibility and clarity

Summary

This study highlights the role of the septin cytoskeleton in plasma membrane repair in HeLa cells perforated by the pore-forming toxin listeriolysin O (LLO). The authors performed a silencing RNA screen targeting protein-coding genes involved in endocytosis, exocytosis and intracellular trafficking. Besides the recovery of proteins that were previously identified to be part of the membrane repair machinery, they uncovered novel plasma membrane repair candidates, including septin 7 (SEPT7).

They found that upon LLO treatment, septins redistribute from actin stress fibers to the cell surface where they form knobs and loops together with F-actin, Myosin-IIA and Annexin A2 (ANXA2). Using super resolution microscopy and 3D reconstruction, they showed that these structures often protruding from the cell surface are formed by septins and F-actin that are organized in intertwined filaments associated with Annexin A2, and that they are functionally correlated with plasma membrane repair efficiency. Silencing SEPT7 further revealed that the remodeling of the repair protein ANXA2 at the cell surface is greatly decreased in LLO-injured cells, whereas the down regulation of ANXA2 had no impact on the arrangement of septins and F-actin into knobs and loops.

Altogether, their results evidenced that the septin cytoskeleton triggers the organization of membrane domains containing the actomyosin cytoskeleton and ANXA2, that are essential for the repair to occur.

Thank you for the great summary!

Major comments:

1- The authors show that silencing SEPT6 or SEPT7, but not SEPT2 or SEPT9, perturbed plasma membrane repair of LLO-injured cells.

The authors explain this result by indicating that the reduced expression of SEPT7 and SEPT6 (according to the siRNA), but not that of SEPT2 results in a reduced expression of septins from other groups.

This comment is accurate for SEPT7. However, our data do not support that SEPT6 silencing reduces the expression of other septins, and we did not claim that silencing SEPT6 resulted in reduced expression of septins from other groups.

This could have been an explanation but, in Fig. S2B, downregulating SEPT2 clearly seem to impact the expression of SEPT6 and SEPT7 (except for siRNA#3) once normalized with the loading control tubulin. Moreover, it is well accepted in the literature and has been observed in many cell types, including HeLa cells, that knocking down a septin from one group (with sometimes the exception of septins of Group 3) induces the downregulation of septins of the other groups, and that it consistently results in the loss of septin filaments.

Therefore, the fact that silencing SEPT2 does not perturb plasma membrane repair is quite surprising. This could suggest that SEPT6 and SEPT7, independently of their filament organization, play a role in membrane repair after LLO treatment. Nevertheless, the SEPT2 staining to study the fate of septin filaments following LLO exposure indicated that it is the redistribution of septin filaments that is crucial in this repair process.

We reported that silencing SEPT2 had little effect on the expression of other septins such as SEPT7 (page 6, line 143; SF 2B). If requested, we can include additional examples of western blots. Quantification of western blots showed that SEPT7 expression was only decreased in the range of 10 to 30% in SEPT2 KD cells, which is not sufficient to abolish the formation of the septin filaments. These results are consistent with the literature that reported "*Depletion of SEPT2 results only in a modest decrease of other septins.*" (PMID: 21737677). The same article showed that another member of the SEPT2 subgroup, such as SEPT5, needs to be depleted together with SEPT2 to strongly affect SEPT7 expression. Another article showed that knocking down SEPT2 expression had no noticeable effect on SEPT7 expression in HeLa cells (PMID: 27215606). Therefore, it is not surprising that SEPT2 silencing in our experiments does not perturb plasma membrane repair. To clarify this important point, we included fluorescence images of SEPT2 and SEPT7 knocked down cells in comparison to cells exposed to non-targeting siRNAs (new S. Fig. 3). These images importantly illustrate that the formation of septin filaments was severely decreased in SEPT7 knocked down cells. However, in SEPT2 knocked down cells, septin filaments were only slightly decreased (new S. Fig.3). We added this information to the result section (page 5, lines 141-144).

Interestingly, BORG proteins which are involved in the association of septin filaments to the actin cytoskeleton in interphase cells bind to the SEPT6/SEPT7 coiled-coil region of septin polymers. Could these proteins be involved, knowing that they are Cdc42 effector proteins, and that links exist between Cdc42 activation and Ca²⁺ entry?

OPTIONAL: silencing BORG proteins (BORG2 for example) and studying septin and F-actin remodeling following LLO exposure could help the authors to understand the reason of such a redistribution.

We agree that BORG proteins might be involved in remodeling the septins during plasma membrane repair; we thank the reviewer for this great comment. As the reviewer mentions, this would be optional. Our article is already very dense and not only reports the general role of the septins in plasma membrane repair, but it also characterizes the septin remodeling during repair and provides mechanistic information about the structural role of the septins in remodeling the actomyosin cytoskeleton and recruitment of repair molecules such as ANXA2 for cell repair. We feel that the study of BORG proteins will be the object of our future studies and another research article.

2- What about the terms "knob" and "loop": Are they structurally related to the "specks" described in other papers? Or are they new structures that no one observed before? Nobody has never looked at septins in this repair process before, but actin has long been described to be involved.

We do not know what is meant by “specks”, a reference would have been helpful. Previous studies, including our studies, showed that LLO and other cholesterol-dependent cytolysins remodel the cortical actin cytoskeleton in a calcium-dependent fashion, but this redistribution had not been attributed to cell repair (PMID: 17301241, PMID: 29187576). These published results are consistent with our findings. As explained by the reviewer, nobody has ever looked at the structure of the septins and co-localization between septins and cortical actin during plasma membrane repair. This explains why these structures have not been previously noticed.

3- It seems that knobs are formed before loops take over. This would deserve further investigation. Is that a reality? Or are they two independent structures?

- OPTIONAL: it would be interesting to do time-lapse video microscopy to follow the fate of a knob.

As written in the discussion: “The knobs may be an early stage of the loop formation based on the timing of their respective appearance (Figure 2C, SF 5C)”. We agree that performing live-cell imaging in future studies will help in establishing if the two structures are independent or not. We feel that refining their detailed dynamics will be the object of future research studies.

Related to the previous point: why to show 3 sets of images in the LLO condition in Fig. 2A?

Several images are required to properly document septin redistribution: (1) Fig2Aa (no LLO) and 2Ab (+LLO) are single plan images focused on the bottom membranes where actin stress fibers are most abundant. Together, these pictures highlight that a fraction of the septins dissociates from the actin stress fibers. (2) Fig2Ac and 2Ad are reconstructions from z-stacks to show the overall structure of the septin cytoskeleton at all focal plans in LLO-treated cells.

These images highlight that the septin cytoskeleton reorganizes into knob- and loop-like structures in association with cortical F-actin.

Does the top b-panel represent the knob stage? The top b panel shows that a fraction of septins dissociated from actin stress fibers and shows examples of knob structures. We also added a zoomed-in image of a knob structure in 2Abi.

Where there are still many stress fibers indicating that septins have not yet fully redistributed?

We report that septins partially dissociate from actin stress fibers which are still present in LLO-exposed cells as shown in Figure 2Ab and 2Abii and SF 5Abi, 5Bbii.

And when septins are fully dissociated from actin cables, which are then lost, loops are forming (middle c-panel) and then increase in size (bottom d-panel)? We did not claim that the septins fully dissociate from actin stress fibers, we did not claim that actin stress fibers were lost in LLO-treated cells, and we did not claim that the septin loops increase in size over time. We explained that the septins partially dissociate from the stress fibers and stress fibers are still present in LLO-injured cells. For scientific rigor, our fluorescence images are presented with the same scaling. Therefore, comparing control to LLO-injured cells, we see an important drop in septin fluorescence using the same scaling. However, we increased the display of the septin fluorescence in the zoomed images, as in Fig. 2Abii, where we can appreciate that a proportion of septins is still associated with the actin stress fibers. Our images showed the presence of actin stress fibers in LLO-injured cells and a smaller proportion of septins is still associated with the actin stress fibers compared to control cells (Fig. 2Ab, Fig. 2Abii, S. Fig. 5Ab, S. Fig. 5Abi, S. Fig. 5Bb, and S. Fig. 5Bbii). This is clarified on page 7 lines 204-206 and in figure 2 legend.

4- Even though, some information is given in the discussion section, it would be helpful to mention in the introduction section the different pathways that cells activate to repair plasma

membrane defects, and to precise which one(s) has(ve) already been described in the literature to be switched on in response to the LLO toxin.

We added a sentence in the introduction on page 3, lines 76-80.

5-Some experiments are not rigorous enough: Sometimes, they have not been repeated, as exemplified in Fig. 7B (now Fig. 8B). Count less cells but repeat the experiment at least three times.

All of our conclusions were made based on data obtained from the average of at least three independent experiments ($N \geq 3$). Sometimes data were presented with $N=2$ or 1 for additional, non-essential qualitative information. For example, the septin cytoskeleton is made of the assembly of SEPT 2, 7 and 9. As shown in Fig. 2, S. Figs.5 and 6, septin filaments can be indiscriminately visualized using anti-SEPT2, anti-SEPT7, or anti-SEPT9 antibodies. For example, in Figure 4 C, we counted $N=3$ using cells labeled with anti-SEPT2 antibodies and $N=3$ with cells labeled with anti SEPT-7 antibodies leading to highly similar data. All quantitative analyses were performed counting cells labeled with anti-SEPT2 antibodies ($N \geq 3$) in Fig. 8B. In Fig. 8B, we presented additional data counted on cells labeled with anti-SEPT7 and anti-SEPT9 antibodies, which showed again similar results as with cells labeled with anti-SEPT2 antibodies, as expected. Therefore, it would make little sense to repeat counting cells labeled with anti-SEPT9 or anti-SEPT7 antibodies in Fig. 8B. All cell counting at time point 15 min were performed with \$N \geq 3\$. We agree that it is important to reach \$N=3\$ for the time points 5 and 10 min. These experiments are already carried out and we will further quantify time points 5 and 10 min in Fig. 2B, 2C, S. Fig. 5C, and Fig. 4C. We will also reach \$N=3\$ for ALIX colocalization at 15 min in Fig. 4C.

Sometimes, one condition is missing, as in Fig. 6C (now Fig. 7C). Where is the DMSO condition? What about the statistics?

Fig. 7C: the DMSO control was added to the figure. All statistical analyses of all data presented in the entire article are described in the methods and in the figure legends.

Fig. 6A (now Fig. 7A): In the calcium free condition, it seems that the two cells that are illustrated depict a telophase. The subcellular organization is obviously different at the end of cell division. Show only one interphase cell as in the top panels.

We do not think that the cells depict a telophase. Indeed, in calcium-free medium and in the presence of LLO, cell morphology is affected by excessive damage. To avoid any confusion, we selected another representative image in Fig. 7A.

Fig. 6B and C (now Fig. 7B and C): It is mentioned in the figure legend: "Cells were treated as indicated in (A) and (D)". But the "C" condition is not mentioned anywhere: Does "C" stand for no DMSO or FCF treatment, in the presence of calcium, but under LLO treatment?

Thanks for the comment, we clarified the figure and the figure legend in Fig. 7B and C. Yes, C was the condition without DMSO or FCF and with calcium.

Likewise, it would be very helpful to indicate in each figure panel whether cells have been treated with LLO or FCF. Please help the reader.

Thanks for the comment, we clarified all figure legends.

Fig. 7A (now Fig. 8A): Whatever SEPT7 is expressed or downregulated, the actin stress fibers are still present. If these cells were not well transfected, replace the images.

All presented images are representative. There were less actin stress fibers in SEPT7 silenced cells, but stress fibers were still present. Please see Fig.8A; S. Fig. 6B, C; S. Fig. 8.

6- Concerning the FCF experiments: The FCF cytokinin has been used by many authors to perturb septin dynamics. It induces the stabilization of septin polymers, thus promoting the formation of thick ectopic fibers. It is a potent inducer of septin polymerization and acts as a stabilizer.

Fig. 6D (now Fig. 7D): In the Ctr condition, a DMSO condition is needed to visualize the impact of FCF on septin filaments.

An additional image corresponding to the control DMSO was added.

Does FCF stabilize septins and induce the formation of thick filaments?

This is correct, we observed thicker septin filaments in FCF-treated cells, as shown in Sup. Figure 7.

The SEPT2 image in the FCF condition without LLO is of bad quality (see above remark).

Corresponding images were replaced, and a new S. Fig. 7 was added.

Also in Fig. 6D (now Fig. 7D) (FCF condition without LLO), the F-actin staining revealed that there are no stress fibers!???

There are actin stress fibers in FCF-treated cells, we replaced images of FCF-treated cells (without LLO) in the new S. Fig. 7.

Usually, the more septins are associated with actin, the thicker stress fibers you get, since septins stabilize actin cables. FCF treatment often induces thick ectopic septin filaments that are not associated with stress fibers (which are therefore lost). Was it the case in all FCF-treated cells?

We did not observe a large number of ectopic septin filaments, septins were mostly associated with F-actin in FCF-treated cells. We did lose some stress fibers (stress fibers were less frequent) but not in all cells. Images in new S. Fig 7 show less stress fibers, which is consistent across all experiments. Cortical actin was still very much present in the loop structures. A lot of the FCF-treated cells had “thicker bundles” of septins located either in the middle in a clump or along the edges of the cells.

Does FCF treatment really mimic what happens physiologically in the cell? Many off-target effects have been observed with this molecule in non-plant cells.

FCF is the only drug known to specifically affect septin filaments. FCF was used as an alternative approach to perturb septin organization. As with other drugs, we cannot rule out potential off-target effects, this is the reason why we used diverse experimental approaches. Importantly, FCF increases the formation of septin loops only in damaged cells (both mechanically damaged and injured by LLO). Collectively, these findings support that stabilization of the septin cytoskeleton by FCF stabilizes the septin structures formed during cell repair.

7- The image quality in Figs 3A and B (now 4A and B), and 6A and D (now 7A and D) needs to be improved regarding the septin staining. In control conditions, septin filaments cannot be clearly distinguished.

Septin filaments are clearly seen in Fig. 4A, we replaced the No LLO ANXA2-GFP cells in Fig. 4B. Images 7A and D were replaced.

8- Fig. 3B (now 4B): It seems that ANXA2 is overexpressed in LLO-injured cells. Its accumulation level between both conditions should be compared by immunoblot.

ANXA2 is not overexpressed within the time frame of the experiment (15 min). The difference observed in control versus LLO-treated cells comes from the redistribution of ANXA2. ANXA2 is diffusively dispersed in non-injured, control cells and is massively redistributed to accumulate at the plasma membrane in LLO-injured cells leading to a brighter labeling. This is a common phenomenon observed with cytoplasmic proteins which staining becomes bright and obvious when suddenly recruited to the plasma membrane or organelles, and ANXA2 is well-known to be recruited to the plasma membrane of injured cells.

ANXA2 is indeed recovered on loops, but it is difficult to consider whether it is a redistribution. The loops were not formed on non-injured cells and ANXA2 was cytoplasmic in non-injured cells. Therefore, the presence of ANXA2 on the newly formed septin loops can only correspond to ANXA2 redistribution.

9- Fig. 7D (now 8D): Compared to the control condition (we have to refer to Fig. S5D, now SF 6D), ANXA2 again seems to be overexpressed under LLO treatment.

ANXA2 is not overexpressed, see comments above.

To affirm that ANXA2 remodeling in LLO-injured cells requires the formation of septin/F-actin knobs and loops, data in Fig. 7D (now Fig. 8D) must be quantified.

Figure 8D shows ANXA2 silenced cells, therefore we cannot measure ANXA2 remodeling in this experimental condition. We quantified ANXA2 remodeling in SEPT7-deficient cells in comparison to cells treated with non-targeting siRNA, this result is presented in Figure 8C.

10- Fig. 6 (B-D (now 7 B-D)): In panel B, there is a significant difference between the "C" and "FCF" conditions regarding the number of knobs + loops per cell. Where are the images corresponding to the "C" condition?

Images corresponding to the "C" was added to Fig. 7. We also clarified the legends of this figure and added control images corresponding to the same experimental conditions without exposure to LLO in the **new S. Fig.7**. These modifications were necessary to improve clarity.

Minor comments:

11- Fig. 2A: Report the white squares (selected enlarged areas) in all panels (SEPT2 and overlay). In panels b, do not place an arrowhead where we are supposed to observe an enlarged area. Also, from panels b, it would be worth showing an enlarged area including a knob. Show enlarged areas also from panels d.

White squares were added. We also added magnified images to highlight the knob structures (2A_{bi}) and additional loop structures (2A_{di}).

12- Fig. 3B'i (now 4B'i): Septins are not on stress fibers. Select a transfected cell where septins still coalign with actin fibers, not a cell that was impaired by the transfection.

We agree. Image 4B was changed, its corresponding enlarged panel shows the co-alignment of the septins with F-actin in 4B_i'.

13- Fig. 3C (now 4C): Add the time point "0min". What was the % of colocalization before LLO treatment? and in DMSO condition?

At the time points 0, 5, 10 and 15 min, in absence of LLO, ANXA2 is not recruited to the plasma membrane and the septin knobs and loops are not formed. This is why we presented only the time point 15 min without LLO. There is no DMSO in these experiments.

What about ALIX at 5 and 10min? Again, it's only one experiment.

We observed minimal colocalization between ALIX and the septins at all studied time points in multiple experiments. However, we noticed a very clear and high level of co-localization between ANXA2 and the septins. This is the reason why we prefer investing our time in characterizing the co-localization between ANXA2 and the septins, the topic of this article. We show that ALIX displays poor co-localization with the septins at all time points, counting cells from one of our representative experiments. We will count more experiments to reach N=3 at 15 min. Of note, only conclusions were made on ANXA2, we do not make any conclusions about the ESCRT-III (ALIX) in this article.

14- Figs 4 and 5 (now 5 and 6): Very nice images but obtained following FCF exposure. Hopefully FCF would not have induced an aberrant organization!

Thank you. We showed numerous images of septin redistribution in the presence and absence of FCF throughout the article. In both experimental conditions, septins redistribute to form knobs and loop-like structures only on damaged cells. The difference with the FCF treatment was quantified in Fig. 4, which shows the more pronounced redistribution of the septins in FCF-treated cells, in accordance with the stabilization of septins by FCF. These structures are only observed in damaged cells which collectively shows that their formation is not an artifact due to FCF.

15- The "Ctr" abbreviation is often used, in different conditions, and may be confusing. Precise in the figure (not in the figure legend) whether it is siRNA ("Ctr siRNA"). Mention "DMSO" for the controls of your drugs (like in Figs S4 and S5, now S5 and S6).

We clarified all control abbreviations in all figures.

16- Fig. S4C (now S5C): How is this figure different or does it provide additional information compared to Fig. 2C?

Fig. S5C shows the number of knobs and loops per cell that display knobs or loops, whereas Fig. 2C shows the number of knobs and loops per total cells (including cells that do not display knob or loops). The supplemental figure is not essential and could be removed if requested.

17- Fig. S5D (now S6D): It is hard to know that the ANXA2 siRNA worked, since no difference of staining between the Ctr and the transfected cells can be observed. Were these cells really transfected? It would have been helpful to use fluorescent siRNAs.

ANXA2 silencing was effective, we quantified 88% silencing with siRNA3 by western blot in S. Fig. 4H.

The same applies to Fig. S5C (now S6C): Silencing SEPT7 supposedly greatly reduces the level of expression of all septins. The SEPT2 staining is still high, and many actin stress fibers are still observable (whereas the loss of septin filaments results in the loss of actin stress fibers, as observed by many authors, including in HeLa cells). Same remark for Fig. S6 (now S7), regarding the SEPT7 silencing in the Ctr condition (no LLO). No impact on stress fibers! Are these cells transfected? The authors themselves mention that sometimes cells are less effectively silenced (like in Fig. 7A, B, now 8A, B). Why not to show cells effectively silenced!!

As mentioned in our responses above, SEPT7 silencing does reduce (but does not abolish) the expression of other septins as we showed by western blotting and imaging. F-actin stress fibers were still present in SEPT7 silenced cells. The presented images are representative of our experimental data.

18- In the abstract, it is specified that SEPT7 also plays a role in membrane repair after mechanical wounding. Based only on one type of experiment (SEPT7 silencing, Fig. 1H), this statement should only be mentioned in the text or used to discuss the putative repair mechanisms that septins are involved in, but not stated in the abstract as a main conclusion. The assay used to assess repair of mechanical damage is a reliable assay commonly used in the literature. This assay is sufficient to report that septins are required for the repair of mechanical wounds. Importantly, we included ANXA2 silenced cells as a positive control to validate our assay, as shown previously in the literature. Furthermore, we added a new figure (new Fig. 3) showing the redistribution of the septins in mechanically wounded cells. Finally, this new figure shows that the redistribution occurs at the site of mechanical damages, and FCF enhanced the redistribution of septins in mechanically wounded cells. Collectively, we feel confident to write in the abstract that the septin cytoskeleton is required for the repair of mechanical wounds.

Reviewer #2 (Significance (Required)):

2. SIGNIFICANCE

Strengths:

Despite septins have been involved in endocytosis, exocytosis, membrane protrusions, cell junction integrity or actomyosin constriction at cytokinesis, the involvement of the septin cytoskeleton in the plasma membrane repair machinery has, to my knowledge, never been reported before. The authors not only showed that septins are present in specific membrane protrusions (knobs and loops) but also evidenced that septin filaments trigger the formation of these plasma membrane repair domains by recruiting F-actin and ANXA2, essential for the repair to occur. The novelty of this study has therefore to be acknowledged, and these data will benefit the scientific community, and the septin community in particular.

This is a descriptive paper that nevertheless clearly shows, by different means, the reorganization of the septin cytoskeleton in LLO-injured cells. The use of high-resolution microscopy coupled to 3D reconstruction which enables to easily appreciate the organization of septins, F-actin and ANXA2 in the knobs and loops is a true strength of the paper.

Limitations:

The authors mention in the abstract that septins act as scaffolds to recruit contractile actin fibers and ANXA2. Biochemical experiments such as co-immunoprecipitations could strengthen this notion.

The last sentence of our abstract is as follows: “*Collectively, our data support a novel model in which the septin cytoskeleton acts as a scaffold to promote the formation of plasma membrane repair domains containing contractile F-actin and annexin A2.*”

We believe that the presented data strongly support this proposed model without the need for further experiments. Indeed, we demonstrated that (i) septins are required for plasma membrane repair; (ii) septins are required for the remodeling of the submembranous actin cytoskeleton to form intertwined actin/septin filaments in close association with the repair protein ANXA2, as shown by super-resolution microscopy; and (iii) septins are also required for the recruitment of ANXA2. Finally, ANXA2 is well-known to be critical for the repair of transmembrane pores and the repair of mechanical wounds. Collectively, these results strongly

support the structural scaffolding role of the septin cytoskeleton in membrane repair domain formation.

The molecular mechanism by which septins are involved in this repair process has not been addressed at all in the paper.

See comments above, our studies are mechanistic and address the mechanism by which the septins are involved in repair: they control the formation of subdomains of the plasma membrane for the recruitment of molecules that repair the plasma membrane, i.e. annexin A2.

Even though the silencing RNA screen highlighted several proteins involved in known membrane repair mechanisms, the authors just presented a few data concerning ALIX, a component of the ESCRT-III machinery. A % of colocalization of SEPT2 and SEPT7 with ALIX is reported in Fig. 3C but this experiment has only been done once (n=1) and only following 15-min exposure to LLO. Is that too late? Immunofluorescence images of SEPT2 and ALIX with or without LLO (15min) are also provided in Fig. S5A but no quantification is reported. Is it sufficient to say that the ESCRT machinery is not involved?

We reply to the comment about ALIX co-localization in point # 13 above. Our article did not make a conclusion about the ESCRT-III machinery. We understand the enthusiasm of the reviewer in asking additional questions. We do as well. However, one research article is not sufficient to discover everything about the novel role of the septins in plasma membrane repair. This article demonstrates the novel role of the septins in the repair of the plasma membrane damaged by mechanical wounds and transmembrane pores. The article also describes how the septins are redistributed during cell repair in a manner that correlates with repair efficiency, and their role in redistributing cortical actin and recruiting repair molecules such as ANXA2. We did not observe a co-localization with ALIX as we did with ANXA2, but we did not conclude that ESCRT-III is not involved.

Dear Stephanie,

Thank you for the transfer of your research manuscript to our journal. As discussed, we are interested in publishing your study at EMBO Reports, revised according to the submitted revision plan. You have already addressed most of the referee concerns in the manuscript, except for the analysis of additional samples according to the suggestions from referee #2, point 5. I am now inviting formal revision at EMBO Reports, which will allow you to submit the updated manuscript files and referee response. Once we have received the re-submission, your manuscript will be sent back to the referees who evaluated your study at Review Commons.

I detail our formatting guidelines below but list here a few points that are specific to your manuscript.

- 1) Please remove the movie legends from the manuscript and supply them as separate README.txt files. Then zip the movie together with its legend and upload the zipped file.
- 2) You have 7 supplemental figures. You can either promote 5 of these to the 'Expanded View' format (see below) or bundle all of them in an Appendix PDF. If you choose the first option, the figure legends remain in the manuscript text as a separate paragraph titled "Expanded View Figure legends". If you choose the Appendix option, then all figures and their legends are part of the Appendix PDF, we need a title page with page numbers.
- 3) Tables are black and white only. If you wish to keep the colour of Table 1 you could generate a Supplemental Table (Appendix Table S1) or supply it as Table EV1 in .xls format with color.
- 4) Supplemental table 1 and 2 are Datasets. Please upload these as Dataset EV1 and EV2 in .xls format. The legend is part of the .xls file (first tab).
- 5) Supplemental Table 3 could be a Table in the methods.
- 6) Please remove the Abbreviations paragraph and make sure to explain abbreviations when they are first used in the text.
- 7) We need a 'Disclosure and competing interests statement'.
- 8) References: 'et al' is used if there are more than 10 authors, i.e., the first 10 are listed followed by 'et al'.

General formatting guidelines

- 1) a .docx formatted version of the manuscript text (including legends for main figures, EV figures and tables). Please make sure that the changes are highlighted to be clearly visible.
- 2) individual production quality figure files as .eps, .tif, .jpg (one file per figure). Please download our Figure Preparation Guidelines (figure preparation pdf) from our Author Guidelines pages <https://www.embopress.org/page/journal/14693178/authorguide> for more info on how to prepare your figures.
- 3) a .docx formatted letter INCLUDING the reviewers' reports and your detailed point-by-point responses to their comments. As part of the EMBO Press transparent editorial process, the point-by-point response is part of the Review Process File (RPF), which will be published alongside your paper.
- 4) a complete author checklist, which you can download from our author guidelines (). Please insert information in the checklist that is also reflected in the manuscript. The completed author checklist will also be part of the RPF.
- 5) Please note that all corresponding authors are required to supply an ORCID ID for their name upon submission of a revised manuscript (). Please find instructions on how to link your ORCID ID to your account in our manuscript tracking system in our Author guidelines ()
- 6) We replaced Supplementary Information with Expanded View (EV) Figures and Tables that are collapsible/expandable online. A maximum of 5 EV Figures can be typeset. EV Figures should be cited as 'Figure EV1, Figure EV2' etc... in the text and their respective legends should be included in the main text after the legends of regular figures.

- For the figures that you do NOT wish to display as Expanded View figures, they should be bundled together with their legends in a single PDF file called *Appendix*, which should start with a short Table of Content. Appendix figures should be referred to in

the main text as: "Appendix Figure S1, Appendix Figure S2" etc. See detailed instructions regarding expanded view here:

7) Please note that a Data Availability section at the end of Materials and Methods is now mandatory. In case you have no data that requires deposition in a public database, please state so instead of refereeing to the database.

See also < <https://www.embopress.org/page/journal/14693178/authorguide#dataavailability>>. Please note that the Data Availability Section is restricted to new primary data that are part of this study.

Additional information on source data and instruction on how to label the files are available .

10) Figure legends and data quantification:

- the name of the statistical test used to generate error bars and P values,
- the number (n) of independent experiments (please specify technical or biological replicates) underlying each data point,
- the nature of the bars and error bars (s.d., s.e.m.)

- If the data are obtained from n {less than or equal to} 5, show the individual data points in addition to the SD or SEM.

- If the data are obtained from n {less than or equal to} 2, use scatter blots showing the individual data points.

11) Our journal encourages inclusion of *data citations in the reference list* to directly cite datasets that were re-used and obtained from public databases. Data citations in the article text are distinct from normal bibliographical citations and should directly link to the database records from which the data can be accessed. In the main text, data citations are formatted as follows: "Data ref: Smith et al, 2001" or "Data ref: NCBI Sequence Read Archive PRJNA342805, 2017". In the Reference list, data citations must be labeled with "[DATASET]". A data reference must provide the database name, accession number/identifiers and a resolvable link to the landing page from which the data can be accessed at the end of the reference. Further instructions are available at .

12) All Materials and Methods need to be described in the main text. We would encourage you to use 'Structured Methods', our new Methods format. According to this format, the Methods section should include a Reagents and Tools Table (listing key reagents, experimental models, software and relevant equipment and including their sources and relevant identifiers) followed by a Methods and Protocols section in which we encourage the authors to describe their methods using a step-by-step protocol format with bullet points, to facilitate the adoption of the methodologies across labs. More information on how to adhere to this format as well as downloadable templates (.doc or .xls) for the Reagents and Tools Table can be found in our author guidelines: < <https://www.embopress.org/page/journal/14693178/authorguide#manuscriptpreparation>>.

An example of a Method paper with Structured Methods can be found here: .

13) As part of the EMBO publication's Transparent Editorial Process, EMBO Reports publishes online a Review Process File to accompany accepted manuscripts. This File will be published in conjunction with your paper and will include the referee reports, your point-by-point response and all pertinent correspondence relating to the manuscript.

You are able to opt out of this by letting the editorial office know (emboreports@embo.org). If you do opt out, the Review Process File link will point to the following statement: "No Review Process File is available with this article, as the authors have

chosen not to make the review process public in this case."

I look forward to seeing a revised form of your manuscript when it is ready.

Kind regards,

Martina

Manuscript number: RC-2023-02328

Corresponding author(s): Stephanie Seveau

We are grateful to the reviewers for their insightful comments and their enthusiasm about the novelty of our findings.

Reviewer 1 [***This is an exciting study that is also very well-documented. The excitement is provided by the following observations: first, septins have not previously been implicated in cell repair; second, the association of the septins with F-actin and myosin-2 in ring-like structures at the plasma membrane is suggestive of the possibility that local contraction may promote healing, a long-standing idea derived from studies of frog oocyte healing". And "third, a link between septins and annexins in cell repair or, for that matter, any other process, is novel. With respect to the support for their claims, the authors go above and beyond to make their case-every point is supported by multiple approaches-for example the importance of septins is shown via siRNA, shRNA, and inducible depletion-and the imaging is very, very nice."***]

Reviewer 2 [***Altogether, their results evidenced that the septin cytoskeleton triggers the organization of membrane domains containing the actomyosin cytoskeleton and ANXA2, that are essential for the repair to occur."*** And ***The authors not only showed that septins are present in specific membrane protrusions (knobs and loops) but also evidenced that septin filaments trigger the formation of these plasma membrane repair domains by recruiting F-actin and ANXA2, essential for the repair to occur. The novelty of this study has therefore to be acknowledged, and these data will benefit the scientific community, and the septin community in particular."***].

Below are point-by-point responses to all comments (our responses are indicated in blue, and modifications included in the revised article are underlined). The article and figures were updated based on our added results and the article has been reformatted.

Reviewer #1 (Evidence, reproducibility and clarity (Required)):

Review of "The septin cytoskeleton is required for plasma membrane repair" by Prislusky et al.

Eukaryotic cells rapidly repair damage to their plasma membrane and underlying cortical cytoskeleton. Such repair is increasingly recognized as being of major importance to human health (PMID: 33849525). Two broadly conserved cell damage responses have been described: a very rapid membrane resealing response which commences within a second or so following damage, and a cortical cytoskeletal response which commences within ~15-30s and which is based on activation of the Rho GTPases. However, our understanding of either of these responses is extremely limited, a situation which has engendered considerable debate about not only the mechanistic bases of these responses but also their relative roles and the extent to which they may be interdependent.

In the current study, the authors use an siRNA screen to identify Septin7 (hereinafter SEPT7) as a critical participant in the cell repair response. They further demonstrate that cell damage, as induced either by bacterial pore-forming proteins or by mechanical abrasion results in accumulation of septins (including SEPT7) in curious ring-like structures associated at the

plasma membranes; these structures are often associated with plasma membrane protrusions, which are a common feature of damaged cells. Additionally, the authors show that the septins colocalize with F-actin, myosin-2 (an F-actin-based motor protein) and annexin-2A, a protein previously implicated in cell repair. Lastly, the authors show that depletion of septins reduces the recruitment of annexin-2A to the plasma membrane in wounded cells, implying that the septins are upstream of the annexin in the wound response.

This is an exciting study that is also very well-documented. The excitement is provided by the following observations: first, septins have not previously been implicated in cell repair; second, the association of the septins with F-actin and myosin-2 in ring-like structures at the plasma membrane is suggestive of the possibility that local contraction may promote healing, a long-standing idea derived from studies of frog oocyte healing (PMID: 10359696; PMID: 11502762) which has proven controversial for healing of other cell types (see below); third, a link between septins and annexins in cell repair or, for that matter, any other process, is novel. With respect to the support for their claims, the authors go above and beyond to make their case—every point is supported by multiple approaches—for example the importance of septins is shown via siRNA, shRNA, and inducible depletion—and the imaging is very, very nice.

The potential role for actomyosin-powered contraction in healing of wounds made in cultured mammalian cells has been largely discounted because of studies wherein cells are wounded after pharmacological treatment with actin poisons have shown that healing is actually improved. The problem with such studies, is that depolymerization of actin prior to cell damage will dramatically alter the response to damage due to loss of cortical tension (PMID: 19846787). Thus, besides being important in its own right, the current study opens new doors for experimental assessment of the possible roles for cortical actomyosin in cell repair.

We thank the reviewer for his/her enthusiasm about our findings.

I have only minor concerns or questions:

1. What is the spatial relationship between the septin rings and actual damage sites? This could be addressed by wounding in the presence of a lysine fixable dextran.

This is a great suggestion. The presence of lysine fixable fluorescent dextran in the culture medium during cell wounding may allow visualizing the plasma membrane wounds as cell areas with high dextran fluorescence intensities (due to dextran influx via the wounds). After washes, chemical fixation, and fluorescent labeling of the septins, we should be able to visualize the location of the membrane wounds (as dextran-positive cell portions) relative to the redistributed septins. We tried this approach: Visualization of dextran fluxes through the 30-50 nm toxin transmembrane pores was very challenging at non-lethal toxin concentrations (0.5 nM). Dextran diffusion could only be observed in cells damaged with higher toxin concentrations or in cells damaged with non-lethal toxin concentrations in calcium-free medium. Furthermore, when cells were dextran-positive, we could not visualize the sites of damage as the entire cytosol was uniformly fluorescent. We also used lysine-fixable fluorescent dextran in mechanically damaged cells. We found that although the dextran entered the entire cell cytoplasm, its fluorescence intensity was increased at locations where the septins were redistributed. This is consistent with the notion that septins are redistributed in proximity to the wound sites which are dextran entry sites (new Figure EV 2C). Importantly, septins were only redistributed in damaged cells (dextran positive cells) but not in non-damaged surrounding cells which did not incorporate the fluorescent dextran. Also, cell treatment with FCF increased septin redistribution in mechanically wounded cells (dextran positive), as we previously observed in toxin-perforated cells, but not in dextran negative cells. Again, in FCF-treated cells, redistributed septins co-localized with cell portions of high dextran fluorescence. Collectively, these data support that the septins are

redistributed on the edges of the mechanical wounds. We thank the reviewer for this great suggestion; we accordingly modified the results section (page 7, lines 203-205, and page 9, lines 265/266).

2. The information in table 1 (now Table EV1) could be made more reader-friendly. In particular, it is not clear how the authors are getting their gene/protein names for their hits and what they correspond to. This was most noticeable for IQSEC1, ABI1, and GBF1 which the authors describe in the text as “genes that control the actin cytoskeleton” but in the table are listed as “Signaling proteins”. I may have the abbreviations wrong (which is more reason for additional clarity) but GBF1 is the abbreviation for a protein involved in intracellular trafficking; IQSEC1 is a GEF for Arf proteins, and ABI1 is best known as a subunit of the WAVE complex.

We verified all gene names using the HUGO Gene Nomenclature (HGNC, genenames.org) (PMID: 32747822) and the Human Gene Database (GeneCards.org). This information has been added to the legend of Table EV1. We also rearranged Table EV1 as follows: changed *M6PRBP1* for *PLIN3*, which is the approved gene name, and italicized all gene symbols. We replaced “signaling” with “Guanine Nucleotide Exchange Factors” including the GEFs IQSEC1 and GBF1. We created the category “Cytoskeletal Regulation” including SEPT7 and Abl1 and modified accordingly the text of the results section (page 5, lines 126-128).

3. The statement that begins the abstract “Mammalian cells are frequently exposed to mechanical and biochemical stresses...” could just as easily be “Eukaryotic cells...” or even “Cells...” as the membrane repair response is apparently universal and, indeed, was first described in nonmammalian cells. Similarly, the introduction begins “The plasma membrane of mammalian cells forms a biophysical barrier that separates the cell from its external environment”. As far as I know, this is not a specific feature of mammalian plasma membranes but rather all plasma membranes. I don't know if it is the author's intention to imply their work is only relevant to mammals, but that is certainly not the case and they end up reducing the impact of their work by making it sound like cell repair is a phenomenon specific to mammalian cells. We accordingly modified the abstract and introduction. We shortened the abstract to improve its overall clarity and impact.

4. The word “subplasmalemmal” is likely to be confusing for those who are not aware that plasmalemma is an antiquated term for the plasma membrane. It might be easier for the reader if the authors refer to “subdomains of the plasma membrane”.

We replaced subplasmalemmal with “subdomains of the plasma membrane” or equivalent (e.g. submembranous domains) throughout the article.

Reviewer #2 (Evidence, reproducibility and clarity (Required)):

1. Evidence, reproducibility and clarity

Summary

This study highlights the role of the septin cytoskeleton in plasma membrane repair in HeLa cells perforated by the pore-forming toxin listeriolysin O (LLO). The authors performed a silencing RNA screen targeting protein-coding genes involved in endocytosis, exocytosis and intracellular trafficking. Besides the recovery of proteins that were previously identified to be part of the membrane repair machinery, they uncovered novel plasma membrane repair candidates, including septin 7 (SEPT7).

They found that upon LLO treatment, septins redistribute from actin stress fibers to the cell

surface where they form knobs and loops together with F-actin, Myosin-IIA and Annexin A2 (ANXA2). Using super resolution microscopy and 3D reconstruction, they showed that these structures often protruding from the cell surface are formed by septins and F-actin that are organized in intertwined filaments associated with Annexin A2, and that they are functionally correlated with plasma membrane repair efficiency. Silencing SEPT7 further revealed that the remodeling of the repair protein ANXA2 at the cell surface is greatly decreased in LLO-injured cells, whereas the down regulation of ANXA2 had no impact on the arrangement of septins and F-actin into knobs and loops.

Altogether, their results evidenced that the septin cytoskeleton triggers the organization of membrane domains containing the actomyosin cytoskeleton and ANXA2, that are essential for the repair to occur.

Thank you for the great summary!

Major comments:

1- The authors show that silencing SEPT6 or SEPT7, but not SEPT2 or SEPT9, perturbed plasma membrane repair of LLO-injured cells. Yes.

The authors explain this result by indicating that the reduced expression of SEPT7 and SEPT6 (according to the siRNA), but not that of SEPT2 results in a reduced expression of septins from other groups.

This comment is accurate for SEPT7. However, our data do not support that SEPT6 silencing reduces the expression of other septins, and the article does not mention that silencing SEPT6 resulted in reduced expression of septins from other groups.

This could have been an explanation but, in Fig. S2B (now Figure EV1C), downregulating SEPT2 clearly seem to impact the expression of SEPT6 and SEPT7 (except for siRNA#3) once normalized with the loading control tubulin. Moreover, it is well accepted in the literature and has been observed in many cell types, including HeLa cells, that knocking down a septin from one group (with sometimes the exception of septins of Group 3) induces the downregulation of septins of the other groups, and that it consistently results in the loss of septin filaments. Therefore, the fact that silencing SEPT2 does not perturb plasma membrane repair is quite surprising. This could suggest that SEPT6 and SEPT7, independently of their filament organization, play a role in membrane repair after LLO treatment. Nevertheless, the SEPT2 staining to study the fate of septin filaments following LLO exposure indicated that it is the redistribution of septin filaments that is crucial in this repair process.

We reported that silencing SEPT2 had little effect on the expression of other septins such as SEPT7 (page 6, lines 140-141; Fig. EV1C). If requested, we can include additional examples of western blots. Quantification of western blots showed that SEPT7 expression was only decreased in the range of 10 to 30% in SEPT2 KD cells, which is not sufficient to abolish the formation of the septin filaments. These results are consistent with findings reported by Dr. Spiliotis lab (leading expert in septin biology) "Depletion of SEPT2 results only in a modest decrease of other septins." (PMID: 21737677). The same article showed that another member of the SEPT2 subgroup, such as SEPT5, needs to be depleted together with SEPT2 to strongly affect SEPT7 expression. An article from another group showed that knocking down SEPT2 expression had no noticeable effect on SEPT7 expression in HeLa cells (PMID: 27215606). Therefore, it is not surprising that SEPT2 silencing in our experiments does not perturb septin filament formation, and thus does not affect plasma membrane repair. To clarify this important

point, we included fluorescence images of SEPT2 and SEPT7 knocked down cells in comparison to cells exposed to non-targeting siRNAs (new Figure EV 1G). These images show that formation of septin filaments was severely decreased in SEPT7-, but not in SEPT2-knocked down cells. In SEPT2 knocked down cells, septin filaments were only slightly decreased (page 6, lines 140-141).

Interestingly, BORG proteins which are involved in the association of septin filaments to the actin cytoskeleton in interphase cells bind to the SEPT6/SEPT7 coiled-coil region of septin polymers. Could these proteins be involved, knowing that they are Cdc42 effector proteins, and that links exist between Cdc42 activation and Ca²⁺ entry?

OPTIONAL: silencing BORG proteins (BORG2 for example) and studying septin and F-actin remodeling following LLO exposure could help the authors to understand the reason of such a redistribution.

BORG proteins might be involved in remodeling the septins during plasma membrane repair; we thank the reviewer for this great suggestion. As the reviewer mentions, this would be optional. Our article reports for the first time: (i) the role of the septins in plasma membrane repair of mechanical and toxin-induced plasma membrane wounds, (ii) the septin redistribution during repair and (iii) provides mechanistic information about the structural role of the septins in remodeling the actomyosin cytoskeleton and recruitment of repair molecule ANXA2 during plasma membrane repair. We feel that the search for the proteins that anchor the septins to the plasma membrane during cell repair will be the object of future studies.

2- What about the terms "knob" and "loop": Are they structurally related to the "specks" described in other papers? Or are they new structures that no one observed before? Nobody has never looked at septins in this repair process before, but actin has long been described to be involved.

We do not know what is meant by "specks", a reference would have been helpful. Previous studies, including our studies, showed that LLO and other cholesterol-dependent cytolysins remodel the cortical actin cytoskeleton in a calcium-dependent fashion, but this redistribution had not been attributed to cell repair (PMID: 17301241, PMID: 29187576). As explained by the reviewer, nobody has ever looked at the structure of the septins and co-localization between septins and cortical actin during plasma membrane repair; this explains why these structures have not been previously noticed.

3- It seems that knobs are formed before loops take over. This would deserve further investigation. Is that a reality? Or are they two independent structures?

- OPTIONAL: it would be interesting to do time-lapse video microscopy to follow the fate of a knob.

As written in the discussion: "The knobs may be an early stage of the loop formation based on the timing of their respective appearance (Figure 2C)". Future studies will investigate septin dynamics during plasma membrane repair.

Related to the previous point: why to show 3 sets of images in the LLO condition in Fig. 2A. We report for the first time the redistribution of the septin cytoskeleton during plasma membrane repair. We thought that several images were required to properly document septin redistribution: (1) Fig. 2Aa (no LLO) and 2Ab (+LLO) are single-plan images focused on the bottom adherent membranes where actin stress fibers are most abundant to anchor cells to the culture dish. These pictures highlight that a fraction of septins dissociates from the actin stress fibers. (2) Fig. 2Ac and 2Ad are 3D reconstructions from z-stacks to show the overall redistribution of the

septin cytoskeleton in association with the actin cytoskeleton in the entire cell including upper plasma membranes.

Does the top b-panel represent the knob stage? The top b panel shows that a fraction of septins dissociated from actin stress fibers and shows several examples of knob structures. We also added a zoomed-in image of a knob structure in Fig. 2Abi.

Where there are still many stress fibers indicating that septins have not yet fully redistributed? We report that septins partially dissociate from actin stress fibers. Actin stress fibers are still present in LLO-exposed cells and are still associated with some septins, as shown in Figs. 2Ab and 2Abii and Figs. EV2Abi, 2Bbii.

And when septins are fully dissociated from actin cables, which are then lost, loops are forming (middle c-panel) and then increase in size (bottom d-panel)? Our data do not show that the septins fully dissociate from actin stress fibers and do not show that actin stress fibers were lost in LLO-treated cells. Also, we do not claim that the septin loops increase in size over time. Our images showed the presence of actin stress fibers in LLO-injured cells and a smaller proportion of septins associated with the actin stress fibers compared to control cells (Figs. 2Ab, 2Abii, Figs. EV2Ab, 2Abi, 2Bb, and 2Bbii). We report that the septins partially dissociate from the actin stress fibers in LLO-injured cells. For scientific rigor, our fluorescence images are presented with the same scaling across different experimental conditions (with or without LLO exposure). Relative to control cells (no LLO), there is a marked decrease in septin fluorescence intensity in LLO-injured cells. However, this does not mean that ALL septins dissociated from the actin stress fibers. As shown by increasing the display of the septin fluorescence intensity in zoomed images, there are still some septins in association with the F-actin stress fibers (Fig. 2Abii). This is now clarified on page 7 lines 200-202 and in Figure 2 legend.

4- Even though, some information is given in the discussion section, it would be helpful to mention in the introduction section the different pathways that cells activate to repair plasma membrane defects, and to precise which one(s) has(ve) already been described in the literature to be switched on in response to the LLO toxin.

We added a sentence in the introduction on page 3, lines 72-77.

5-Some experiments are not rigorous enough: Sometimes, they have not been repeated, as exemplified in Fig. 7B. Count less cells but repeat the experiment at least three times.

All of our conclusions were made based on data obtained from the average of at least three independent experiments ($N \geq 3$) at time point 15 min. We presented data at time points 5 and 10 min to show that septins redistribute overtime, these time points were counted with $N=1$ and were representative of several experiments. We now have enumerated the septin redistribution at time points 5 and 10 min with $N=3$ in Figs. 2B, C. Sometimes data were presented with $N=2$ or 1 for additional, non-essential qualitative information. For example, the septin cytoskeleton is made of the co-assembly of SEPT 2, 7 and 9, and septin filaments can be indiscriminately visualized using anti-SEPT2, anti-SEPT7, or anti-SEPT9 antibodies as shown in Fig. 2, Fig. EV2. For example, in Fig. 2C, we enumerated septin redistribution using anti-SEPT2 Abs and, in parallel, using anti-SEPT7 Abs, both with $N=3$ leading to highly similar data. All quantitative analyses were performed by counting cells labeled with anti-SEPT2 antibodies ($N \geq 3$). In Fig. 7B, we presented additional data counted on cells labeled with anti-SEPT7 and anti-SEPT9 antibodies, which showed again similar results as with cells labeled with anti-SEPT2 antibodies, as expected. Therefore, it would make little sense to repeat counting cells labeled with anti-SEPT9 or anti-SEPT7 antibodies in Fig. 7B. In Figs. 2B, 2C, and Fig. 3C. We also counted $N=3$ for ALIX colocalization at 15 min in Fig. 3C

Sometimes, one condition is missing, as in Fig. 6C. Where is the DMSO condition? What about the statistics?

Fig. 6C: the DMSO control was added to the figure. All statistical analyses of all data are presented in the methods and figure legends.

Fig. 6A: In the calcium-free condition, it seems that the two cells that are illustrated depict a telophase. The subcellular organization is obviously different at the end of cell division. Show only one interphase cell as in the top panels.

We do not think that the cells depict a telophase. Cell morphology is affected by excessive damage in the presence of LLO in calcium-free medium. To avoid any confusion, we selected another representative image in Fig. 6A.

Fig. 6B and C: It is mentioned in the figure legend: "Cells were treated as indicated in (A) and (D)". But the "C" condition is not mentioned anywhere: Does "C" stand for no DMSO or FCF treatment, in the presence of calcium, but under LLO treatment?

We clarified the figure and the figure legend in Fig. 6B and C. Yes, C was the condition without DMSO or FCF and with calcium.

Likewise, it would be very helpful to indicate in each figure panel whether cells have been treated with LLO or FCF. Please help the reader.

We clarified all figure panels and legends.

Fig. 7A: Whatever SEPT7 is expressed or downregulated, the actin stress fibers are still present. If these cells were not well transfected, replace the images.

All presented images are representative. There were less actin stress fibers in SEPT7-silenced cells, but stress fibers were still present. Please see Fig. 7A; Fig. EV1G and Fig. EV5A-C.

6- Concerning the FCF experiments: The FCF cytokinin has been used by many authors to perturb septin dynamics. It induces the stabilization of septin polymers, thus promoting the formation of thick ectopic fibers. It is a potent inducer of septin polymerization and acts as a stabilizer.

Fig. 6D: In the Ctr condition, a DMSO condition is needed to visualize the impact of FCF on septin filaments.

An additional image corresponding to the control DMSO was added in Fig. EV4

Does FCF stabilize septins and induce the formation of thick filaments?

This is correct, we observe thicker septin filaments in FCF-treated cells, as shown in Fig. EV4

The SEPT2 image in the FCF condition without LLO is of bad quality (see above remark).

Corresponding images were replaced, and a new Fig. EV4 was added.

Also in Fig. 6D (FCF condition without LLO), the F-actin staining revealed that there are no stress fibers!?!???

There are actin stress fibers in FCF-treated cells, to make this point more evident, we replaced images of FCF-treated cells (without LLO) in the new Fig. EV4.

Usually, the more septins are associated with actin, the thicker stress fibers you get, since septins stabilize actin cables. FCF treatment often induces thick ectopic septin filaments that are not associated with stress fibers (which are therefore lost). Was it the case in all FCF-treated cells?

We did not observe a large number of ectopic septin filaments and septins were mostly associated with F-actin in FCF-treated cells. We did lose some stress fibers (stress fibers were less frequent) but not in all cells. Images in new Fig. EV4 show less stress fibers, which is

consistent across all experiments. Cortical actin was still very much present in the loop structures. A lot of the FCF-treated cells had “thicker bundles” of septins located either in the middle in a clump or along the edges of the cells.

Does FCF treatment really mimic what happens physiologically in the cell? Many off-target effects have been observed with this molecule in non-plant cells.

FCF is the only drug known to specifically affect septin filaments. FCF was used as an alternative approach to affect septin organization. As with other drugs, we cannot exclude potential off-target effects, this is the reason why we used diverse experimental approaches. Importantly, FCF increases the formation of septin loops which were observed in the absence of FCF. Also, FCF increased formation of septin loops only in damaged cells (both mechanically damaged and injured by LLO). Collectively, these findings support that stabilization of the septin cytoskeleton by FCF stabilizes the septin structures formed during cell repair.

7- The image quality in Figs 3A and B, and 6A and D needs to be improved regarding the septin staining. In control conditions, septin filaments cannot be clearly distinguished.

Septin filaments are clearly seen in Fig. 3A, we replaced the No LLO ANXA2-GFP cells in Fig. 3B. Images in 6A and D were also replaced.

8- Fig. 3B: It seems that ANXA2 is overexpressed in LLO-injured cells. Its accumulation level between both conditions should be compared by immunoblot.

ANXA2 is not overexpressed within the time frame of the experiment (15 min). The difference observed in control versus LLO-treated cells comes from the redistribution of ANXA2. ANXA2 is diffusively dispersed in non-injured control cells and is redistributed to accumulate at the plasma membrane in LLO-injured cells leading to a brighter fluorescence. This is a common phenomenon observed when cytosolic proteins are recruited to the plasma membrane or organelles, and ANXA2 is well-known to be recruited to the plasma membrane of injured cells. ANXA2 is indeed recovered on loops, but it is difficult to consider whether it is a redistribution. The loops were not formed on non-injured cells. Thus, the presence of ANXA2 on the newly formed septin loops can only correspond to ANXA2 redistribution.

9- Fig. 7D: Compared to the control condition (we have to refer to Fig. S5D, now Fig. EV5D), ANXA2 again seems to be overexpressed under LLO treatment.

ANXA2 is not overexpressed, see comment above.

To affirm that ANXA2 remodeling in LLO-injured cells requires the formation of septin/F-actin knobs and loops, data in Fig. 7D must be quantified.

Figure 7D shows ANXA2 silenced cells, therefore we cannot measure ANXA2 remodeling in this experimental condition. We quantified ANXA2 remodeling in SEPT7-deficient cells in comparison to cells treated with non-targeting siRNA, this result is presented in Figure 7C.

10- Fig. 6 (B-D): In panel B, there is a significant difference between the "C" and "FCF" conditions regarding the number of knobs + loops per cell. Where are the images corresponding to the "C" condition?

Images corresponding to the “C” were added to Fig. 6. We also clarified the legends of this figure and added control images corresponding to the same experimental conditions without exposure to LLO in the new Fig. EV4. These modifications were necessary to improve clarity.

Minor comments:

11- Fig. 2A: Report the white squares (selected enlarged areas) in all panels (SEPT2 and overlay). In panels b, do not place an arrowhead where we are supposed to observe an enlarged area. Also, from panels b, it would be worth showing an enlarged area including a knob. Show enlarged areas also from panels d.

White squares were added. We also added magnified images to highlight the knob structures (2Abi) and additional loop structures (2Adi).

12- Fig. 3B'i: Septins are not on stress fibers. Select a transfected cell where septins still coalign with actin fibers, not a cell that was impaired by the transfection.

Image 3B was changed, its corresponding enlarged panel shows the co-alignment of the septins with F-actin in 3Bi'.

13- Fig. 3C: Add the time point "0min". What was the % of colocalization before LLO treatment? and in DMSO condition?

At the time points 0, 5, 10 and 15 min, in the absence of LLO, ANXA2 is not recruited to the plasma membrane and the septin knobs and loops are not formed. This is why we presented only the time point 15 min without LLO. There is no DMSO in these experiments.

What about ALIX at 5 and 10min? Again, it's only one experiment.

We observed minimal colocalization between ALIX and the septins at all studied time points in multiple experiments. However, we noticed a very clear and high level of co-localization between ANXA2 and the septins. This is the reason why we focused on studying the co-localization between ANXA2 and the septins, the topic of this article. We show that ALIX displays poor co-localization with the septins at all time points, counting cells from one of our representative experiments. We counted more experiments to reach N=3 at 15 min (Fig. 3C). Of note, only conclusions were made on ANXA2, we do not make any conclusions about the ESCRT-III (ALIX) in this article.

14- Figs 4 and 5: Very nice images but obtained following FCF exposure. Hopefully FCF would not have induced an aberrant organization!

Thank you. We showed numerous images of septin redistribution in the presence and in the absence of FCF throughout the article. In both experimental conditions, septins redistribute to form knobs and loops only on damaged cells. The difference with the FCF treatment was quantified in Fig. 6, which shows the more pronounced redistribution of the septins in FCF-treated cells, in accordance with the stabilization of septins by FCF. Collectively, our data show that their formation is not an aberrant effect of FCF, but FCF increased their formation or their stabilization.

15- The "Ctr" abbreviation is often used, in different conditions, and may be confusing. Precise in the figure (not in the figure legend) whether it is siRNA ("Ctr siRNA"). Mention "DMSO" for the controls of your drugs (like in Figs S4 and S5, now S5 and S6).

We clarified all control abbreviations in all figures.

16- Fig. S4C: How is this figure different or does it provide additional information compared to Fig. 2C?

We agree, this figure was not essential and was removed.

17- Fig. S5D (now Fig. EV5D): It is hard to know that the ANXA2 siRNA worked, since no difference of staining between the Ctr and the transfected cells can be observed. Were these cells really transfected? It would have been helpful to use fluorescent siRNAs.

ANXA2 silencing was effective, we quantified 88% silencing with siRNA3 by western blot in Fig. 1I.

The same applies to Fig. S5C (now Fig. EV5B): Silencing SEPT7 supposedly greatly reduces the level of expression of all septins. The SEPT2 staining is still high, and many actin stress fibers are still observable (whereas the loss of septin filaments results in the loss of actin stress fibers, as observed by many authors, including in HeLa cells). Same remark for Fig. S6 (now Fig. EV5C), regarding the SEPT7 silencing in the Ctr condition (no LLO). No impact on stress fibers! Are these cells transfected? The authors themselves mention that sometimes cells are less effectively silenced (like in Fig. 7A, B, still Fig. 7A, B). Why not to show cells effectively silenced!!

As mentioned in our responses above, SEPT7 silencing reduces, but does not fully abolish, the expression of other septins as we showed by western blotting and imaging. F-actin stress fibers were still present in SEPT7 silenced cells. The presented images are representative of our experimental data.

18- In the abstract, it is specified that SEPT7 also plays a role in membrane repair after mechanical wounding. Based only on one type of experiment (SEPT7 silencing, Fig. 1H, now Fig. 1J), this statement should only be mentioned in the text or used to discuss the putative repair mechanisms that septins are involved in, but not stated in the abstract as a main conclusion.

The assay used to assess repair of mechanical damage is a reliable assay commonly used in the literature. This assay is sufficient to report that septins are required for the repair of mechanical wounds. Importantly, we included ANXA2-silenced cells as a positive control to validate our assay, as shown previously in the literature. Furthermore, we added a new figure (new Fig. EV2C) showing the redistribution of the septins in mechanically wounded cells. This new figure shows that the redistribution occurs at the site of mechanical damages, and FCF enhanced the redistribution of septins in mechanically wounded cells. Collectively, we feel confident to write in the abstract that the septin cytoskeleton is required for the repair of mechanical wounds.

Reviewer #2 (Significance (Required)):

2. SIGNIFICANCE

Strengths:

Despite septins have been involved in endocytosis, exocytosis, membrane protrusions, cell junction integrity or actomyosin constriction at cytokinesis, the involvement of the septin cytoskeleton in the plasma membrane repair machinery has, to my knowledge, never been reported before. The authors not only showed that septins are present in specific membrane protrusions (knobs and loops) but also evidenced that septin filaments trigger the formation of these plasma membrane repair domains by recruiting F-actin and ANXA2, essential for the repair to occur. The novelty of this study has therefore to be acknowledged, and these data will

benefit the scientific community, and the septin community in particular.

This is a descriptive paper that nevertheless clearly shows, by different means, the reorganization of the septin cytoskeleton in LLO-injured cells. The use of high-resolution microscopy coupled to 3D reconstruction which enables to easily appreciate the organization of septins, F-actin and ANXA2 in the knobs and loops is a true strength of the paper.

Limitations:

The authors mention in the abstract that septins act as scaffolds to recruit contractile actin fibers and ANXA2. Biochemical experiments such as co-immunoprecipitations could strengthen this notion.

The last sentence of our abstract was as follows: *“Collectively, our data support a novel model in which the septin cytoskeleton acts as a scaffold to promote the formation of plasma membrane repair domains containing contractile F-actin and annexin A2.”*

We believe that the presented data strongly support this proposed model. Indeed, we demonstrated that (i) septins are required for plasma membrane repair; (ii) septins are required for the remodeling of the submembranous actin cytoskeleton to form intertwined actin/septin filaments in close association with the repair protein ANXA2, as shown by super-resolution microscopy; and (iii) septins are also required for the recruitment of ANXA2. Finally, ANXA2 is well-known to be critical for the repair of transmembrane pores and the repair of mechanical wounds. Collectively, these results strongly support the structural scaffolding role of the septin cytoskeleton in membrane repair domain formation. Anyhow, the term “scaffold” is not included in the shorten version of the abstract.

The molecular mechanism by which septins are involved in this repair process has not been addressed at all in the paper.

See comments above, our studies are mechanistic and address the mechanism by which the septins are involved in repair: they control the formation of subdomains of the plasma membrane for the recruitment of molecules that repair the plasma membrane such as annexin A2.

Even though the silencing RNA screen highlighted several proteins involved in known membrane repair mechanisms, the authors just presented a few data concerning ALIX, a component of the ESCRT-III machinery. A % of colocalization of SEPT2 and SEPT7 with ALIX is reported in Fig. 3C but this experiment has only been done once (n=1) and only following 15-min exposure to LLO. Is that too late? Immunofluorescence images of SEPT2 and ALIX with or without LLO (15min) are also provided in Fig. S5A but no quantification is reported. Is it sufficient to say that the ESCRT machinery is not involved?

We reply to the comment about ALIX co-localization in point # 13 above. Our article did not make conclusion about the ESCRT-III machinery. We understand the enthusiasm of the reviewer in asking additional questions. We do as well. However, one research article is not sufficient to discover everything about the novel role of the septins in plasma membrane repair. We did not observe a co-localization with ALIX as we did with ANXA2, but we did not conclude that ESCRT-III machinery is not involved. In addition, we performed additional fluorescence labeling of the protein S100A11 (new Fig. EV3B), previously demonstrated to act together with ANXA2 and the actin cytoskeleton to repair mechanically damaged cells. This colocalization further supports our hypothesis of the role of the septins in organizing F-actin and ANXA2-dependent plasma membrane repair domains.

Dear Stephanie,

Thank you for the submission of your revised manuscript to EMBO reports. It was evaluated by former referee 2, who now supports publication after minor revisions to clarify text, figures and a few control experiments.

From the editorial side, there are also a few things that we need before we can proceed with the official acceptance of your study.

- Please reduce the number of keywords to 5.
- Please change the header "Conflict of Interest" to "Disclosure and Competing Interests Statement "
- MATERIALS and METHODS should be METHODS.
- References need to be alphabetical, 'et al' should be used after 10 author names.
- Please note that "data not shown" is not permitted. All conclusions made in the manuscript based on data require that these data are included in the manuscript. In this respect we note 2 instances of "data not shown" (page 6 and page 9). This needs to be rectified.
- We need separate figure files for all main and EV figures.
- Appendix Tables 1 and 2 are quite complex and should be uploaded as Dataset EV1 and Dataset EV2 (.xls file, legend in first tab). Please also correct the callouts in the text.
- Each movie needs to be zipped up with its legend and uploaded as separate zip folders; the legends need a title beginning with "Movie EV1" and "Movie EV2".
- Callouts for the movies need to be changed to "Movie EV1" and "Movie EV2".
- Table EV1: the legend needs to be corrected to Table EV1 instead of Table Expanded View 1.
- Our production/data editors have asked you to clarify several points in the figure legends (see below). Please incorporate these changes in the manuscript and return the revised file with tracked changes with your final manuscript submission.
 - A) Please note that a separate 'Data Information' section is required in the legends of figures 1b-d, f-h, j; 7b-c; EV 1a', b; EV 3a-b. (The Data Information section is used to define features that apply to several panels in the figure, e.g., Data Information: In (B-D), data are presented as mean +/- SEM, p values xxx).
 - B) Please indicate the statistical test used for data analysis in the legends of figures 1b, d, f-g; 6b-c; EV 1a', b.
 - C) Please note that in figures 1b-d, f-h, j; there is a mismatch between the annotated p values in the figure legend and the annotated p values in the figure file that should be corrected.
 - D) Although 'n' is provided, please describe the nature of entity for 'n' in the legends of figures 1b-d, f-h; EV 1a', b, h, k.
 - E) Please note that the error bars are not defined in the legends of figures 1j; 3c; 6b-c; EV 1h, k.
- We routinely screen source data .xls files with a software that detects duplications of numbers. Can you please check the two attached .xls files that show such potential duplications (either labeled in red or in blue) and clarify these? Thank you.
- Finally, EMBO Reports papers are accompanied online by
 - A) a short (1-2 sentences) summary of the findings and their significance,
 - B) 2-3 bullet points highlighting key results and
 - C) a schematic summary figure that provides a sketch of the major findings (not a data image). Please provide the summary figure as a separate file in PNG or JPG format at a size of 550x300-600 pixels (width x height). Please note that the size is rather small and that text needs to be readable at the final size. Please send us this information along with the revised manuscript.
- On a different note, I would like to alert you that EMBO Press offers a new format for a video-synopsis of work published with us, which essentially is a short, author-generated film explaining the core findings in hand drawings, and, as we believe, can be very useful to increase visibility of the work. This has proven to offer a nice opportunity for exposure i.p. for the first author(s) of the study. Please see the following link for representative examples and their integration into the article web page:

<https://www.embopress.org/doi/full/10.15252/emj.2019103932>

With kind regards,

Martina

Referee #2:

The authors have answered most of the points raised, which has greatly improved and strengthened their manuscript, particularly regarding quantification of data (Fig. 2B-C, Fig. 3C, Fig. 6B-C) and clarification of figure headings and legends. They also added further convincing data like the Emerald Dextran experiments to link their septin redistribution phenotype with damaged cells (Fig. EV2C) or the immunolabeling of the S100A11-M protein (Fig. EV3). They have shortened the abstract, which is now more streamlined and straight to the point.

Nevertheless, in line with these improvements, some points still have to be addressed to help understand the figures at first glance:

- Fig. 1B: given the data points of siRNA 3, it is surprising that its effect is as significant as that of siRNAs 1 and 2. P is really <0.0001 ?
- Fig. 1H and J: add "siRNA" at the x-axis
- Fig. 2 is much clearer. Nevertheless, it would be interesting to also add 3D reconstruction images for the condition without LLO. As the loops seem to be rather localized to the upper plasma membrane, this would indeed be the real control to show that they are absent in this control condition.
- Figs. 3B, 7A and EV2A and B: please add the white squares to all the panels as you did in Fig. 2
- Fig. 3C: add "of LLO treatment" at the x-axis
- Fig. 6D : add "in the presence of Ca" after LLO (15min)
- Fig. 7B: also report the legend SEPT7 siRNA and Control siRNA (as in panel C)
- Fig. 7C: write "ANXA2 specks/Cell" at the y-axis
- Fig. 7D: it appears from the illustrated images that, under LLO treatment, more knobs and loops are formed in the Ctr. siRNA versus ANXA2 siRNA3, which is not what is stated in the text (lines 286-287). These data must be quantified (as for the SEPT siRNAs) to consider more than 2 or 3 cells.
- Line 298: do not write "specific" because it was shown that FCF has off target effects. Rather write "septin-binding, septin-targeting or septin-stabilizing molecule"
- Lastly, regarding the ANXA2 level of expression, it is right that it is hard to believe that ANXA2 is overexpressed within the time frame (15min) of the experiment. While a redistribution (from cytoplasmic puncta to loops) can clearly be seen for S100A11 (Fig. EV3B), it is difficult to assess such a relocalization for ANXA2. We do not observe a reduction in cytoplasmic signal in favor of increased signal in specific structures following LLO treatment. Adding a Western blot showing the accumulation of ANXA2 in the absence and presence of LLO would definitively solve this ambiguity.

Referee #2

The authors have answered most of the points raised, which has greatly improved and strengthened their manuscript, particularly regarding quantification of data (Fig. 2B-C, Fig. 3C, Fig. 6B-C) and clarification of figure headings and legends. They also added further convincing data like the Emerald Dextran experiments to link their septin redistribution phenotype with damaged cells (Fig. EV2C) or the immunolabeling of the S100A11-M protein (Fig. EV3). They have shortened the abstract, which is now more streamlined and straight to the point.

Nevertheless, in line with these improvements, some points still have to be addressed to help understand the figures at first glance:

- Fig. 1B: given the data points of siRNA 3, it is surprising that its effect is as significant as that of siRNAs 1 and 2. P is really

<0.0001?

- Fig. 1H and J: add "siRNA" at the x-axis
- Fig. 2 is much clearer. Nevertheless, it would be interesting to also add 3D reconstruction images for the condition without LLO. As the loops seem to be rather localized to the upper plasma membrane, this would indeed be the real control to show that they are absent in this control condition.
- Figs. 3B, 7A and EV2A and B: please add the white squares to all the panels as you did in Fig. 2
- Fig. 3C: add "of LLO treatment" at the x-axis
- Fig. 6D : add "in the presence of Ca" after LLO (15min)
- Fig. 7B: also report the legend SEPT7 siRNA and Control siRNA (as in panel C)
- Fig. 7C: write "ANXA2 specks/Cell" at the y-axis
- Fig. 7D: it appears from the illustrated images that, under LLO treatment, more knobs and loops are formed in the Ctr. siRNA versus ANXA2 siRNA3, which is not what is stated in the text (lines 286-287). These data must be quantified (as for the SEPT siRNAs) to consider more than 2 or 3 cells.
- Line 298: do not write "specific" because it was shown that FCF has off target effects. Rather write "septin-binding, septin-targeting or septin-stabilizing molecule"
- Lastly, regarding the ANXA2 level of expression, it is right that it is hard to believe that ANXA2 is overexpressed within the time frame (15min) of the experiment. While a redistribution (from cytoplasmic puncta to loops) can clearly be seen for S100A11 (Fig. EV3B), it is difficult to assess such a relocalization for ANXA2. We do not observe a reduction in cytoplasmic signal in favor of increased signal in specific structures following LLO treatment. Adding a Western blot showing the accumulation of ANXA2 in the absence and presence of LLO would definitively solve this ambiguity.

Rev_Com_number: N/a

New_manu_number: EMBOR-2024-59290V2

Corr_author: Seveau

Title: The Septin Cytoskeleton is Required for Plasma Membrane Repair

Referee #2:

We are thankful to reviewer 2 for taking the time to review our article. Below are our point-by-point answers to all questions including requested additional experiments (our answers are in blue font).

The authors have answered most of the points raised, which has greatly improved and strengthened their manuscript, particularly regarding quantification of data (Fig. 2B-C, Fig. 3C, Fig. 6B-C) and clarification of figure headings and legends. They also added further convincing data like the Emerald Dextran experiments to link their septin redistribution phenotype with damaged cells (Fig. EV2C) or the immunolabeling of the S100A11-M protein (Fig. EV3). They have shortened the abstract, which is now more streamlined and straight to the point.

Nevertheless, in line with these improvements, some points still have to be addressed to help understand the figures at first glance:

- Fig. 1B: given the data points of siRNA 3, it is surprising that its effect is as significant as that of siRNAs 1 and 2. P is really <0.0001 ? As confirmed by our statistician who repeated the statistical analyses, P value is <0.0001 for the SEPT7 siRNA 3 experimental condition. The magnitude of the difference between siRNA 3 and the control non-targeting siRNA is not as large as for SEPT7 siRNA 1 and 2; however, SEPT7 siRNA 3 is consistently higher for each experimental replicate compared to the control in the log-transformed data.

- Fig. 1H and J: add "siRNA" at the x-axis "siRNA" was added.

- Fig. 2 is much clearer. Nevertheless, it would be interesting to also add 3D reconstruction images for the condition without LLO. As the loops seem to be rather localized to the upper plasma membrane, this would indeed be the real control to show that they are absent in this control condition. A 3D reconstruction of LLO-treated versus control cells with a z-depth code was added to Figure 4B. In this figure, it is clear that septins redistribute to form protruding loops and knobs on the upper surface of the LLO-treated cells but not in control cells.

- Figs. 3B, 7A and EV2A and B: please add the white squares to all the panels as you did in Fig. 2. White squares were added to the images that correspond to a zoomed-in region.

- Fig. 3C: add "of LLO treatment" at the x-axis. We added "LLO Exposure Time (min)"

- Fig. 6D : add "in the presence of Ca" after LLO (15min).

We added "1.2 mM Ca^{2+} "

- Fig. 7B: also report the legend SEPT7 siRNA and Control siRNA (as in panel C). We added the legend for SEPT7 and the Control siRNA.

- Fig. 7C: write "ANXA2 specks/Cell" at the y-axis. The Y axis was changed to ANXA2 Specks/ Cell

- Fig. 7D: it appears from the illustrated images that, under LLO treatment, more knobs and loops are formed in the Ctr. siRNA versus ANXA2 siRNA3, which is not what is stated in the text (lines 286-287). These data must be quantified (as for the SEPT siRNAs) to consider more than 2 or 3 cells. We quantified the numbers of knobs and loops formed in ANXA2 (siRNA3) knocked down cells in comparison to cells

treated with the non-targeting siRNA. The results are included in Figure 7B (N=4 independent experiments). As we previously observed, knocking down expression of ANXA2 had no significant effect on the formation of the septin loops and knobs in LLO-treated cells.

- Line 298: do not write "specific" because it was shown that FCF has off-target effects. Rather write "septin-binding, septin-targeting or septin-stabilizing molecule". This was changed to "septin-targeting"

- Lastly, regarding the ANXA2 level of expression, it is right that it is hard to believe that ANXA2 is overexpressed within the time frame (15min) of the experiment. While a redistribution (from cytoplasmic puncta to loops) can clearly be seen for S100A11 (Fig. EV3B), it is difficult to assess such a relocalization for ANXA2. We do not observe a reduction in cytoplasmic signal in favor of increased signal in specific structures following LLO treatment. Adding a Western blot showing the accumulation of ANXA2 in the absence and presence of LLO would definitively solve this ambiguity. As previously answered, this labeling pattern is typical of cytosolic proteins that are recruited to the plasma membrane. As requested, we performed western blotting analysis of ANXA2 in cells treated for 15 min with LLO in comparison to control cells incubated in the absence of LLO (Figure EV3B). The data demonstrate that ANXA2 expression was not increased in this timeline, as expected. If anything, in LLO-treated cells, ANXA2 expression was slightly decreased, likely due to the shedding of ANXA2-containing vesicles during plasma membrane repair.

Dr. Stephanie Seveau
Ohio State University
Microbial Infection and Immunity
460 W 12th
Columbus, OH 43210
United States

Dear Stephanie,

Thank you once more for submitting your revised article and please apologise my delayed response, which is due to travel. After checking all files, I am very pleased to accept your manuscript for publication in the next available issue of EMBO reports. Thank you for your contribution to our journal.

Kind regards,

Martina

Rev_Com_number: N/a
New_manu_number: EMBOR-2024-59290V3
Corr_author: Seveau
Title: The Septin Cytoskeleton is Required for Plasma Membrane Repair